# Role of wind, mesoscale dynamics and coastal circulation in the interannual variability of South Vietnam Upwelling, South China Sea. Answers from a high-resolution ocean model.

Thai To Duy[1,2,3], Marine Herrmann[1,2*], Claude Estournel[1], Patrick Marsaleix[1], Thomas Duhaut[1], Long Bui Hong [3], Ngoc Trinh Bich [2]

[1] LEGOS, IRD, UMR5566, IRD/CNES/CNRS/Université de Toulouse, 31400 Toulouse, France

[2] LOTUS Laboratory, University of Science and Technology of Hanoi (USTH), Vietnam Academy of Science and Technology (VAST), 18 Hoang Quoc Viet, Cau Giay, Hanoi, Vietnam

[3] Institute of Oceanography (IO), Vietnam Academy of Science and Technology (VAST), Nha Trang, Vietnam

*Correspondence to :* Marine Herrmann (marine.herrmann@ird.fr)

Short summary

The South Vietnam Upwelling develops in the coastal and offshore regions of the southwestern South China Sea under the influence of summer monsoon winds. Cold, nutrient-rich waters rise to the surface, where photosynthesis occurs, and is essential for the fishing activity. We have developed a very high-resolution model to better understand the factors that drive the variability of this upwelling at different scales: daily chronology to summer mean of wind, mesoscale to regional circulation.

Abstract.

The South Vietnam Upwelling (SVU) develops in the South China Sea (SCS) under the influence of southwest monsoon winds. To study the role of small spatiotemporal scales on the SVU functioning and variability, a simulation was performed over 2009-2018 with a high-resolution configuration (1 km at the coast) of the SYMPHONIE model

implemented over the western region of the SCS. Its capability to represent ocean dynamics and water masses from daily to interannual scales and from coastal to regional areas is quantitatively demonstrated by comparison with available satellite data and four in-situ datasets. The SVU interannual variability is examined for the three development areas previously known: the southern (SCU) and northern (NCU) coastal areas and the offshore area (OFU). Our high-resolution model, together with in-situ observations and high-resolution satellite data, moreover shows for the first time

that upwelling develops over the Sunda Shelf off the Mekong Delta (MKU).
Our results confirm for the SCU and OFU, and show for the MKU, the role of the mean summer intensity of wind and cyclonic circulation over the offshore area in driving the interannual variability of the upwelling intensity. They further reveal that other factors contribute to SCU and OFU variability. First, the intraseasonal wind chronology strengthens (in the case of regular wind peaks occurring throughout the summer for SCU, or of stronger winds in July-August for OFU)

or weakens (in the case of intermittent wind peaks for SCU) the summer average upwelling intensity. Second, the mesoscale circulation influences this intensity (multiple dipoles eddies and associated eastward jets developing along the coast enhances the SCU intensity). The NCU interannual variability is less driven by the regional-scale wind (with weaker monsoon favoring stronger NCU), and more by the mesoscale circulation in the NCU area: the NCU is prevented (favored) when alongshore (offshore) currents prevail.

1.  Introduction

Ocean circulation in the South China Sea (SCS), one of the largest semi-enclosed seas in the world (Figure 1), is under the influence of several factors of variability at different scales : typhoons, seasonal monsoon, interannual to decadal variability, climate change (Wyrtki 1961, Herrmann et al. 2020, 2021). SCS is moreover impacted by human activities, with highly densely populated coastal areas (CIESIN 2018), which generate problems such as marine pollution (release of industrial, agricultural or domestic contaminants, such as pharmaceuticals, pesticides, heavy metals, plastics …). Reciprocally, coastal societies are highly dependent on marine resources (fisheries and aquaculture, tourism…). SCS also influences the functioning and variability of regional climate through air-sea coupling (Xie et al. 2003, 2007, Zheng et al. 2016). It also influences global climate through its role in the transformation of surface water masses of the global ocean circulation (South China Sea Throughflow, Fang et al. 2009, Qu et al. 2009). It is therefore essential to better understand and model SCS different scales dynamics which intervene in the transport and fate of water masses, in the cycle of water and energy, in the functioning of planktonic ecosystem (via their impact on the transport and mixing of nutrients and organic matter, Ulses et al. 2016, Herrmann et al. 2017), and in the propagation of contaminants at sea.

The large-scale summer SCS circulation is usually characterized by an overall anticyclonic circulation, that results from southwest summer monsoon winds (Wang et al. 2004). It is composed more precisely of a western boundary current and a dipole structure that develops along the Vietnamese coast, with an anticyclonic gyre (AC) in the south and cyclonic gyre (C) in the north (see Figure 2l). This dipole structure induces northward/southward alongshore coastal currents which are the western components of the AC/C gyres, and a marked eastward offshore jet that develops in the area of convergence of both coastal currents and between both gyres, around ~12°N (Wyrtki 1961, Xu et al. 1982, Fang et al. 2002, Xie et al. 2003,2007, Chen and Wang 2014, Li et al. 2014, 2017).

Summer wind forcing and the resulting large-scale circulation (AC/C dipole + eastward jet) induce the South Vietnam Upwelling (SVU), one of the main components of the SCS summer ocean dynamics. The SVU strongly influences halieutic resources through its impact on bio-productivity (Bombar et al. 2010, Loick et al. 2007, Loisel et al. 2017, Lu et al. 2018). It also affects regional climate (Xie et al. 2003, Zheng et al. 2016). It is driven at the first order by atmospheric forcing, namely Ekman transport and pumping induced by the southwest monsoon wind (Xie et al. 2003). It is moreover reinforced by the eastward jet that enhances the offshore advection of coastal water (Dippner et al. 2007) and by the combination of wind-induced Ekman pumping and positive surface current vorticity associated with cyclonic eddies in the offshore area (Da et al. 2019, Ngo and Hsin 2021). Those studies showed that upwelling develops over several areas. The most studied areas are the southern coastal area and the offshore area. The southern coastal upwelling (named SCU hereafter) develops along the Vietnamese coast between 10.5°N and 12°N in the area of convergence between the northern southward and southern northward boundary currents (Figures 2l, 3). The offshore upwelling (named OFU hereafter) develops over a large offshore region, in the area of cyclonic circulation northern of the eastward jet. Two studies moreover showed that upwelling sometimes develops along the northern part of the Vietnamese coast (named NCU hereafter), from 11°N to 15°N. Da et al. (2019) revealed from AVHRR SST (sea surface temperature) satellite data and from a 1/12° simulation that upwelling developed during summer 1998 in this area, and Ngo and Hsin (2021) studied the interannual variability of NCU using gridded satellite data.

Previous field campaigns (Dippner et al. 2007, Bombar et al. 2010, Loick-Wilde et al. 2017), satellite observations (Xie et al. 2003, Kuo et al. 2004, Ngo and Hsin 2021) and modeling studies (Li et al. 2014, Da et al. 2019), showed that the

SVU intensity strongly varies from one year to another. This interannual variability depends in particular on the intensity of the summer monsoon southwest wind and on ENSO (El Niño Southern Oscillation), but also on cross-equatorial winds (Wu et al. 2019) and decadal climate variability (Wang et al. 2020). Previous studies showed that the interannual variability of the SCU is mainly triggered by the interannual variability of the summer averaged wind stress that induces Ekman transport (Chao et al. 1996, Dippner et al. 2007, Da et al. 2019, Ngo and Hsin 2021). Using long simulations or datasets, two recent studies established statistical relationships between the interannual variability of SCU and OFU intensity, the atmospheric forcing and the ocean circulation. Da et al. (2019) used a 1/12° (~9 km) 14-years simulation. Ngo and Hsin (2021) used ¼° (~28 km) datasets of 38-years SST and reanalysis winds and of 28-years satellite-altimeter derived sea surface current. Da et al. (2019) simulated an Ekman pumping-related OFU that develops under the influence of strong positive wind curl. They obtained a highly significant correlation between the yearly intensity of upwelling and the summer average of wind stress (0.70, p<0.01) and wind stress curl (0.72, p<0.01) over the region. They showed that SVU intensity is also correlated with the summer average of positive vorticity of surface current over the offshore area (0.79, p<0.01). This vorticity, which quantifies the intensity of the AC/C dipole and eastward jet circulation, partly results from the wind forcing, with a 0.73 (p<0.01) and 0.63 (p<0.02) correlation with summer average wind stress and wind stress curl, respectively. Some authors moreover showed that the wind also influences the SVU at the intraseasonal scale, with SVU developing after peaks of strong winds induced by Madden Julian Oscillation (MJO) and typhoons (Xie et al. 2007, Liu et al. 2012). Da et al. (2019) moreover revealed and quantified the important contribution of ocean intrinsic variability (OIV) in the SVU interannual variability in the offshore area. They showed that this OIV contribution is related to the role of vorticity associated with mesoscale structures, which propagation is strongly stochastic: cyclonic (anticyclonic) mesoscale eddies located in the area of positive wind curl enhance (weaken) the summer average intensity of offshore upwelling. Ngo and Hsin (2021) confirmed the relationship between the interannual variability of the summer regional wind stress curl, the induced offshore circulation (i.e. eastward jet, AC/C dipole and associated northward/southward boundary currents) and the SCU and OFU intensity : they showed that upwelling is stronger (weaker) for years of stronger (weaker) wind stress curl and large scale circulation (eastward jet and dipole). Finally, Chen et al. (2012) showed that tides and river plumes could be additional mechanisms involved in SVU dynamics, using idealized simulations. Tide effect was not included in most modeling studies, including Da et al. (2019).

Most of previous studies were therefore based on numerical or observational datasets associated with several methodological choices and hypothesis : short study periods and limited spatial coverage of in-situ datasets ; models resolution coarser than ~10 km ; cloud cover and proximity of the coast hindering satellite data quality ; gridded current datasets built from along-track sea level satellite data that can not capture small scale and ageostrophic dynamics ; synoptic view based on spatially and seasonally integrated atmospheric and oceanic variables ; no or simplified representation of tides and river plumes… This limited their capability to capture or represent the full range of scales involved in the SVU variability, that goes from submesoscale and coastal dynamics to regional circulation, and from daily to interannual variability. In particular, the settings of those modeling and observational studies did not allow them to capture the variations at small spatial scales and high frequency of ocean SST and dynamics, including the effect of tides. To improve our knowledge of SVU functioning and variability, further investigations are therefore needed to better monitor, represent and understand the behavior of upwelling at smaller spatial (meso to submesoscales) and temporal (intraseasonal variability) scales, and over detailed areas (coastal to offshore). First, this requires to understand how ocean dynamics at those smaller spatial scales, which largely contributes to OIV (Sérazin et al. 2016, Waldman et al. 2018), and their interactions with larger scale circulation contribute to the SVU functioning and variability. Second, this requires to examine how high frequency variability (daily to intraseasonal) of atmospheric forcing and ocean dynamics, including

tides, influences the summer averaged SVU intensity and its interannual variability. With the goal of better modeling and
understanding the functioning and variability of ocean dynamics at small spatial and temporal scales and their influence
on the SVU summer intensity and interannual variability, we thus developed a high-resolution configuration of a
numerical ocean model, able to represent ocean dynamics and water masses at all scales from the Vietnamese coast to the
offshore region.

The model is presented in Section 2. Its capability to represent ocean dynamics and water masses is evaluated in Section
3. We then used this model to examine the variability of the SVU at different scales. Here we first focus on its interannual
variability over its 4 main areas of development, examining in particular the role of high frequency and small scale
dynamics. This analysis is presented in Section 4. Results are discussed in Section 5 and summarized in Section 6.

## 2. Methodology : numerical model and observation datasets

### 2.1. The regional ocean hydrodynamical model SYMPHONIE

The 3-D ocean circulation model SYMPHONIE (Marsaleix et al. 2008, 2019) is based on the Navier-Stokes primitive
equations solved on an Arakawa curvilinear C-grid under the hydrostatic and Boussinesq approximations. The model
makes use of an energy conserving finite difference method described by Marsaleix et al. (2008), a forward-backward
time stepping scheme, a Jacobian pressure gradient scheme (Marsaleix et al. 2009), the equation of state of Jackett et al.
(2006), and the K-epsilon turbulence scheme with the implementation described in Michaud et al. (2012) and Costa et al.
(2017). Horizontal advection and diffusion of tracers are computed using the QUICKEST scheme (Leonard 1979) and
vertical advection using a centered scheme. Horizontal advection and diffusion of momentum are each computed with a
fourth order centered biharmonic scheme as in Damien et al. (2017). The viscosity of momentum associated with this
biharmonic scheme is calculated according to a Smagorinsky-like formulation derived from Griffies and Hallberg (2000).
The lateral open boundary conditions for temperature, salinity, current and sea surface height, based on radiation
conditions combined with nudging conditions, are described in Marsaleix et al. (2006) and Toublanc et al. (2018). The
VQS (vanishing quasi-sigma) vertical coordinate described in Estournel et al. (2021) is used to avoid an excess of vertical
levels in very shallow areas while maintaining an accurate description of the bathymetry and reducing the truncation
errors associated with the sigma coordinate (Siddorn et al. 2013).

The configuration used in this study is based on a standard horizontal polar grid (Estournel et al. 2012) with a resolution
decreasing linearly seaward, from 1 km at the Vietnamese coast to 4.5 km offshore (Figure 1a), and 50 vertical levels.
This configuration with a refined resolution on the Viet Nam coast is hereafter referred to as the VNC configuration. The
simulation runs from January $1^{st}$, 2009 to December $31^{st}$, 2018. The atmospheric forcing is computed from the bulk
formulae of Large and Yeager (2004) using the 3-hourly output of the European Center for Medium-Range Weather
Forecasts (ECMWF) 1/8° atmospheric analysis, distributed on http://www.ecmwf.int. In particular wind stress $\vec{\tau}$ is
computed from ECMWF wind velocity based on the bulk formula of *Large and Yeager (2004)* : $\vec{\tau} = \rho_a \sqrt[2]{C_d} \overrightarrow{u_{10}}$ where
$\overrightarrow{u_{10}}$ is the wind velocity at 10 m height , $\rho_a$ is the air density computed from sea level pressure *SLP* and air temperature
at 2m $T_{2m}$, $\rho_a = \frac{SLP}{287,058\, T_{2m}}$ , and $C_d$ is the nonlinear drag coefficient computed from Large and Yeager (2004). The
horizontal wind stress curl WS is then computed as the vertical component of wind stress rotational: $WS = \frac{\partial \tau_y}{\partial x} - \frac{\partial \tau_x}{\partial y}$
where $\tau_x$ and $\tau_y$ are respectively the zonal and meridional components of $\vec{\tau}$. Initial ocean conditions and lateral ocean

boundary conditions are prescribed from the daily outputs of the global ocean 1/12° analysis PSY4QV3R1 distributed by the Copernicus Marine and Environment Monitoring Service (CMEMS) on http://marine.copernicus.eu. The implementation of tides follows Pairaud et al. (2008, 2010). It includes tidal amplitude and phase introduced at open lateral boundaries and the astronomical plus loading and self-attraction potentials, and considers the 9 main tidal harmonics, provided by the 2014 release of the FES global tidal model (Lyard et al. 2006). As the absence of tides in the 3D COPERNICUS simulation led to an underestimation of T and S mixing near the bottom, we add to the COPERNICUS fields the effect of the tidal mixing effect, using the parameterization described in Nguyen-Duy et al. (2021). The boundary conditions for rivers are described in Reffray et al. (2004) and Nguyen-Duy et al. (2021). 3 kinds of river runoff data were available over the VNC region: monthly climatology data for 17 rivers, yearly climatology data for 1 river, and daily real-time data for 18 rivers (see Figure 1a for locations of river mouths). The total freshwater discharge of the 36 rivers is equal on average over 2009-2018 to 21.4. $10^3$ $m^3s^{-1}$. To account for the interannual variability of river discharges for climatological data, we applied to those climatological data a yearly multiplying coefficient obtained from daily data time series.

## 2.2.    Observation data

### 2.2.1.    Satellite data

Available satellite observations are used to evaluate the capability of the model to reproduce sea surface characteristics over the study area:

●    OSTIA (Operational Sea Surface Temperature and Sea Ice Analysis) level 4 analysis daily SST produced using an optimal interpolation approach by the Group for High-resolution SST (GHRSST, UK MetOffice, 2005) on a global 0.05° grid over the period 2009-present, available at ftp://data.nodc.noaa.gov/pub/data.nodc/ghrsst/L4/GLOB/UKMO/OSTIA/.

●    Boutin et al. (2018) version 3 of 9-day-averaged level 3 sea surface salinity (SSS) derived from SMOS microwave imaging radiometer with aperture synthesis (MIRAS) measurements at 0.25° resolution over 2010-2017, available at ftp://ext-catds-cecos-locean:catds2010@ftp.ifremer.fr.

●    Daily altimetry sea level (SSH, sea surface height) data produced by AVISO (Archiving, Validation and Interpretation of Satellite Oceanographic data) at a 0.25° resolution over the period from 1993 up to now (Ablain et al. 2015, Ray and Zaron 2015), distributed by the CMEMS on http://marine.copernicus.eu.

●    JAXA daily SST (Level 3) dataset provided from Himawari Standard Data by JAXA over the period 2015 - present with a 2 km spatial resolution. We only consider JAXA data of quality levels ≥4.

For model-data comparisons, model outputs are interpolated on spatial and temporal grids of data

### 2.2.2.    In-situ data

To evaluate the capability of the model to reproduce temperature and salinity (TS) characteristics of water masses, we collected and processed temperature and salinity in-situ data available over the study area (see measurements characteristics in Table 1 and locations in Figure 1b). The modeled outputs were spatially and temporally co-localized with observations used for comparison.

●    ARGO floats : over the period 2009-2018 more than 3 600 TS profiles were collected in the VNC domain. The data are available on ftp://ftp.ifremer.fr/ifremer/argo/geo/pacific_ocean/

●    Sea-glider Vietnam – United States campaign : under the framework of a cooperative international research program (Rogowski et al. 2019), a Seaglider sg206 was deployed on 22 Jan 2017 and crossed the shelf break approximately 4

times before the steering mechanism malfunctioned on 21 Apr 2017. The mission ended on 16 May 2017. The glider
       collected 552 vertical profiles over the 114 days of deployment (hereafter called GLIDER data).

● ALIS research vessel (R/V) data : the VITEL cruise was organized in May – July 2014 on the French R/V ALIS.
       During ALIS trajectory in the SCS, surface temperature and salinity were measured every 6 seconds, from 10 May 2014
       to 28 July 2014 by the automatic thermosalinometer (TSG) Seabird SBE21 (hereafter called ALIS-TSG data).

● IO-18 data : Meteorological, current, temperature, and salinity parameters were measured by IO (Institute of
       Oceanography) in the framework of the Vietnamese State project in September 2018. Measurements were performed by
       a Sea-Bird 19plus CTD (Conductivity, Temperature, Depth) at 19 stations, including 16 instantaneous measurement
       stations and 3 stations where measurements were collected every 3 hours during 24 hours, producing 43 TS profiles.

## 2.3.    Definition of upwelling areas and indicators

Figure 3 shows the simulated and observed SST averaged over the period of SVU, June-September (JJAS), over 2009-
       2018. Low SST (< 29°C) simulated by SYMPHONIE and observed in OSTIA along and off the central Vietnam coast is
       the signature of the SVU, and reveals the same areas of upwelling as Da et al. (2019) and Ngo and Hsin (2021). The
       coldest water around 109°E - 11.5°N (<27°C at the coast) corresponds to the coastal center of the upwelling. The coastal
       area of cold water (~28-29°C) extends south and north of this cold center, from 8.8°N, off the Mekong mouth, and to
15.5°N, along the Vietnamese coast. Cold water is also observed offshore, from 109°E to 114°E and from 10°N to 14°N.
       We therefore define four areas of upwelling, shown in Figure 3 (see caption for exact coordinates). BoxSC corresponds
       to the well-known coastal upwelling (SCU) that develops under the effect of the eastward jet. BoxOF corresponds to the
       offshore upwelling, OFU. BoxNC shows the northern coastal area between 12°N and 15°N where the upwelling develops
       less frequently and has been much less studied (NCU). The fourth area of minimum SST is also visible both in observed
and in simulated summer SST off the Mekong Delta, in the wake of Con Dau Island between 8° and 10°N and 106.5°E
       and 109°E, named BoxMK hereafter. Figure 1c shows the details of the grid and bathymetry over those boxes.

       To examine the interannual variability of the upwelling over each area, we compute from our simulation a SST-based
       upwelling index for each box. For that, we followed exactly the same methodology as Da (2018) and Da et al. (2019),
who studied into details the implications this methodology. To estimate the strength of the upwelling over the area *boxN*,
       we define the daily upwelling index $UI_{d,boxN}$:

$$UI_{d,boxN}(t) = \frac{\iint_{(x,y) in\, boxN\, so\, that\, SST(x,y,t)<To} (T_{ref} - SST(x,y,t)).dx.dy}{A_{boxN}} \qquad (1)$$

       where $A_{boxN}$ is the size of *boxN*, *Tref* the reference temperature and *To* the threshold temperature. We also define the
       yearly upwelling index $UI_{y,boxN}$:

$$UI_{y,boxN} = \frac{\int_{JJAS} UI_{d,boxN}(t)dt}{ND_{JJAS}} \qquad (2)$$

       where $ND_{JJAS}$ = 122 days is the duration of the JJAS period. To examine the spatial distribution of the upwelling, we
       introduce a spatially dependent upwelling index $UI_y$:

$$UI_y(x,y) = \frac{\int_{t\, in\, JJAS\, so\, that\, SST(x,y,t)<To} (T_{ref} - SST(x,y,t))dt}{ND_{JJAS}} \qquad (3)$$

       To compute *Tref*, we define a common reference box BoxT_Ref as the area east of the upwelling region that is the less
impacted by upwelling (see Figure 3): *Tref* is computed as the average JJAS SST over the 2009-2018 period over BoxT_Ref.
       Using the same $T_{Ref}$ for all the four boxes allows to obtain a continuous spatial field of upwelling index (Eq. 1,2,3), and
       limits the potential influence of advection of water upwelled in upwelling areas into BoxT_Ref. We obtain $T_{Ref}$ = 29.20°C.

Using again the same method as Da (2018) and Da et al. (2019), the threshold temperature $To$ below which upwelling happens is defined from the analysis of the occurrence of cold surface water: it is defined as the optimal temperature threshold that allows to cover the largest number of upwelling occurrences but avoids to include cold water horizontally advected between upwelling areas. For that, we vary $To$ between 26.0°C and 28.0°C every 0.1°C, finally selecting $To = 27.6$°C.

3. Model evaluation

3.1.    Upwelling over BoxMK

To our knowledge, upwelling over BoxMK, referred to as MKU hereafter, has not been detected and studied so far. It was however already visible in AVHRR SST data over 1991-2004 (*cf* Figure 1 of Da et al. 2019) and in SODA reanalysis SST over 1987-2008 (*cf* Figure 5 of Fang et al. 2012). The summer surface cooling over BoxMK can actually be observed in OSTIA and COPERNICUS analysis data, but with much warmer values than in SYMPHONIE (Figure 3). This can be explained by the scarce satellite cover over the MKU area and the smoothing associated with the construction of reanalysis data. The percentage of days during which JAXA data are available is shown as an example in Figure 1d for summer 2018: data availability never exceeds ~85% over the SCS, is ~75-80% over BoxOF, and is lower than 60% over BoxMK. To determine if SST cooling really occurs over BoxMK, we thus examine in-situ and high-resolution satellite data that cover BoxMK during the period of upwelling.

First, ALIS R/V crossed BoxMK in June 2014. A surface cooling over BoxMK was indeed observed by ALIS-TSG on 25/06/2014, and simulated by SYMPHONIE, with values of minimum SST ~28.2°C (Figure 4a,b). On this day, OSTIA and COPERNICUS analysis show a significant cooling in the same area as SYMPHONIE (Figure 4c-e). However, minimum SST does not go below ~28°C in analysis data, whereas it reaches ~27.5°C in SYMPHONIE. Figure 5a shows the daily time series of minimum SST over BoxMK for SYMPHONIE, OSTIA and COPERNICUS during summer 2014. All times series show similar temporal variations, with a minimum at the end of June – beginning of July. However, minimum SST values over BoxMK are ~1.5°C lower in SYMPHONIE than in OSTIA and COPERNICUS during this period. Second, Figure 5b shows the daily time series of minimum SST over BoxMK for SYMPHONIE, OSTIA, COPERNICUS and JAXA during summer 2018. Again, all time series are following similar variations, but with much colder values (by ~1.5°C) in SYMPHONIE than in OSTIA and COPERNICUS. Moreover, values simulated in SYMPHONIE are very close to JAXA observations, with minimum peaks occurring at the same periods: mid-June (~26.6°C), mid-July (~25.6°C) and mid-August (~26.2°C). The spatial coverage of BoxMK by JAXA pixels of quality lever ≥ 4 exceeds 60% during more than half of the 2018 summer (Figure 5b), in particular during the periods of low SST (mid-June, Mid to end July, Mid to end August). It is therefore statistically meaningfull to use JAXA observations to assess the evolution of minimum SST during the summer period over BoxMK. Figure 6 shows the SST maps during those three upwelling peaks. A surface cooling over BoxMK is produced by JAXA and SYMPHONIE during those peak periods, with SST < 28°C. Again, this surface cooling is also observed by OSTIA and COPERNICUS analysis, but with warmer values, although OSTIA also produces a strong cooling during mid-July with minimum SST reaching ~27.5°C. For both summers 2014 and 2018, surface cooling is stronger in SYMHONIE than in analysis data, but occurs over similar areas (Figures 4,6).

This analysis confirms that summer upwelling really occurs over BoxMK. It is captured by ALIS TSG in-situ data and JAXA high-resolution satellite data, and simulated accordingly by SYMPHONIE. It is also captured by OSTIA and COPERNICUS, but is strongly attenuated. These results therefore highlight the added-value of a high-resolution (~1 km at the coast) resolution model to study the upwelling in the SVU area, related to its capability to simulate better than coarser resolution gridded satellite data the fine spatiotemporal scale structures of SST and surface dynamics.

3.2.    Surface circulation, temperature and salinity in the SCS and SVU regions

Figure 7 shows the time series of climatological monthly averages and of yearly averages of simulated and observed SST, SSS and SLA averaged over the VNC domain and over the SVU area shown in Figure 3 (104-116°E; 7-16°N). In Section 3 VNC and SVU regions are called simply VNC and SVU for the sake of conciseness. For both simulated and observed time series, the daily altimetry sea level anomaly (SLA) is obtained by removing at each point the averaged SSH over 2009-2018 : $SLA(x, y, t) = SSH(x, y, t) - \overline{SSH(x, y, t)}$. The normalized root-mean square error of a simulated time series compared to an observed time series is computed as $NRMSE = \frac{\sqrt{\frac{1}{N}\sum_{i=1}^{n}(S_i - O_i)^2}}{O_{max} - O_{min}}$ where $S_i$ and $O_i$ are respectively the simulated and observed values, and $O_{max}$ and $O_{min}$ the maximum and minimum observed values. Figure 2 shows the maps of winter and summer climatological averages over 2009-2018 of observed and simulated SST, SSS and SSH.

3.2.1.    Annual cycle and seasonal variability of surface circulation, temperature and salinity

SST seasonal cycles over SVU and VNC are similar (Figure 7a,g). The seasonal cycle of SST is well reproduced by the simulation. The bias compared to OSTIA data is equal to 0.08°C with a 0.02 NRMSE over VNC, and to -0.02°C with a 0.01 NRMSE over SVU. The correlation between the simulated and observed SST is highly statistically significant (1.0, p<0.01) for both VNC and SVU. Under the influence of Southeast Asia Monsoon, SST is maximum in May-June and minimum in January-February, consistently with Siew et al. (2013). SST spatial patterns are also realistically simulated, with highly significant spatial correlations between simulated and observed fields (Figure 2a-d, R=0.99 in winter and 0.85 in summer). In winter, the cold atmospheric fluxes and basin wide cyclonic circulation result in a strong meridional SST gradient. The summer SVU is clearly visible from observed and simulated SST minimum (< 28°C). Though SYMPHONIE is in very good overall agreement with OSTIA in terms of spatial variability and temperature range, it however slightly overestimates the observed surface cooling in very coastal areas near Hainan and southern Vietnam coasts. Piton et al. (2021) suggested that this surface cooling could be induced by tides. As explained in Section 3.1 when examining the surface cooling over BoxMK, and as also shown by Woo et al. (2020), this difference is also due to the SST smoothing in OSTIA and the lower satellite and in-situ data availability over the coastal regions (due in particular to the high cloud cover). Other factors could however explain an overestimation of surface cooling in the model: biases in atmospheric fluxes, which are difficult to estimate due to the scarcity of appropriate data; overestimation of vertical mixing due to the numerical model design (schemes of advection and diffusion, vertical coordinates …). The comparison with in-situ temperature and salinity profiles however shows the good performance of the model in the representation of water masses characteristics (see Section 3.3 below).

SSS seasonal cycles over SVU and VNC are very similar (Figure 7c,i). The simulated SSS follows well the observed SMOS SSS, with a highly statistically significant correlation (0.89 over VNC, 0.95 over SVU, p<0.01). Our simulation shows a fresh bias (-0.30 on average over VNC, from -0.22 in winter to -0.36 in summer, -0.21 over SVU), and an associated NRMSE of 0.76 over VNC and 0.40 over SVU. This salinity difference could be due to the coarse resolution

of SMOS (28 km), but also to an overestimation of atmospheric freshwater flux in the atmospheric forcing. Spatial patterns are well reproduced, with highly significant spatial correlations between observed and simulated SSS fields (0.84 in winter and 0.88 in summer, p<0.01, Figure 2e-h). Both simulation and data show low SSS over the Gulf of Thailand and the northwestern Gulf of Tonkin, associated with the strong river and atmospheric freshwater fluxes. In the northeastern SCS, they show high SSS round the year, due to the influence of water entering through the Luzon Strait from the Western Pacific Ocean (Qu et al. 2000; Zeng et al. 2016).

The simulated seasonal cycle of SLA is in very good agreement with AVISO observations (Figure 7e,k), with a 1.0 correlation coefficient (p < 0.01) and a 0.06 NRMSE over VNC, and respectively 0.86 (p<0.01) and 0.19 over SVU. SLA variations are much weaker over SVU compared to VNC. For VNC, SLA is maximum in November-January and minimum in June-July, in agreement with studies made from TOPEX/Poseidon altimeter data on the seasonal variability of SLA over the SCS between 1992 and 1997 by Shaw et al. (1999) and Ho et al. (2000). Winter and summer SSH spatial patterns and associated total geostrophic currents are realistically reproduced (Figure 2i-l), with highly significant spatial correlations between simulated and observed fields (R>0.94). The geostrophic winter circulation is basin-wide cyclonic. The summer circulation is anticyclonic, with a northward alongshore current, composed in particular of an anticyclonic (AC) gyre centered at ~11°N and a cyclonic (C) gyre centered at ~13°N. These structures form the well-known dipole and associated eastward jet and are highlighted by grey arrows in Figure 2k-l.

### 3.2.2. Interannual variability of surface circulation, temperature and salinity

VNC and SVU regions show very similar interannual chronologies of SST, SSS and SLA (Figure 7, right panel).

The interannual variability of SST is very well simulated, with a 0.95 (p<0.01) correlation coefficient and a 0.13 NRMSE compared to OSTIA dataset for VNC, and respectively 0.98 (p<0.01) and 0.06 for SVU (Figure 7b,h). SST shows strong variations over the period, with a 1°C difference between the coldest (2011, 27.5°C over VNC) and warmest (2010, 28.5°C) years. As shown by Yu et al. (2019) over the period 2003-2017, those interannual variations are strongly related to ENSO : the 2010 and 2015-2016 El Niño events induce SST maxima, while the 2011 La Niña event induces a SST minimum.

Simulated interannual SSS time series shows a statistically significant correlation (0.80 over VNC, 0.86 over SVU, p<0.01) and a NRSME of 0.48 for VNC and 0.39 for SVU compared to SMOS observations (Figure 7d,j). Both model and data show an increase between 2012 and 2016, also observed by Zeng et al. (2014, 2018). Through a budget analysis based on a SCS 4 km SYMPHONIE configuration over the SCS, Trinh (2020) attributed this salinification to the rainfall deficit during this period.

The interannual variability of SLA is very well simulated, with a 0.94 (p<0.01) correlation coefficient for both VNC and SVU and a NRMSE of respectively 0.13 and 0.12 compared to AVISO dataset (Figure 7f,l). Both observed and simulated datasets show minimum values in 2009, 2011 and 2015, and maximum in 2010, 2012 and 2013, with an amplitude of ~3-4 cm.

3.3.    Representation of TS characteristics and vertical distribution of water masses, at the daily to interannual scales

In section 4 below, we use surface variables to quantify the SVU strength and investigate the factors involved in its interannual variability. Surface dynamics are however influenced by the ocean vertical stratification and dynamics. We therefore also assess the capability of the model to realistically reproduce the tridimensional evolution of water masses in the computational area from the open-sea region to the coastal areas. For that, we use ARGO, GLIDER, IO-18 and ALIS-TSG datasets. We first examine TS diagrams built from ARGO (Figure 8a), GLIDER (Figure 8b) and IO-18 (Figure 8c) observations and colocalized SYMPHONIE outputs.

At the regional scale, ARGO data cover the whole open-sea region over 2009-2018. The TS characteristics of water masses defined by Uu and Brankard (1997) and Dippner et al. (2011) are well reproduced by the model (see Figure 8 for the acronyms of water masses), with similar contours of simulated and observed scatterplots. In particular the whole range of surface water masses is well simulated, with salinity varying between 32 and 34 and temperature between 20°C and 30°C : the fresh MKGTW, the more saline OWW and OSW and the colder ECSW. Salinity maximum (reaching ~34.8) of the MSW and salinity minimum (reaching ~34.4) of the PTW are also correctly reproduced by the model, as well as the DW mass.

GLIDER data were collected between January and May 2017, i.e. during the period of northeast monsoon and the transition period to the southwest monsoon. They cover the open-sea region and the coastal area (Figure 1b). During this 9th year of the simulation, the model is able to reproduce in detail the observed temperature and salinity profiles (Figure 8b), though with slightly higher biases of salinity (up to 0.15) and temperature (up to 0.4°C) than for ARGO dataset. In particular it reproduces realistically not only the deep water masses down to 1000 m depth, but also the 2 branches of surface water masses that were sampled by the glider (Figure SM1) : the fresher branch (SSS<34.5) observed at the beginning of the glider cruise south of 16°N, and the warmer branch (reaching 20°C) observed at the end of the cruise north of 17°N. SYMPHONIE is therefore able to capture the variability of water masses over the winter to summer transition period, from the coastal areas to the open-sea region, without producing a significant drift after more than 8 years of simulation.

IO-18 and ALIS data were collected during the period and over the region of upwelling, respectively in 2014 and 2018. Figure 9 shows the values of observed and colocalized simulated SST and SSS along the ALIS trajectory, and their bias. The high frequency variability of SST and SSS recorded by the TSG along the ALIS trajectory during summer 2014 is very well simulated, with a 0.89, resp. 0.93, correlation between observed and simulated time series. In particular both observed and simulated outputs clearly reveal the surface cooling (by more than 1 °C, with water colder than 28°C) and saltening (with salinity reaching 34) observed at the end of June - beginning of July 2014 in the SVU area. Comparison with those data therefore shows the ability of the model to reproduce the surface TS characteristics during the period and over the region of upwelling.

IO-18 data allow to explore the vertical dimension of the upwelling region. They covered both the shallow shelf area (8 stations, Figure 1) and the coastal to offshore upwelling area (10 stations) at the end of summer 2018 (12 to 25/09/2018). Figure 10 shows the simulated and observed vertical TS profiles during I0-18 cruises. The coastal shallow region, submitted to the influence of the Mekong freshwater, shows warm (> 27°C) and fresh (<34) water, with a halocline between 20 and 10 m, correctly simulated by the model. It corresponds to the nearly horizontal branch of the TS diagram (Figure 8c). IO-18 TS diagram and profiles moreover reveal two types of profiles in the deeper regions (i.e. reaching at

least 150 m depth), corresponding to the nearly vertical part of the TS diagram. The first type of profiles was sampled along the section at ~12.7°N (stations 2.1 to 2.5, located in BoxNC) and in the coastal part of the section along ~10.5°N (stations 3.2, 3.4, 3.5, located in BoxSC). It shows high salinities (> 34.5) and low temperatures (< 25°C) below a pycnocline shallower than ~30 m. It corresponds to locations where upwelling still occurs at this period. The second type of profiles is sampled in the offshore part of the section at 10.5°N (stations 3.6 to 3.8, located in boxOF). It has deeper haloclines and thermoclines, reaching 90 m, with SSS between 32.8 and 33.6 and warmer SST around 28°C : the upwelling already ceased in this region at this period. Both the TS diagram (Figure 8c) and the TS profiles (Figure 10) show that thee model is able to reproduce this diversity of TS profiles in the coastal and offshore upwelling regions in very good agreement with IO-18 data, without any significant bias, even after 9 years of simulation.

4. Interannual variability of the South Vietnam Upwelling

Results of Section 3 show that our high-resolution simulation reproduces realistically the surface circulation, temperature and salinity and the water masses characteristics over the water column, and their spatial (from shelf to open sea) and temporal (seasonal to interannual) variability. Based on this 10-year long simulation, we examine in detail the functioning and interannual variability of SVU over the four regions of interest : southern (BoxSC) and northern (BoxNC) coasts, offshore region (BoxOF) and Sunda shelf (BoxMK).

Figure 11 shows for each summer of the simulation the maps of JJAS averaged wind stress field and curl over the SVU region computed from ECMWF atmospheric variables. Figure 12 shows the maps of spatial upwelling index $UI_y$ and of JJAS averaged surface current and of its vorticity, and the iso-contours of JJAS positive wind stress curl. The surface current vorticity $\zeta$ is computed as the vertical component of horizontal surface current rotation: $\zeta = \frac{\partial v}{\partial x} - \frac{\partial u}{\partial y}$ where $u$ and $v$ are respectively the zonal and meridional components of the surface current. The surface current is taken as the current of the first layer of the model, whose depth varies from ~1.00m over most of the domain to ~0.7 m in very shallow and coastal areas. Figure 13 shows the time series of yearly upwelling index over the period 2009-2018 for the 4 boxes. Table 2 shows the mean and standard deviation of those time series, the coefficient of variation CV defined as the ratio between the standard deviation and mean. It also shows correlations between time series of significant factors, in particular yearly upwelling indexes, summer wind stress and wind stress curl, summer surface current vorticity.

4.1. Interannual variability of wind and offshore summer circulation

4.1.1. Relationship between wind stress and wind stress curl

The intensity of wind and wind stress curl shows a strong interannual variability (Figures 11,13) but their spatial patterns are quite similar from one year to another and related to the summer monsoon wind. A wind stress curl dipole develops offshore Vietnam, with an area of strong positive curl along and off the Vietnam coast in the north (covering BoxSC and a part of BoxOF and BoxNC), and an area of strong negative curl in the south (covering BoxMK).

Figure 13 shows the yearly time series of the average over each box of JJAS wind stress, $WS_{JJAS,boxN}$. Interannual time series of the summer wind stress over BoxSC and BokMK are almost equal to the summer wind stress over BoxOF and completely correlated with it (correlation higher than 0.96, p<0.01, Table 2). The intensity and interannual variability of summer wind intensity over BoxSC and BoxMK coastal areas is thus completely driven by the large-scale summer monsoon wind intensity over the region. The averages of JJAS wind stress curl over BoxSC (area of maximum positive wind stress curl) and BoxMK (area of maximum negative wind stress curl) are indicators of the intensity of the wind curl dipole that develops offshore Vietnam. There is a highly significant correlation (absolute value greater than 0.91, p<0.01, Table 2) between these indicators on one side and the wind stress on BoxMK, BoxSC and BoxOF on the other side. The intensity of wind stress over those three regions of upwelling and the intensity of the wind stress curl dipole are therefore strongly related. On the contrary, wind stress and wind stress curl over BoxNC is not significantly correlated with wind over any of the other boxes (Table 2).

4.1.2. Relationship between wind and offshore summer circulation

Positive surface summer current vorticity develops over the SVU region along the northern flank of the eastward jet (a more intense and narrow jet being characterized by a higher vorticity) and in the area of cyclonic activity north of the jet (Figure 12). We compute the time series of $\zeta_{+,OF}$, the spatial integral over BoxOF of the positive part of JJAS averaged relative vorticity (Figure 13). $\zeta_{+,OF}$, integrates both the cyclonic activity north of the jet and the positive vorticity on the northern flank of the eastward jet. It is thus an indicator of the intensity of the cyclonic part of the summer circulation (AC/C dipole + eastward jet) in the offshore region. The interannual variability of $\zeta_{+,OF}$ is strongly induced by the variability of intensity of wind stress and of the wind stress curl dipole. There is a highly significant correlation between the time series of $\zeta_{+,OF}$, and the time series of wind stress over BoxSC and BoxOF (0.89, p<0.01, Table 2) and of wind stress curl over BoxMK and BoxSC (-0.81 and -0.82, p<0.01, respectively).

4.2.    BoxSC

The inter-annual variability of SCU is significant, although it is the lowest of the 4 boxes (CV = 53%, Table 2). In 2010, $WS_{JJAS,SC}$ is minimum and significantly weaker than for the other years and almost no upwelling develops over BoxSC ($UI_{y,SC} \simeq 0.05$ °C, one order of magnitude smaller than the average). Conversely, when wind stress peaks in 2018 ($WS_{JJAS} \simeq 0.11$ N.m$^{-2}$, Figure 13) SCU reaches its maximum intensity ($UI_{y,SC} \simeq 1.50$ °C, almost twice stronger than the average). We obtain a highly significant correlation between $UI_{y,SC}$ and $WS_{JJAS,SC}$ (R=0.85, p<0.01, Table 2). $UI_{y,SC}$ is also significantly correlated with $\zeta_{+,OF}$ (0.60, p=0.07, Table 2), which confirms that SCU is stronger/weaker for years of intense/weak summer circulation off the Vietnam coast. Our results therefore confirm conclusions from previous lower resolution studies (see Introduction): the interannual variability of SCU is driven at the first order by the intensity of the summer averaged monsoon wind over the region, and by the intensity of the regional summer circulation.

Our study moreover reveals that other factors than the summer average of wind intensity contribute to the interannual variability of SCU, namely intraseasonal wind chronology and mesoscale circulation over BoxSC.

First, the daily to intraseasonal variability of wind forcing contributes to the summer average of SCU intensity. For example, even though 2009 and 2012 show the same average summer wind over BoxSC ($WS_{JJAS,SC} \simeq 0.09$ N.m$^{-2}$), SCU is 25% weaker in 2009 ($UI_{y,SC} \simeq 0.75$ °C) than in 2012 ($UI_{y,SC} \simeq 0.95$ °C). Figure 14 shows the time series of daily upwelling index $UI_{d,SC}$ and of daily average wind stress over BoxSC for 2009 and 2012 (and 2011). For both years, the development of SCU is related to the daily chronology of wind, with upwelling peaks occurring during wind peaks. The wind over BoxSC is stronger in July and September for 2009. It is stronger in June and August for 2012. In 2012, the SCU consequently develops strongly in June, and persists throughout the summer, especially in August. Conversely in 2009 SCU mainly develops in July, and does not persist in August due to the weaker wind during this month. The daily to intraseasonal variability of wind forcing during the summer therefore contributes to the summer average of SCU intensity, hence to the SCU interannual variability.

The second factor that influences the summer average SCU intensity is the mesoscale circulation over the coastal area. For example, 2011 shows the second strongest SCU ($UI_{y,SC} \simeq 1.35$ °C), ~90% stronger than for 2009 and ~40% than for 2012 (Figure 13). SCU develops strongly in June 2011 and persists throughout the summer, reaching much higher values than in 2009 and 2012 (Figure 14). However, $WS_{JJAS,SC}$ in 2011 ($\simeq 0.09$ N.m$^{-2}$) is ~10% lower than in 2009 and 2012. Moreover, wind stress in June 2011 is slightly weaker than in June 2009 and 2012 in average over the month (Figure 14).

Figure 14 show that it also does not differ in terms of maximum intensity of high frequency peaks, though the chronology of those peaks differs. This shows that the higher upwelling intensity in 2011 does not result from a difference in summer wind, neither in average nor at the daily to intraseasonal scales. Instead, it results from the ocean circulation. 2009 and 2012 show the AC/C dipole and associated eastward jet at ~12°N described in the literature (Figure 12e,h). Instead, during summer 2011, two AC/C structures develop along the Vietnamese coast, as can be seen in Figure 12g where they are highlighted by black arrows. These structures are associated with two eastward jets departing respectively from 10°N and 13.5°N. This results in an overall offward current along the coast, hence in a wider coastal upwelling. The coastal circulation and mesoscale circulation are therefore a second factor that contributes to the summer average of SCU intensity and to its interannual variability.

### 4.3.    BoxOF

OFU shows the strongest interannual variability (CV≃126%, Table 2), explained partly by the very strong OFU in 2018 compared to other years induced by the strong wind in summer 2018 (Figure 13). The interannual chronology of OFU is very similar to that of SCU, with a minimum in 2010 and a maximum in 2018, and $UI_{y,SC}$ and $UI_{y,OF}$ are highly significantly correlated (0.73, p=0.02, Table 2).

Our results first confirm that summer wind strength is a key factor involved in OFU interannual variability : $UI_{y,OF}$ and $WS_{JJAS,OF}$ are highly statistically significantly correlated (R = 0.77, p=0.01, Table 2). They also confirm the link between the intensity of OFU and of summer offshore circulation and related vorticity, with a significant correlation between time series of $UI_{y,OF}$ and $\zeta_{+,OF}$ (0.69, p=0.03, Table 2). Figure 12 indeed reveals that OFU indeed occurs mainly in areas of positive surface current vorticity : along the northern flank of the eastward jet and north of the jet as the result of mesoscale cyclonic structures. For some cases, as suggested by Da et al. (2019) and Ngo and Hsin (2021), OFU can even be inhibited. In 2010 upwelling does practically not develop over BoxOF and BoxSC (Figures 12,13). The usual eastward offshore jet and dipole structure do not exist during summer 2010 (Figure 12). $\zeta_{+,OF}$ is indeed much smaller than for the nine other years (Figure 13, equal to $0.5x10^{-6}$ s$^{-1}$, vs. $2.0x10^{-6}$ s$^{-1}$ in average, Table 2). Moreover, an alongshore northward current flows between 9°N and 16°N (Figure 12). Those circulation conditions prevent the formation of an offshore current and of the associated upwelling. For this year, extreme weak wind conditions as well as circulation patterns thus both prevent the development of upwelling.

Our study moreover reveals again that factors other than summer wind intensity and summer circulation over the offshore area contribute to the interannual variability of OFU. For example, in 2018, the intensity of both wind and circulation intensity is only ~10% higher than in 2009 and 2012 (Figure 13): in 2018, $WS_{JJAS,OF}$ and $\zeta_{+,OF}$ are equal to 0.11 N.m$^{-2}$ and $3.4x10^{-6}$ s$^{-1}$, vs. 0.10 N.m$^{-2}$ and $3.1x10^{-6}$ s$^{-1}$ in 2009 and 2012. However, $UI_{y,OF}$ in 2018 is respectively 2.6 and 4.6 times larger than in 2009 and 2012 (0.32°C vs. 0.12°C and 0.07°C), and four times higher than the average (0.07°C, Table 2). Comparatively, for BoxSC, $UI_{y,SC}$ in 2018 it is only twice larger than the average value. Last, the correlation between $UI_{y,boxN}$ and $WS_{JJAS,boxN}$ is lower for BoxOF (0.77, Table 2) than for BoxSC (0.85). This suggests that the influence of factors others than summer averaged wind and circulation on the interannual variability of OFU could be more important than for SCU.

The intraseasonal variability of wind is an important factor that contributes to the summer average of OFU intensity, hence to its interannual variability. Figure 15 shows the time series of daily upwelling index $UI_{d,OF}$ and daily average wind stress over BoxOF for 2009, 2012 and 2018. Wind stress peaks mostly occur in July and September in 2009, and in

June and September in 2012. In contrast, in 2018, wind stress peaks occur in July and August, inducing a very strong upwelling during those months. The monthly averaged wind is consequently significantly stronger in July and August 2018 than in July and August of the other years, and than in June and September 2018 (Figure 15). This suggests that the summer intensity of OFU is higher when wind peaks occur during the core of the summer season (July-August) than at the beginning (June) or end (September). The correlation of time series of $UI_{y,OF}$ with time series of July-August averaged wind stress over BoxOF is indeed higher (0.84, p<0.01, Table 2) than the correlation with JJAS wind stress (0.77, p<0.01), confirming this hypothesis. Strong wind stress during July-August thus favor the development of OFU, more than during June and September. This is presumably related to a favorable combination of wind forcing and offshore circulation during this period. Conversely, for BoxSC, the correlation between $UI_{y,SC}$ and $WS_{JA,SC}$ is weaker than for $WSJ_{JAS,SC}$ (0.70, p=0.03 for $WS_{JA,SC}$ vs. 0.85, p<0.01, for $WS_{JJAS,SC}$, Table 2). The development of SCU indeed occurs during the whole JJAS period as soon as the wind conditions are favorable, and is thus mainly induced by wind forcing which prevails on and drives circulation patterns (see Section 4.2).

## 4.4. BoxNC

Upwelling also develops, though to a weaker extent, along the northern part of the central Vietnam coast (BoxNC). $UI_{y,NC}$ shows a relatively strong interannual variability (71%, Table 2). We obtain no statistically significant correlation between $UI_{y,NC}$ time series and time series of the average wind stress, whatever the period and area of averaging (Table 2). Moreover we obtain no statistically significant correlation between $UI_{y,NC}$ time series and the upwelling index over the three other boxes (Table 2). This suggests that contrary to SCU and OFU, the interannual variability of NCU is not driven by the summer wind or the summer offshore circulation. Examining into details the 10 summers of the simulation in Figure 12, we show instead that the development of NCU is inhibited or favored depending on the circulation that prevails over BoxNC.

Ngo and Hsin (2021) examined the role of each component of the summer wind in the development of NCU. Following their analysis, Figure 16 shows the maps of correlations between yearly time series of $UI_y$ (for SCU, NCU and OFU) and JJAS wind stress intensity over BoxOF on one side, and the (spatially dependent) times series of JJAS wind stress components and intensity on the other side. First, the intensity of JJAS wind stress over boxOF is highly significantly positively correlated (p<0.01) with the components and the intensity of wind stress over most of the domain (Figure 16e). But it is negatively correlated with the zonal component and the intensity of wind stress in the northeastern part of the domain (including BoxNC). The intensification of the summer monsoon wind over the region is thus associated with an intensification of the wind over most of the area, but with a weakening of its intensity and meridional component, i.e. a less northward and more eastward direction, in the northwestern part ; and vice versa. Second, correlations maps for OFU and SCU are very similar to the correlations maps for JJAS wind stress intensity over boxOF, with larger areas of highly significant correlations for SCU than for OFU (Figure 16c,d,e). This confirms the link shown in section 4.2 and 4.3, stronger for SCU than for OFU, between the intensity of upwelling over those regions and the intensity of monsoon wind. Third, the correlations maps for NCU are of opposite signs to the other maps (Figure 16a). When the summer monsoon wind intensifies (favorable to SCU and OFU), the wind stress and its meridional component weaken in the northeastern region covering BoxNC. The wind is therefore more eastward, i.e. less parallel to the coast and more offshore, i.e. not favorable to NCU. It is the opposite when the summer monsoon wind weakens. Last, correlations are much more statistically significant (p<0.01) for SCU and, to a lesser extent, OFU, than for NCU (p<0.05 or 0.01 only in very small coastal areas over BoxNC). These results therefore suggest that wind also participates to the interannual variability of NCU, with favorable conditions opposite of those favorable to SCU, OFU and MKU, as already shown by Ngo and Hsin

(2021). However, they further reveal that this influence of wind is not as important as for the other areas, and suggest that other factors may be playing roles of equal or stronger importance, in particular OIV and mesoscale structures.


Examining the spatial patterns of circulation indeed shows that they strongly impact the NCU development. The NCU is inhibited when alongshore currents, either southward or northward, prevail over BoxMK. Southward alongshore currents prevail during summers of strong wind over the SVU region, when the AC/C dipole and eastward jet, hence positive vorticity $\zeta_{+,OF}$, are highly marked, inducing strong OFU and SCU (see summers 2009, 2012, 2018, Figures 12,13). These

southward currents are associated with the western part of the northern cyclonic gyre and a divergent circulation, hence with a coastward component and a coastal downwelling which inhibits the NCU. Northward alongshore currents prevail during years of weak or average wind over the region (see summers 2010, 2013, 2017). During those years, offshore circulation (AC/C dipole and eastward jet) is average (2017), weak (2013) or even absent (2010), resulting in weak average Ekman transport and pumping, hence in weak SCU and OFU. The weakness of the offshore circulation allows

the development of an alongshore northward current all along the Vietnamese coast, which also inhibits the NCU. Southward or northward longshore currents over BoxNC therefore result from two opposite situations in terms of wind and offshore circulation, but induces both NCU inhibition.

The NCU is enhanced when offshore oriented circulation prevails over BoxMK. Offshore oriented circulation can result

first from the development of a secondary dipole north of the usual dipole structure: see for summers 2011, 2014 and 2015 the alternation of negative and positive vorticity between 12°N and 16°N along the coast (Figure 12). This secondary dipole is associated with a second coastal area of convergence over BoxNC, hence a secondary eastward jet that induces the strong NCU. This situation is not related to the intensity of summer wind or offshore circulation (Figures 12, 13). It indeed occurs both for summers of wind slightly stronger (2011 and 2014) or weaker than average (2015), and of strong

(2014) or weak (2011, 2015) offshore circulation. Offshore oriented circulation also develops when a weaker but wider than average eastward jet prevails over a large part of the coastal region, including BoxNC (see summer 2016). This results in the offshore advection of cold water all along the coast hence in the development of a stronger than average NCU. NCU is therefore favored by the development of offshore oriented currents along the coast, that result from a favorable spatial organization of submesoscale to mesoscale dynamics.

4.5. BoxMK

MKU shows the weakest mean value of the 4 boxes (<0.07°C, Table 2), and a relatively strong interannual variability (CV=85%). The interannual chronology of MKU is very similar to that of OFU and SCU, with a minimum in 2010 and a maximum in 2018 (Figure 13). $UI_{y,MK}$ is highly significantly correlated with $UI_{y,SC}$ (0.83, p<0.01, Table 2) and $UI_{y,OF}$ (0.92, p<0.01). MKU interannual variability is therefore likely driven by the same factors as OFU and, to a slightly lesser

extent, SCU. As for SCU and OFU, the summer wind strength and summer offshore circulation are indeed key factors involved in MKU interannual variability. $UI_{y,MK}$ shows a highly statistically significant correlation with summer wind over the region ($WS_{JJAS,SC}$ : 0.83, p<0.01, Table 2), and with the offshore circulation ($\zeta_{+,OF}$ : 0.74, p=0.01). This correlation with $\zeta_{+,OF}$ is even higher than for SCU (0.60) and OFU (0.69), suggesting an even stronger influence of summer offshore circulation (eastward jet and AC/C dipole) in MKU interannual variability.


MKU develops in the area of positive vorticity along the western flank of the northeastward current that flows off the Mekong delta (Figure 12). Its intensity is stronger when this current, and the associated positive vorticity, is stronger. This current constitutes the western part of the southern anticyclonic gyre of the AC/C dipole (Figure 12). It forms the

eastward jet when it encounters the northern cyclonic gyre. The intensity of this current therefore varies interannually following the intensity of the summer offshore circulation induced by summer monsoon wind. This explains the significant correlation between MKU and the regional offshore circulation and wind.

In contrast with upwelling that develops over the 3 other areas, the position of MKU is very stable from one year to another (Figure 12). MKU develops every year in the same area of positive current vorticity, though with very different intensities (absent in 2010 and 2016, to very strong in 2018). This clearly distinguishes it from the upwelling that develops in the other zones, in particular NCU and OFU, whose position is highly variable. The spatial structure of surface currents and associated vorticity in the MKU region is also very stable from one year to another (Figure 12), which explains this stable position of MKU. This agrees with conclusions from Fang et al. (2012) who showed that this segment of the summer circulation is stable at the interannual scale, whereas the northern and eastern parts are much more unstable. This stable circulation spatial pattern over BoxMK could be explained by the fact that it may be constrained by factors that do not vary interannually, such as topography and tide. BoxMK is indeed located on the shallow continental shelf, with depths varying from ~15 m at the coast to more than 200 m over the continental slope (Figure 1c).

Those results show that MKU interannual variability is driven first by the summer monsoon wind and by the northeastward current, which is completely related to the intensity of the wind-induced summer offshore circulation. MKU develops in area of positive vorticity associated with the northeastward coastal current in BoxMK. The spatial structure of this current, hence of MKU, is interannually stable, presumably because it may be strongly constrained by bathymetry related mechanisms.

## 5. Discussion

### 5.1. Sensitivity to the choice of varying vs. constant reference temperature $T_{Ref}$

To compute the daily to yearly upwelling intensity, a constant reference temperature $T_{Ref}$ was used (see Section 2.3), as done previously by Da et al. (2019). Our choice of a constant $T_{Ref}$ was based on the fact that even if the reference box was chosen as far as possible from the upwelling area and over the same latitudinal region, this choice was constrained by the computational domain (location of eastern boundary, Figure 3). On average, BoxT$_{ref}$ is outside of the upwelling area, however it can be influenced by the eastward advection of cold surface water upwelled in the other boxes, especially for years of strong upwelling. This advection is for example visible during summer 2018 of very strong OFU, and not during summer 2010 of weak OFU (Figure 17a). It results in a colder JJAS average SST over BoxT$_{ref}$ in 2018, and more generally in years of strong OFU (2009, 2011, 2012 and 2018, Figure 1b), than in 2010. Using an interannually varying SST could thus result in weaker yearly upwelling indexes for years of strong OFU. Using a climatological constant $T_{ref}$ contributes to limit this effect. Conversely, other studies used an interannually varying $T_{Ref}$ to take into account the fact that SST over the SCS varies on an interannual basis (e.g. Ngo and Hsin 2021) : the summer SST over the SCS is indeed also affected by other factors than the SVU, in particular by large scale factors like ENSO, with warmer sea surface following El Niño events (Qu et al. 2004, Wang et al. 2006). Taking into the effect of those large scale factors justifies the use of a yearly varying $T_{ref}$ as done by Ngo and Hsin (2021). Previous studies however also showed that ENSO influences the SVU, with weaker SVU following El Niño events (Da et al. 2019), hence weaker advection of cold water. Choosing a constant or a yearly-varying $T_{ref}$ therefore involves tradeoffs between those influence factors. Note that the choice of a seasonally average vs. a daily varying $T_{ref}$ involves similar tradeoffs (taking into account or not the effect of lateral advection vs.

seasonal variability). We evaluated the influence of a varying $T_{Ref}$ on our results, computing upwelling indexes with an interannually varying $T_{Ref}$ instead of a constant $T_{Ref}$ (Figure 17b). The yearly time series of upwelling indexes $UI_{y,boxN}$ very slightly differs, in particular for stronger values as expected (see year 2018), but changes are not significant. Recomputing all the variables from Table 2 (table not shown for the sake of conciseness) shows that the interannual variability only slightly reduces and that correlation coefficients consequently also slightly decrease, but by less than ~0.10). In particular, correlations that were statistically significant (or not), remain statistically significant (or not). Our results and conclusions are therefore robust to the choice of an interannually varying vs. constant $T_{Ref}$.

Last, we investigated the effect of using a common $T_{Ref}$ for the four boxes instead of using a specific $T_{Ref}$ for each box. For that, we used for each area a reference temperature equal to the average SST over an area contiguous to each box (figures not shown). Results show that using a common vs. specific Tref can influence the values of upwelling index, but only by up to 20 % for BoxNC. Moreover, this does not have any significant influence on the indicators and correlations shown in Table 2 : results changed by less than 1% when testing this choice. Our results are therefore also robust to this choice.

### 5.2.     Role of intraseasonal variability of atmospheric forcing

We confirmed in Section 4 the leading role of the seasonal average of summer monsoon wind in the interannual variability of the yearly intensity of SCU, OFU and MKU. We moreover showed in Sections 4.2 and 4.3 that the intraseasonal chronology of wind stress influences the summer average of SCU and OFU intensity, and their interannual variability : for a similar average summer wind, OFU is stronger if wind peaks occur during the July-August period. However other factors may be involved. Indeed, the monthly wind stress over BoxOF is stronger in June and August in 2012 than in 2009, and similar in July, and $\zeta_{+,OF}$ is similar. However, OFU is 50% weaker in 2012 than in 2009 (Figures 13,15). A first factor explaining this difference is the daily wind chronology inside the July-August period : 2009 shows two distinct periods of wind maximum in mid-July and mid-August, whereas 2012 shows two close peaks during the 2nd half of July (Figure 15). Those results suggest that the wind chronology not only at the intraseasonal monthly scale but even at the daily scale inside the July-August period affects the upwelling development. In particular wind peaks need to be sufficiently spaced in time. This result is supported by previous studies (Xie et al. 2007, Liu et al. 2012), who highlighted the importance of intraseasonal wind peaks on SVU development.

This conclusion was obtained from the study of four summers (2009, 2011, 2012 and 2018). To quantitatively highlight this role of wind peaks, we performed an additional simulation from June to September 2018 (the summer that shows the strongest wind stress and upwelling over BoxSC, BoxOF and BoxMK, Figure 13) : we used during the whole summer a temporally constant wind stress, computed for each point of the model as the JJAS 2018 average of the daily wind stress. Initial conditions for this simulation were taken as the conditions of June 1st, 2018 of the 2009-2018 simulation. In the sensitivity simulation, the surface cooling is much weaker than during summer 2018 of the 2009-2018 simulation, and no upwelling develops on any of the boxes during the whole summer. This result quantitatively highlights the fundamental role of wind intraseasonal variability in the development of upwelling and in its summer average intensity: a constant wind, even of strong average value, is not sufficient for the development of SCU, OFU and MKU, and peaks of strong winds are essential. This simulation, and additional sensitivity tests, will be examined into more details in further studies.

Additional simulations would now be required to more quantitatively assess the role of intraseasonal variability vs. seasonal and monthly average of wind stress on the yearly upwelling intensity. In particular, ensembles of simulations with wind of same seasonal average but different daily chronology, and conversely, would help to estimate the variability of summer averaged upwelling induced by the variability of the intraseasonal chronology vs. by the interannual variability of summer average wind.

### 5.3. Role of forced vs. chaotic variability

The interannual variability of the offshore circulation is partly driven by the intensity of regional wind stress and by the intensity of wind stress curl dipole that develops along the Vietnamese coast (see Section 4.1). However, spatial patterns of wind stress curl are quite similar from one year to another (Figure 11), but the current vorticity field over BoxOF shows a strong interannual variability (Figure 12). First, the eastward jet and associated positive vorticity develops almost every year (except in 2010), but its position and intensity vary. Second, the eddy related positive vorticity field northern of the jet and its spatial patterns vary even more strongly, and independently from the wind curl over the area. For example, wind is similar for 2014 and 2016 ($WS_{JJAS,OF} \simeq 0.08$ N.m$^{-2}$, Figure 13), with very similar spatial patterns of wind stress and wind stress curl (Figure 11). However, the spatial organization of mesoscale activity over BoxOF strongly differs (Figure 12). In 2014, an intense cyclonic eddy forms north of the intense and marked eastward jet, favoring the OFU. In 2016, the jet is much wider and less intense and no cyclonic eddy, and related OFU, develops. Consequently $\zeta_{+,OF}$ and OFU are about twice stronger in 2014 (respectively 0.06°C and ~2.6x10$^{-6}$ s$^{-1}$) than in 2016 (0.03°C and ~1.5x10$^{-6}$ s$^{-1}$). The correlation between time series of $\zeta_{+,OF}$ and of wind stress curl over BoxOF is moreover not statistically significant (Table 2). These results show that the interannual variability of the offshore circulation, in particular the eastward jet, is partly driven by the wind intensity and dipole ; but that part of this variability, in particular the variability of the eddy field northern of the jet, is not driven first by the wind stress curl over the offshore region itself, but related to the influence of other factors. Ocean mesoscale eddies largely contributes to OIV (ocean intrinsic variability), which is one of those factors. The choice of an integrated indicator is always associated with limitations, and other indicators can be used to represent the intensity of circulation (e.g. the intensity and position of eastward jet, Ngo and Hsin, 2021). In particular, $\zeta_{+,OF}$ does not account for the spatial characteristics of the positive vorticity. For example, years 2011, 2013 and 2015 show similar values of $\zeta_{+,OF}$ and JJAS wind stress intensity (Figure 13). However, the spatial distribution of this positive vorticity and the values of UIy over BoxSC and BoxOF differ (Figures 12, 13). 2011 shows a double dipole structure favoring the upwelling. 2013 shows an eastward jet located in the south and weak activity northern of the jet, not particularly favorable. 2015 shows an eastward jet located in the north with a strong anticyclone located over BoxOF, preventing the upwelling.

Using an indicator integrated over the whole SVU area, Da et al. (2019) indeed revealed the impact of OIV on interannual variability of the offshore upwelling. Our results highlighted the contribution of high frequency and submesoscale to mesoscale ocean dynamics and coastal circulation of strong chaotic nature to the interannual variability of SCU, OFU and NCU. Further sensitivity simulations are now required, including ensemblist approaches that allow to distinguish and quantify the effect of the chaotic vs. forced component of ocean dynamics at different scales, as done by Waldman et al. (2018). They will allow to better quantify the contribution of OIV in the daily to interannual variability of the SVU over its different areas of development. In particular they will help to understand which part of the variability of the current and its vorticity, and of the upwelling, is wind-induced and which part is chaotic, related in particular to the formation and propagation of submesoscale to mesoscale dynamics. They will allow to go beyond the limitations of an integrated indicator, examining into details the impact of spatial patterns of mesoscale circulation on the upwelling.

We used the current simulated in the 1st layer of our model (i.e. at ~1m) to examine the role of (sub)mesoscale structures and related OIV. To evaluate the capacity of the surface currents and eddies to represent the ocean dynamics, we explored their vertical extension (see Fig. SM2 as an example) : (sub)mesoscale eddies and currents simulated at the surface extend vertically until ~500m depth, and are therefore representative of the (sub)mesoscale surface but also subsurface and intermediate layer circulation of the SCS.

### 5.4. Comparison of simulated and observed interannual variability of upwelling intensity

Ngo and Hsin (2021) produced the most recent and longest time series of observed intensity of upwelling over three of the areas studied here (SCU, NCU and OFU), using a SST-based upwelling index applied on a 0.25° dataset of satellite observed SST over the period 1982-2019 (*cf* Figure 4 of Ngo and Hsin 2021).

For SCU and OFU, the interannual variability of upwelling intensity over the period 2009-2018 is similar at the first order in the time series simulated by our model (Figure 13) and observed by Ngo and Hsin (2021). In particular their intensity is highest in 2018 and lowest in 2010. For SCU, 2011, 2012 and 2014 are more intense than the average in both simulated and observed time series, and 2017 is much less intense than the average. For OFU, 2011 and 2012 are more intense than the average and 2013, 2015, 2016 and 2017 are less intense. However, some years are significantly different in simulated and observed time series. 2015 SCU is stronger than average in observations vs. smaller in our simulation. 2009 OFU is lower than average in observations, vs. the 2nd highest in our simulation. Differences are stronger for NCU, with only 4 years in agreement: 2009 and 2012 show particularly weak (or non-existent) NCU, 2015 shows a strong NCU and 2011 shows an average NCU in both observed and simulated time series. As showed in Section 3.1, the lower resolution of satellite observations and the strong cloud cover over the region in summer (Gentemann et al. 2004) partly explains these differences, especially in the small coastal boxes BoxNC and BoxSC.

Our high-resolution study allowed to deepen to examine in details NCU functioning. Ngo and Hsin (2021) concluded that conditions favorable to SCU and OFU were unfavorable to NCU, and vice versa. We confirmed this relationship in section 4.4, but showed that the correlation between wind and upwelling intensity is much weaker for NCU than for the other boxes, and that the role of circulation is stronger. We indeed identified one situation in agreement with the conclusions above : strong summer monsoon winds and resulting offshore circulation favorable to OFU and SCU result in strong alongshore currents along the northern part of the coast that prevents Ekman transport, hence NCU development. However, we also showed that similar wind conditions and offshore circulation could be associated with different NCU intensities. In particular, different submesoscale and mesoscale circulation develops along the northern coast and influences NCU development. Similar wind conditions (from weak to strong) can indeed result 1) in the development of multiple dipoles and eastward jets along the coast or of a wide eastward jet, favorable to NCU; or 2) in the development of a strong northward alongshore current, unfavorable to the NCU. Similar summer monsoon wind conditions can therefore result in very contrasted NCU intensity, and vice versa. In other words, the first driver of NCU is not only regional summer monsoon wind and resulting offshore circulation, but also the spatial organization of submesoscale to mesoscale circulation that prevails over BoxNC. This circulation is by nature partly chaotic and contributes to OIV.

Submesoscale to mesoscale ocean dynamics were thus shown to contribute to the interannual variability of SCU, OFU and NCU. The differences between our study and the study of Ngo and Hsin (2021) in terms of NCU, OFU and SCU intensities could therefore also be due to the chaotic part of those ocean dynamics. Our results suggest that this OIV could be stronger for BoxNC, since its interannual variability is more influenced by ocean submesoscale to mesoscale dynamics, than by wind. This would explain the larger differences between simulated and observed NCU time series. As written

above, ensemblist approaches will allow to understand and quantify the respective contributions of wind and (sub)mesoscale circulation and related OIV on NCU intensity. Last, the different number of upwelling events in our study and in Ngo and Hsin (2021) also contributes to the difference in results between the two studies.

### 5.5.    Surface cooling over BoxOF : role of offshore upwelling vs. lateral advection of cold water

Most SVU studies assumed that summer surface cooling over BoxOF, which is very similar in OSTIA and SYMPHONIE (Figure 3), results from of an upwelling developing over BoxOF. However, this surface cooling could also result from the advection by the eastward jet of cold water upwelled at the coast. Three-dimensional simulations can help to answer this question, which can not be answered from the study of surface variables (temperature, currents) only. Figure 18 shows the maps the average of simulated vertical velocity at 20 m, daily upwelling index and surface currents over the two periods of upwelling development in summer 2018 (summer of strongest OFU, Figure 13): the first two weeks of July 2018 and the first week of August 2018 (Figure 15). Those figures qualitatively show that BoxOF surface cooling indeed partly results from the advection of cold water from BoxSC, in particular in July. However, strong upward (positive) vertical velocities are simulated, not only along the coast and over BoxMK, but also over BoxOF, in particular in August. This confirms that a significant part of surface cooling over BoxOF results from local upwelling. Further dedicated studies, including box analysis following the method used for dense water formation by Herrmann et al. (2008), are now required. They will help to quantitatively assess the respective contributions of lateral advection, surface forcing, vertical advection and internal mixing to the formation of cold surface water over BoxOF.

The in-situ horizontal, but also vertical coverage of the region and period of SVU is still very scarce. High-resolution campaign dedicated to the tridimensional observation of SVU would allow to evaluate more precisely the capability of the model to reproduce the tridimensional dynamics and water masses, but also to better understand the SVU dynamics, including the role over vertical vs. lateral transport of cold water.

### 6.   Conclusion

The purpose of this study was to simulate the large range of scales of ocean processes and atmospheric forcing, including submesoscale dynamics and high frequency variability, in order to understand better their role in functioning and variability of the SVU. For that, we implemented a South China Sea configuration of the 3D ocean model SYMPHONIE at high-resolution, varying from ~1km along the Vietnamese coast to ~4 km in the open SCS, and ran a simulation over the period 2009-2018. Comparisons with available satellite data and four sets of in-situ observations quantitatively showed the capability of the model to reproduce the ocean circulation and water masses at different scales in the area: from the daily to interannual and climatological scales and from the coastal to offshore regions.

We used this simulation to examine in detail the interannual variability of SVU over three known areas of development: the coastal upwelling, that develops along the southern (SCU) and northern (NCU) parts of the Vietnamese coast and the offshore upwelling (OFU). Moreover, our simulation, together with a careful examination of available satellite and in-situ data, showed for the first time that upwelling also develops on the Sunda shelf offshore the Mekong delta (MKU). Limitations associated with gridded satellite data indeed strongly reduce the spatial observability of small coastal areas, preventing them in particular to capture the MKU. The high-resolution of our simulation allowed to better capture the SST and current variations over those small coastal areas compared to coarser resolution numerical outputs and smoothed

satellite data. It is therefore a highly valuable tool to complement those observations and examine in details the functioning and variability of upwelling over its different areas of development.

For SCU and OFU, our results confirmed previous conclusions at the first order: the interannual variability of SCU and OFU intensity is first driven by the intensity of summer wind stress and wind stress curl, which induce Ekman transport and pumping. The intensity of summer circulation in the offshore area (i.e. the AC/C dipole and associated eastward jet), also partly induced by the summer wind, also influences this interannual variability. Moreover, the high-resolution of our simulation allowed to reveal and examine the role of other factors. First, the interannual variability of SCU and OFU

intensity is not only influenced by the summer and regional average of the offshore circulation, but also by the spatial distribution of (sub)mesoscale and coastal circulation. SCU intensity partly depends on the number and location of AC/C dipoles along the coast and on the meridional location of the associated eastward currents. OFU intensity depends on the occurrence, intensity, size and location of cyclonic eddies north of the eastward jet. Second, our results show that the summer integral of intensity of SCU and OFU is not only driven by the average summer wind, but also by its variability

at higher frequency. Previous studies already showed the role of wind peaks in the development of OFU and SCU, and we further show here that the timing of those peaks inside the season and inside the month plays an important role. For a given summer average wind intensity, SCU is stronger when wind peaks occur regularly throughout the four JJAS months, compared to situations where peaks occur intermittently during some of the months. In contrast, OFU is strengthened when wind peaks mainly occur during the July-August period, potentially due to a stronger offshore circulation during

this period. Moreover, those peaks should be sufficiently spaced to maintain a strong upwelling over a long period. Those conclusions should be verified and deepened in dedicated simulations. For NCU, our study confirms that the interannual variability of upwelling intensity is partly driven by the intensity of summer monsoon wind : conditions that favor to upwelling in the three other areas prevent NCU. However, we reveal that the role of wind is less important than for the other areas, and that the submesoscale to mesoscale circulation that prevails over the coastal area strongly impacts NCU

development. NCU is inhibited when alongshore (northward and southward) currents prevails, and enhanced when offshore circulation prevails. Further studies including ensemble simulations are now required to understand which factors, including OIV, favor or prevent the development of those conditions, including of secondary dipoles, along the northern coast.

We showed that MKU develops along the northern flank of the northeastward current in BoxMK, which is part of the wind-induced summer circulation in the offshore region. MKU interannual variability is mostly driven by the interannual variability of this offshore circulation. However, contrary to what is concluded for SCU, OFU and NCU, MKU is hardly influenced by mesoscale dynamics of strong chaotic nature, and the spatial structure of currents and upwelling over BoxMK is very stable interannually.


Results presented in this paper therefore allowed to 1) quantitatively show the quality of a high-resolution model implemented over the Vietnamese coastal area for the study of the SVU, and 2) to deepen the existing knowledge about the interannual variability of the SVU over its different areas of development. Sensitivity experiments should now be performed to further explore the different physical processes and scales of variability involved in the SVU, including

daily variability of wind and circulation, OIV, rivers, tides and other non-interannually varying factors. In particular, we did not explore in this paper the mechanisms involved in the development of MKU, but dedicated studies are now required to understand into details the functioning and variability of this upwelling. Last, this hydrodynamical numerical model is openly available for the scientific community willing to investigate the functioning, variability and influence of ocean

dynamics in the area. It can be used to study other questions related to ocean circulation and dynamics in the region. It can also be coupled with biogeochemical or sediment models (Herrmann et al. 2014, 2017, Ulses et al. 2016) to examine the influence of ocean dynamics on ecosystems and sediment dynamics, or used for applied purposes such as pollution monitoring (Estournel et al. 2012, Masumoto et al. 2012, Belharet et al. 2016).

Code and data availability

The SYMPHONIE model is available on the webpage of the SIROCCO group, https://sirocco.obs-mip.fr/. Outputs of the simulation performed over the period 2009-2018 over the VNC configuration are freely available upon request to the authors.

Authors Contribution

To Duy Thai, Marine Herrmann and Claude Estournel designed the experiments and To Duy Thai carried them out, with the support of Thomas Duhaut and Patrick Marsaleix. Patrick Marsaleix, Thomas Duhaut and Claude Estournel developed the model code. To Duy Thai, Trinh Bich Ngoc and Patrick Marsaleix worked on the model optimization. Bui Hong Long organized the IO-18 survey. To Duy Thai and Marine Herrmann prepared the manuscript with contributions from all co-authors.

Competing interests

The authors declare that they have no conflict of interest.

Acknowledgements

This work is a part of LOTUS international joint laboratory (lotus.usth.edu.vn). PhD studies of To Duy Thai were funded through an IRD ARTS grant and a "Bourse d'Excellence" from the French Embassy in Vietnam. Numerical simulations were performed using CALMIP HPC facilities (project P13120) and the cluster OCCIGEN from the CINES group (project DARI A0080110098). IO-18 campaign belongs to "Study on processes of air-sea-land interactions and environmental variations of Bien Dong in the context of climate change within the framework of IOC\WESTPAC" project, code: ĐTĐL.CN-28/17. Glider data were collected by the Vietnam Center for Oceanography (CFO, VASI) in the framework of the US-Vietnam collaboration project: "Gulf of Tonkin Circulation study" (NICOP N62909-15-1-2018), financed by the Office of Naval Research (ONR). This paper is a contribution to celebrate the 100 years Anniversary of the Institute of Oceanography, Vietnam Academy of Science and Technology. We warmly thank the two reviewers for the useful and constructive comments that helped improving the quality of the paper.

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

Tables

Table 1. Information about in-situ measurements used in this study

| Name of dataset | Period | Number of stations/points | Type of sensor |
| --- | --- | --- | --- |
| ARGO | 01.01.2009-31.12.2018 | 3675 | CTD, (T,S profiles) |
| ALIS -TSG | 22.05-21.07.2014 | 573,803 (1 point every ~40s) | TSG (SSS, SST) |
| Glider | 22.01-16.05.2017 | 552 | CTD (T,S profiles) |
| IO-18 | 12.09-25.09.2018 | 43 | CTD (T,S profiles) |

Table 2 : From 1st to last row : Temporal mean and standard deviation of $UI_{y,boxN}$ over 2009-2018 for each box and coefficient of variation CV defined as the ratio between the standard deviation (2nd row) and mean (1st row), mean of JJAS average wind stress over each box, mean of integrated positive vorticity over BoxOF ; correlations (correlation coefficient and associate p-values) between time series of significant factors : yearly upwelling index over each box vs. yearly upwelling index over other boxes, vs. JJAS and JA average wind stress averaged over each box, vs. integrated positive vorticity over BoxOF ; average JJAS wind stress over each box vs. average JJAS wind stress over other boxes and vs. integrated positive vorticity over BoxOF ; average JJAS wind stress curl over each box vs. average JJAS wind stress over other boxes and vs. integrated positive vorticity over BoxOF. Correlations significant at more than 95% (p<0.01) are highlighted in bold.

| | BoxNC | BoxSC | BoxOF | BoxMK |
|---|---|---|---|---|
| Mean of $UI_{y,boxN}$ (°C) | 0.16 | 0.80 | 0.07 | 0.07 |
| STD of $UI_{y,boxN}$ (°C) | 0.12 | 0.42 | 0.09 | 0.06 |
| CV (%) | 72 | 53 | 126 | 85 |
| Mean of $WS_{JJAS,boxN}$ (N.m$^{-3}$) | 0.03 | 0.09 | 0.08 | 0.07 |
| Mean of $\zeta_{+,OF}$ (s$^{-1}$) | 1.95x10$^{-6}$ | | | |
| Correlation between : | $UI_{y,NC}$ | $UI_{y,SC}$ | $UI_{y,OF}$ | $UI_{y,MK}$ |
| $UI_{y,SC}$ | +0.00(0.98) | 1 | +0.73(0.02) | +0.83(0.00) |
| $UI_{y,OF}$ | -0.26(0.47) | +0.73(0.02) | 1 | +0.92(0.00) |
| $UI_{y,MK}$ | -0.19(0.59) | +0.83(0.00) | +0.92(0.00) | 1 |
| $WS_{JJAS,NC}$ | -0.09(0.78) | -0.41(0.24) | +0.23(0.53) | +0.07(0.85) |
| $WS_{JJAS,SC}$ | -0.13(0.73) | +0.85(0.00) | +0.76(0.01) | +0.83(0.00) |
| $WS_{JJAS,OF}$ | -0.19(0.61) | +0.81(0.00) | +0.77(0.01) | +0.80(0.01) |
| $WS_{JJAS,MK}$ | -0.08(0.82) | +0.78(0.01) | +0.63(0.05) | +0.72(0.02) |
| $WS_{JA,NC}$ | +0.04(0.90) | +0.18(0.62) | +0.54(0.11) | +0.38(0.28) |
| $WS_{JA,SC}$ | -0.15(0.69) | +0.70(0.03) | +0.84(0.00) | +0.84(0.00) |
| $W_{JA,OF}$ | -0.15(0.67) | +0.69(0.03) | +0.84(0.00) | +0.82(0.00) |
| $W_{JA,MK}$ | -0.11(0.77) | +0.72(0.02) | +0.78(0.01) | +0.82(0.00) |
| Integrated Positive vorticity OF $\zeta_{+,OF}$ | -0.28(0.43) | +0.60(0.07) | +0.69(0.03) | +0.74(0.01) |
| Correlation between : | $WS_{JJAS,NC}$ | $WS_{JJAS,SC}$ | $WS_{JJAS,OF}$ | $WS_{JJAS,MK}$ |
| $WS_{JJAS,SC}$ | -0.20(0.58) | 1 | 0.99(0.00) | +0.97(0.00) |
| $WS_{JJAS,OF}$ | -0.19(0.60) | +0.99(0.00) | 1 | +0.96(0.00) |
| $WS_{JJAS,MK}$ | -0.31(0.39) | +0.97(0.00) | +0.96(0.00) | 1 |
| $\zeta_{+,OF}$ | +0.02(0.95) | +0.89(0.00) | +0.89(0.00) | +0.81(0.01) |
| Correlation between : | $WSC_{JJAS,NC}$ | $WSC_{JJAS,SC}$ | $WSC_{JJAS,OF}$ | $WSC_{JJAS,MK}$ |
| $WS_{JJAS,SC}$ | 0.40(0.26) | 0.95(0.00) | 0.69(0.03) | -0.96(0.00) |
| $WS_{JJAS,OF}$ | 0.42(0.23) | 0.96(0.00) | 0.71(0.02) | -0.92(0.00) |
| $WS_{JJAS,MK}$ | 0.24(0.51) | 0.96(0.00) | 0.83(0.00) | -0.91(0.00) |
| $\zeta_{+,OF}$ | 0.59(0.07) | 0.82(0.00) | 0.46(0.18) | -0.80(0.01) |

Figures


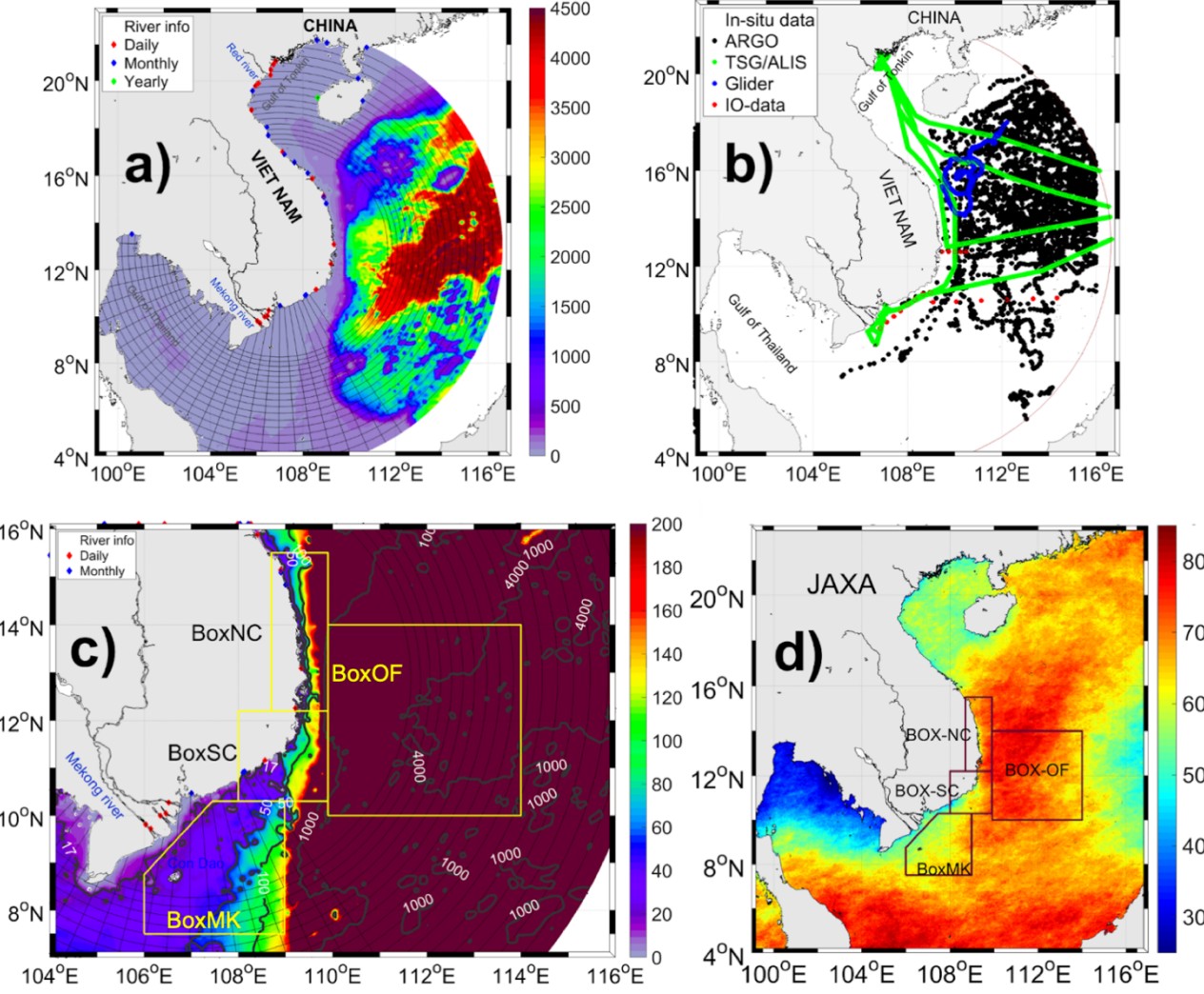

Figure 1. (a) Characteristics of the orthogonal curvilinear computational grid (black lines, not all points are shown for visibility purposes) and bathymetry (colors, m, *GEBCO_2014*) used for the VNC configuration of the SYMPHONIE model. Dots show the location of rivers for which we used daily (red), monthly (blue) and yearly climatology (green) discharge values. (b) Location of *in-situ* data available over the VNC domain: ARGO buoys (black), ALIS R/V campaign (TSG: green), IO-18 data (red), and glider data (blue). (c) Bathymetry (m) over the SVU region. Yellow boxes show the

location of the 4 boxes used for the study of SVU. (d) Percentage of days during which JAXA data are available during summer 2018.

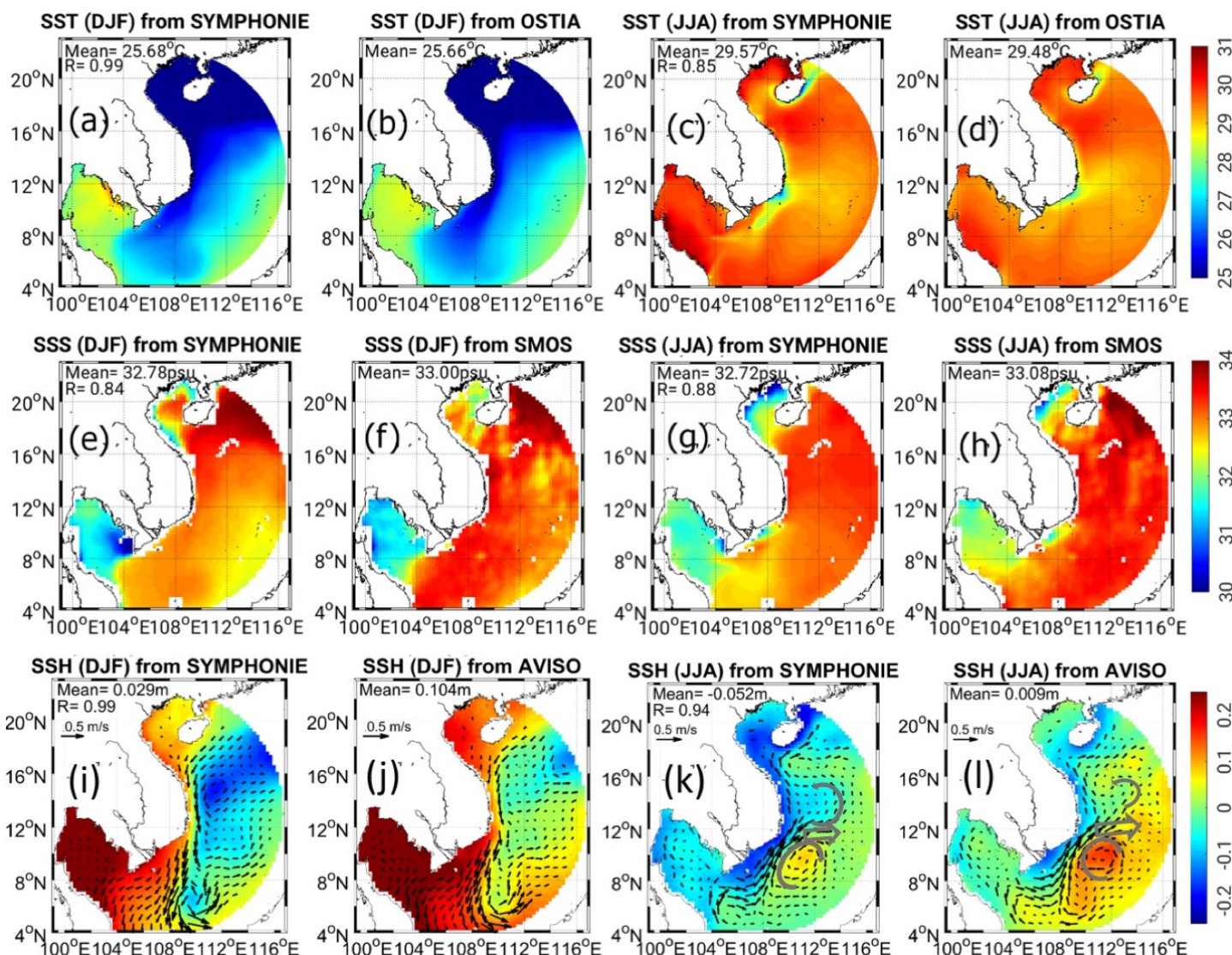

Figure 2: Spatial distribution of simulated and observed winter (DJF) and summer (JJA) climatological averages of SST (a,b,c,d, °C), SSS (e, f, g, h), SSH (i,j,k,l, m) and total surface geostrophic current (m.s$^{-1}$), and spatial correlation coefficient R (here the p-value is always smaller than 0.01). Grey arrows on panels k-l highlight the summer AC/C dipole and eastward jtmet.


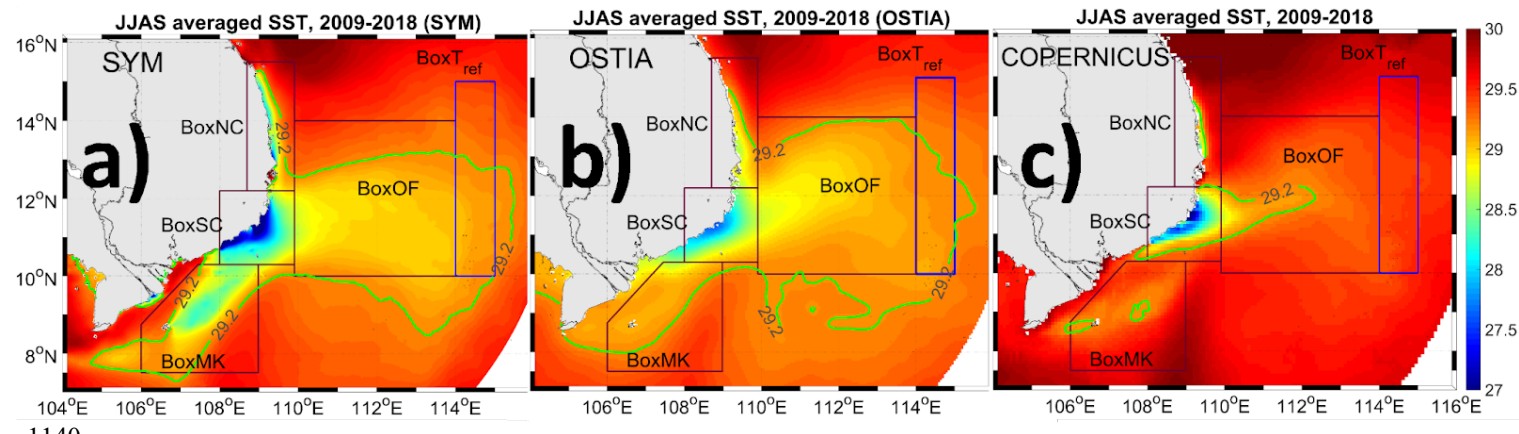


Figure 3: JJAS average over 2009-2018 of SST (°C) from (a) SYMPHONIE, (b) OSTIA and (c) COPERNICUS over the SVU region. Black boxes show the 4 upwelling areas, and the blue rectangles shows the reference box. Coordinates of the boxes : BoxNC (12.2-15.5ºN; 108.7-109.9ºE), BoxSC (10.3-12.2ºN; 108-109.9ºE), BoxOF (10-14ºN; 109.9-114ºE), BoxMK (at depth > 17m, 7.5-10.3ºN; 106-109ºE). BoxT$_{Ref}$ (10-15ºN; 114-115ºE)


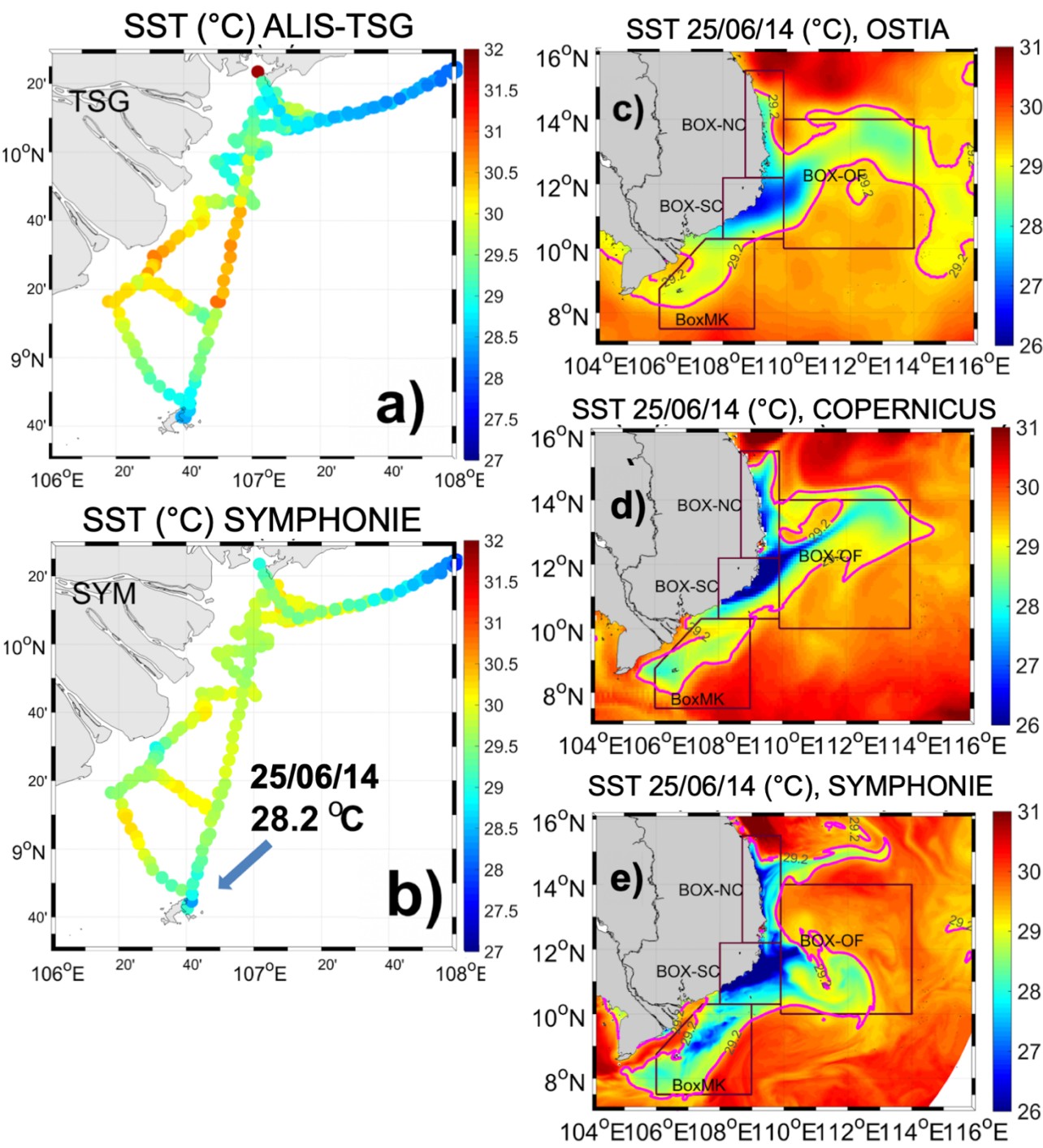

Figure 4: Left : Simulated (SYMPHONIE, a) and observed (ALIS-TSG, b) SST (°C) during ALIS R/V trajectory offshore the Mekong mouth in June 2014. The arrow shows the location of minimum SST (~28.2°C both in data and model) recorded near Con Dao Island (~8.6°E − 106.6°E) on 25/06/2014. Right : SST on 25/06/2015 (°C) in OSTIA (c), COPERNICUS (d) and SYMPHONIE (e). The pink contour shows the isotherm Tref=29.2°C.

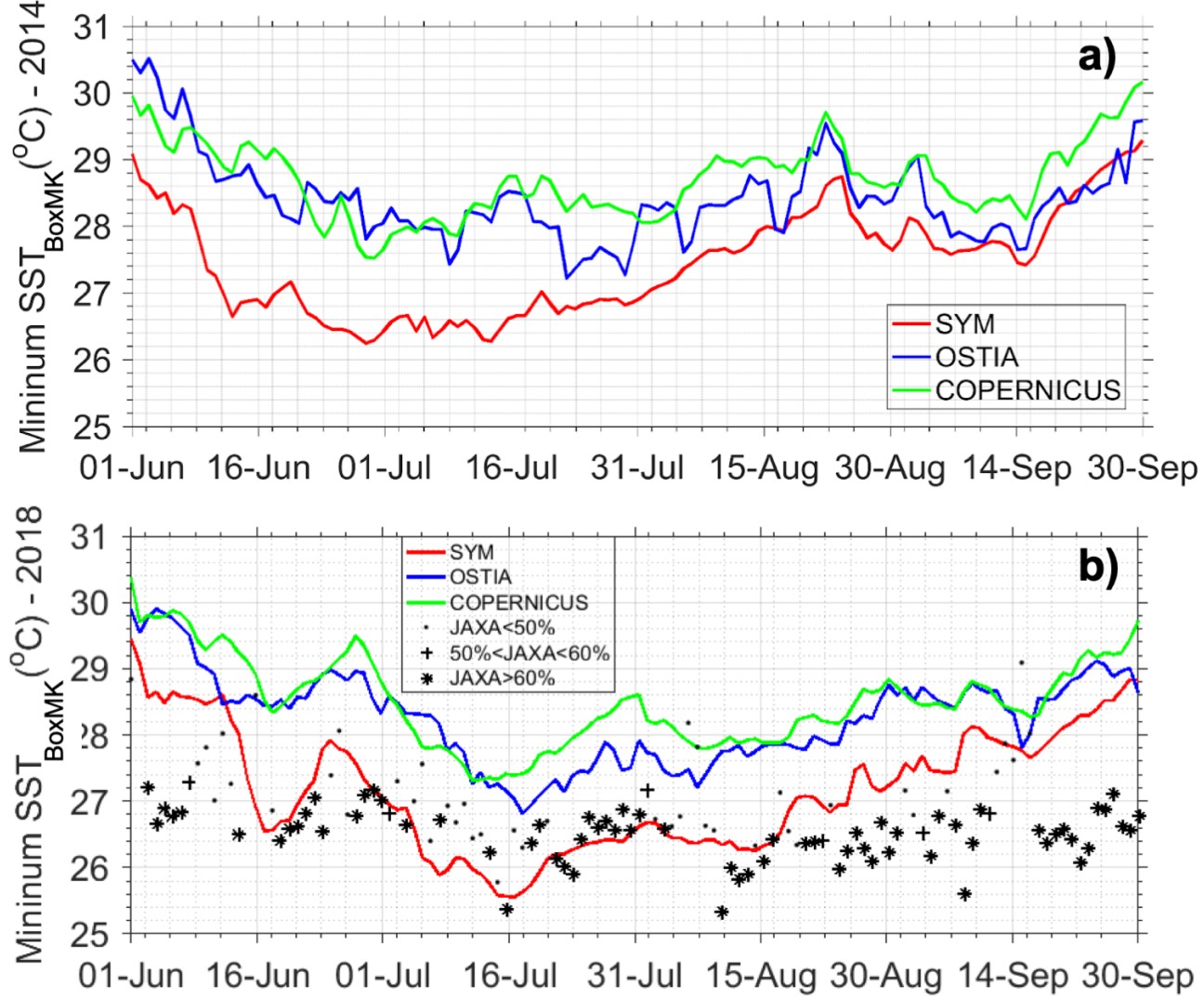

Figure 5: Daily time series of minimum SST (°C) over BoxMK (a) during summer 2014 in SYMPHONIE, OSTIA, COPERNICUS, and (b) during summer 2018 in SYMPHONIE, OSTIA, COPERNICUS and JAXA. Dots, crosses and stars correspond respectively to a spatial coverage of BoxMK lower than 50%, higher than 50%, and higher 60% for JAXA. We only consider JAXA pixels of quality levels ≥ 4.


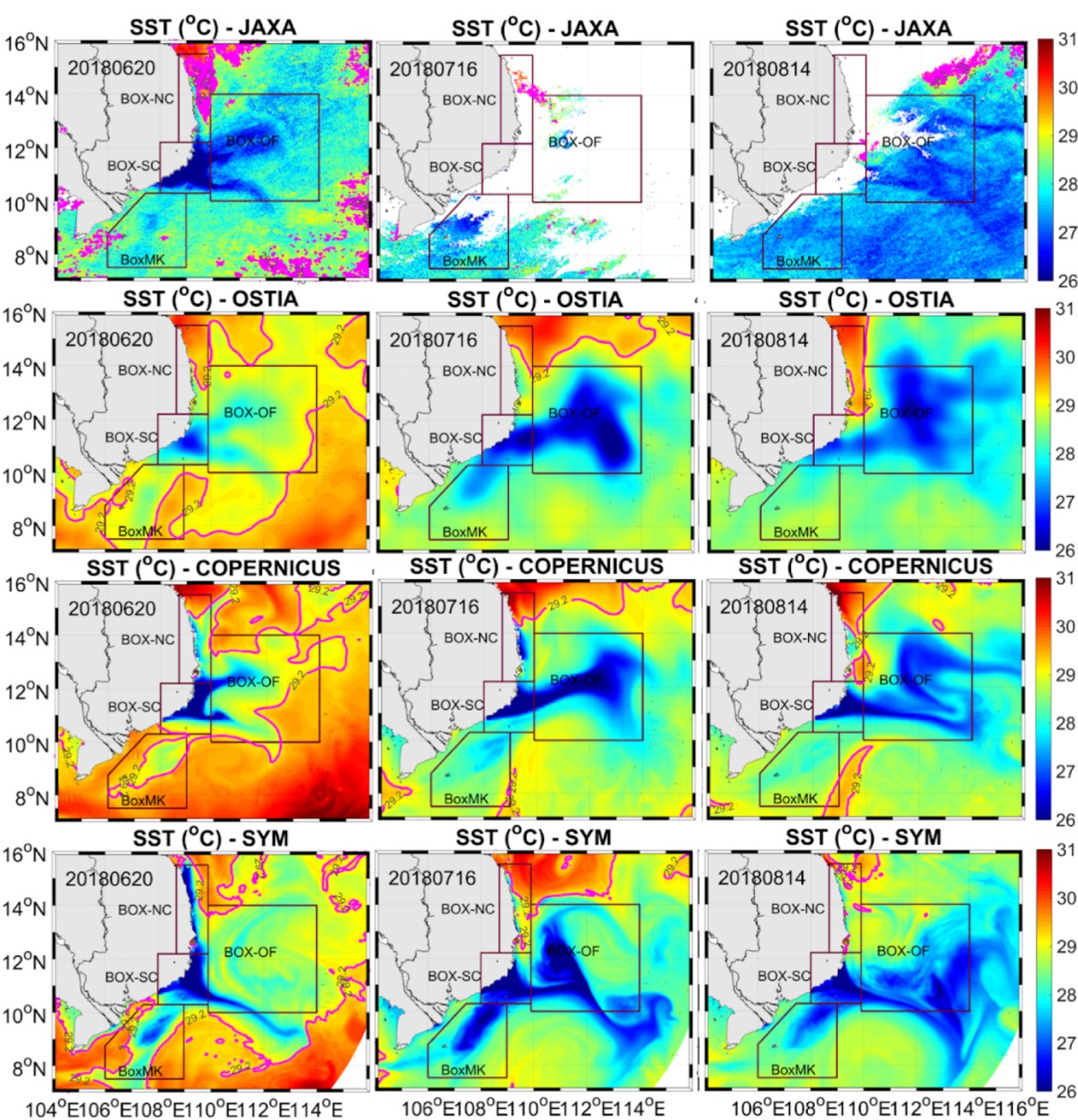


Figure 6: Daily SST (°C) on 20/06/2014 (left), 16/07/2014 (middle) and 14/08/2014 (right) from JAXA (1st row), OSTIA (2nd row), COPERNICUS (3rd row) and SYMPHONIE (4th row). The pink contour shows the isotherm Tref.

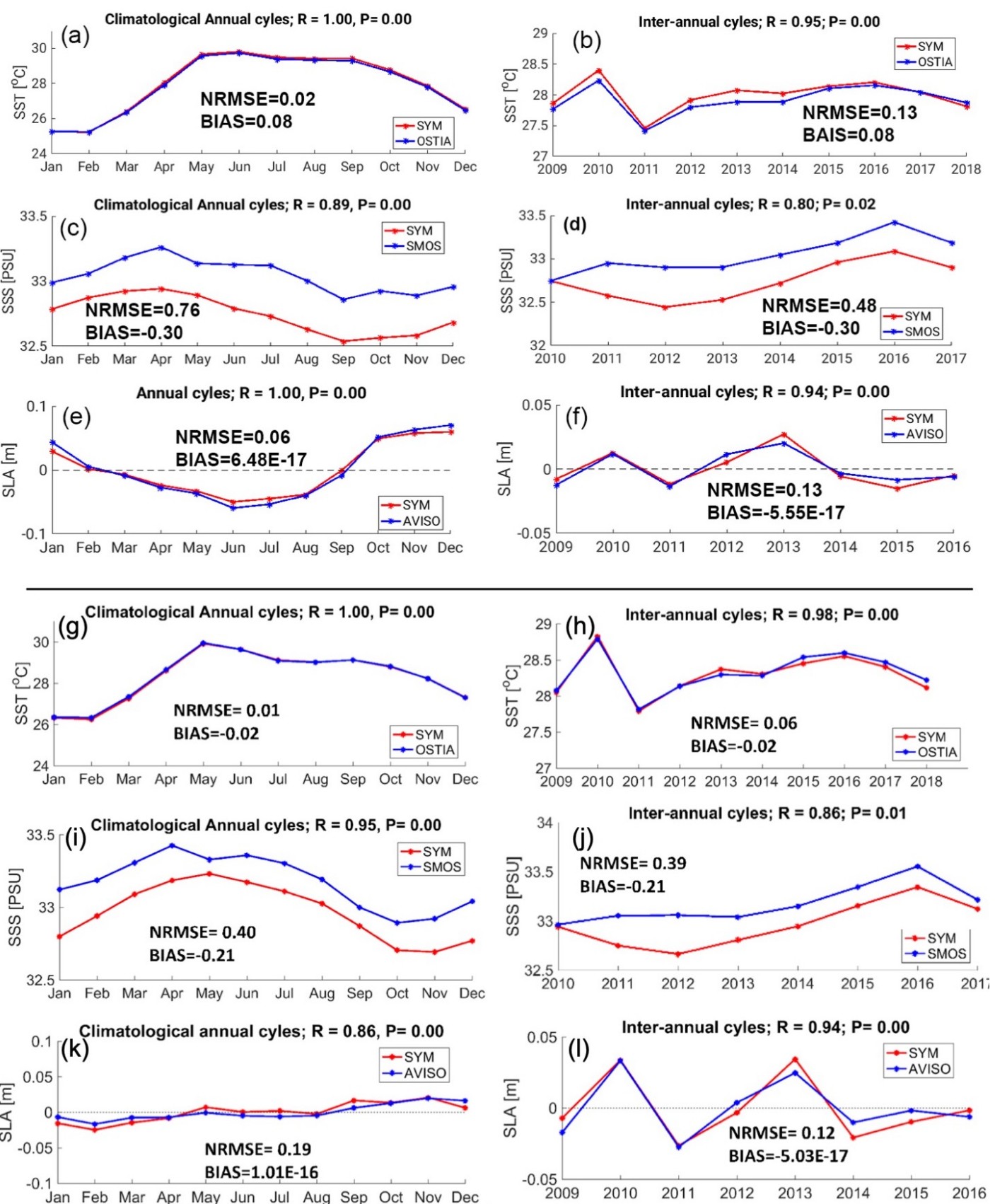


Figure 7: Climatological monthly time series (left) and interannual yearly time series (right) of simulated (red) and observed (blue) SST (°C), SSS and SLA (m) averaged over (a to f) the VNC domain and over (g to l) the SVU area (104-116°E; 7-16°N). Correlation (correlation coefficient R and p-value P), NRMSE and bias of simulated vs. observed values are indicated.

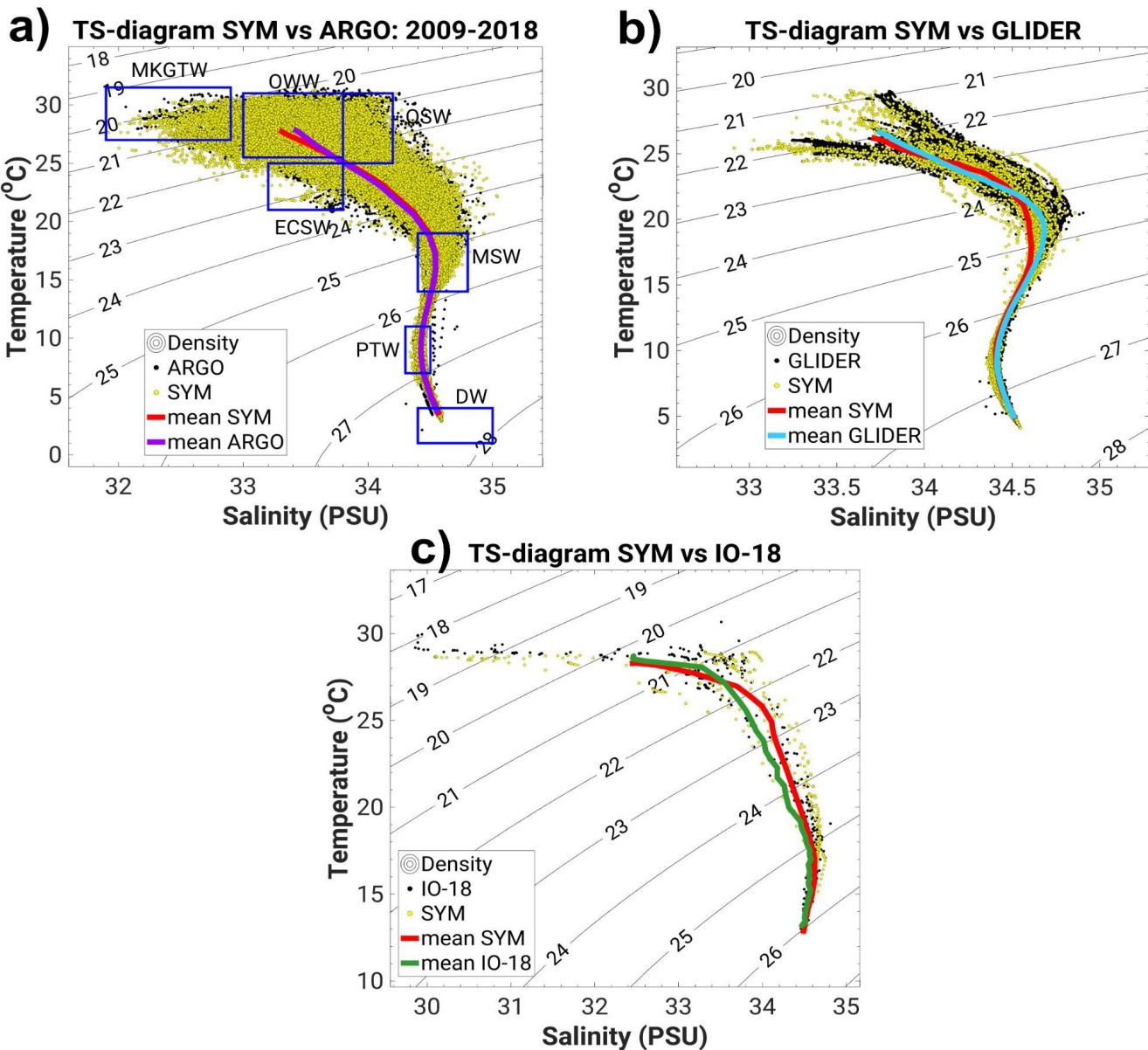

Figure 8: TS-diagram built from observations from (a) ARGO (black dots and purple line for dots average), (b) GLIDER (black dots, cyan line), (c) IO-18 (black dots, green line), and from SYMPHONIE colocalized outputs (yellow dots, red line). Blue rectangles represent water masses in the SCS (Uu and Brankard 1997, Dippner et al. 2011) : MKGTW (Mekong & Gulf of Thailand Water), OWW (Offshore Warm Water), OSW (Offshore Salty Water), ECSW (East China Sea Water), MSW (Maximum Salinity Water), PTW (Permanent Thermocline Water), DW (Deep Water).


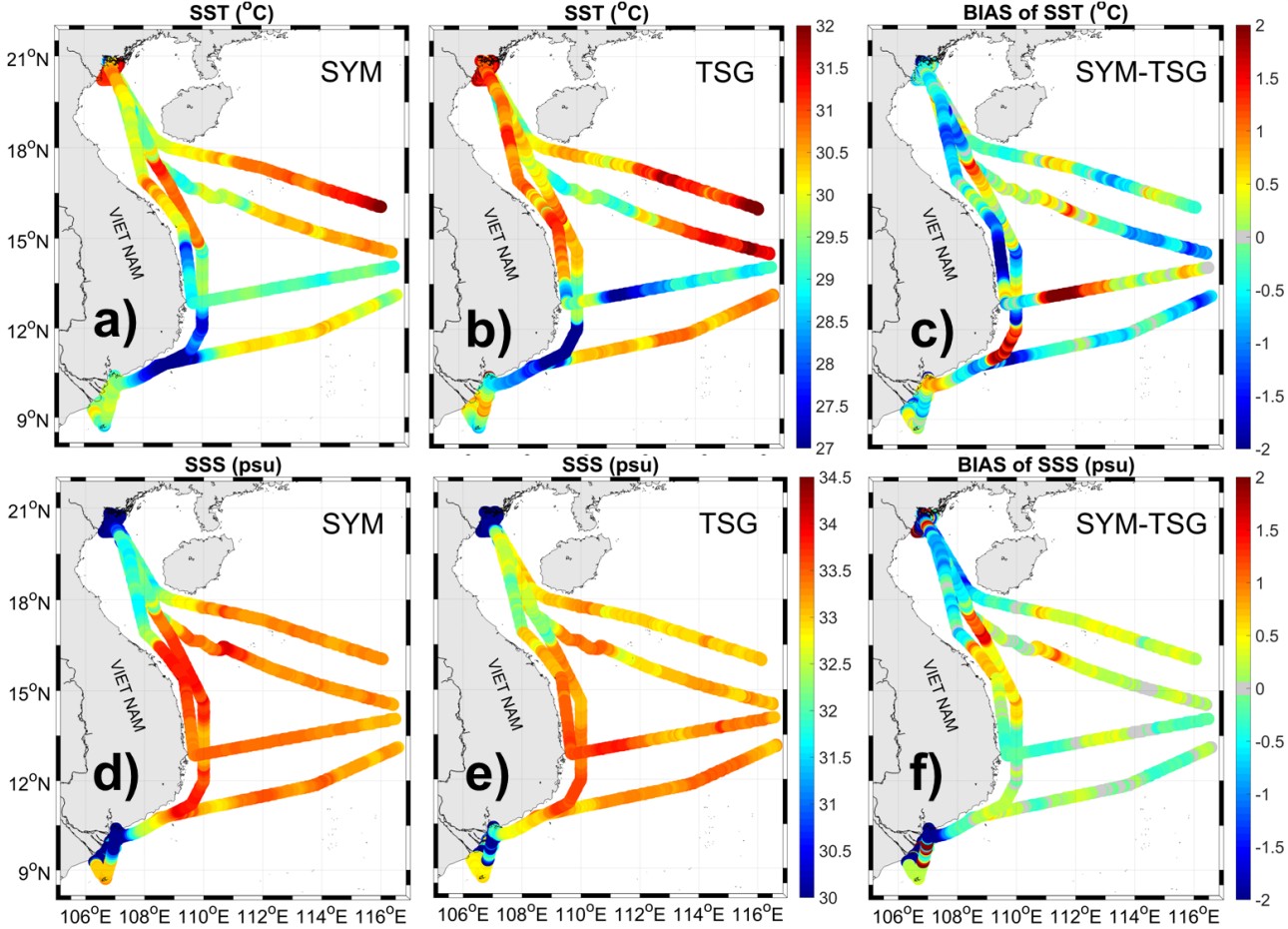

Figure 9: Values of SST (top) and SSS (bottom) from (a,d) ALIS-TSG trajectory during summer 2014 and from (b,e)
SYMPHONIE colocalized outputs, and (c,f) bias between SYMPHONIE and ALIS-TSG.

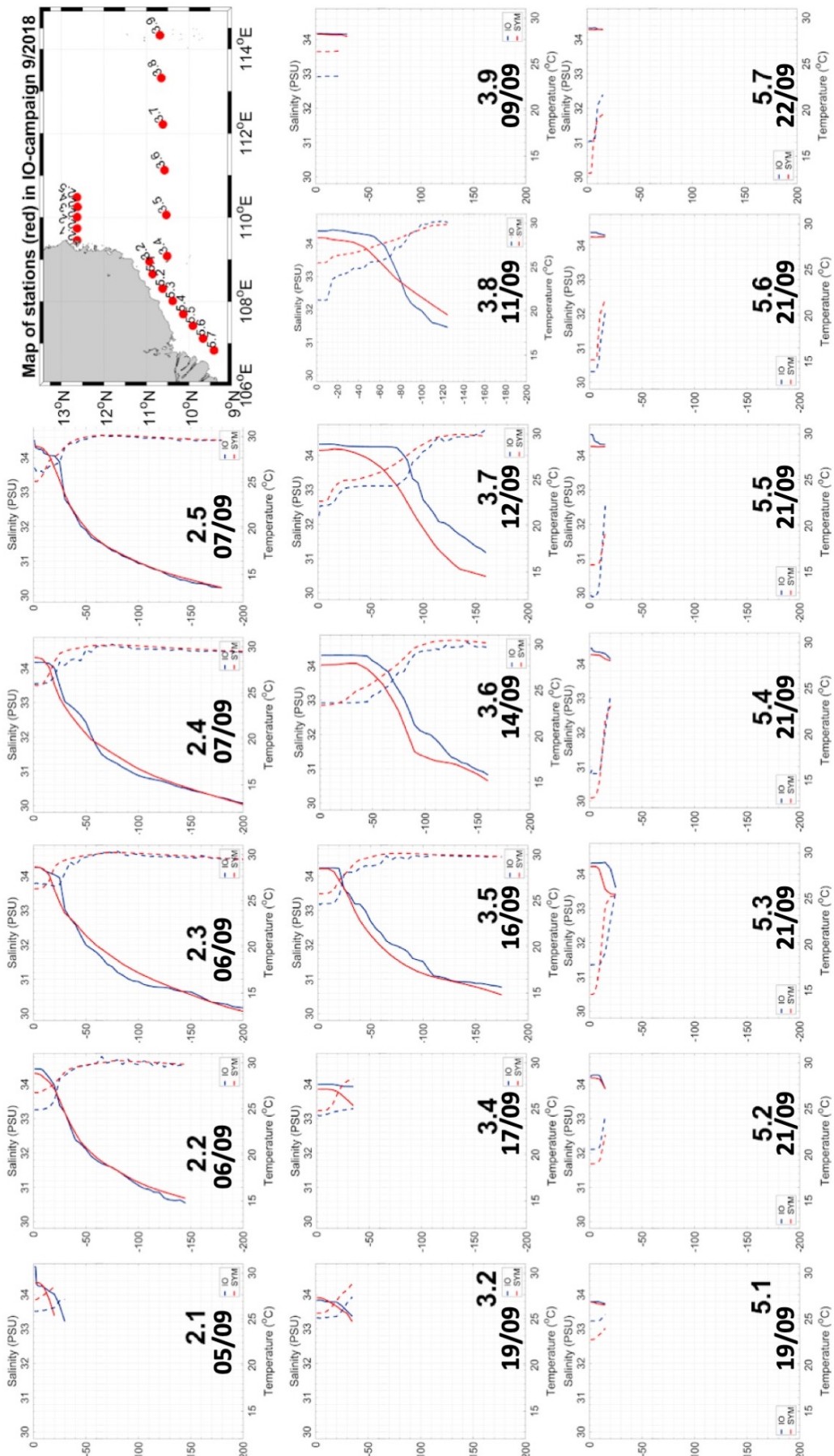

Figure 10: Temperature (°C) and salinity profiles sampled during IO-18 campaign (blue) and simulated by SYMPHONIE (red) between 12/09 and 25/09/2018. The station and day of sampling is indicated for each profile.

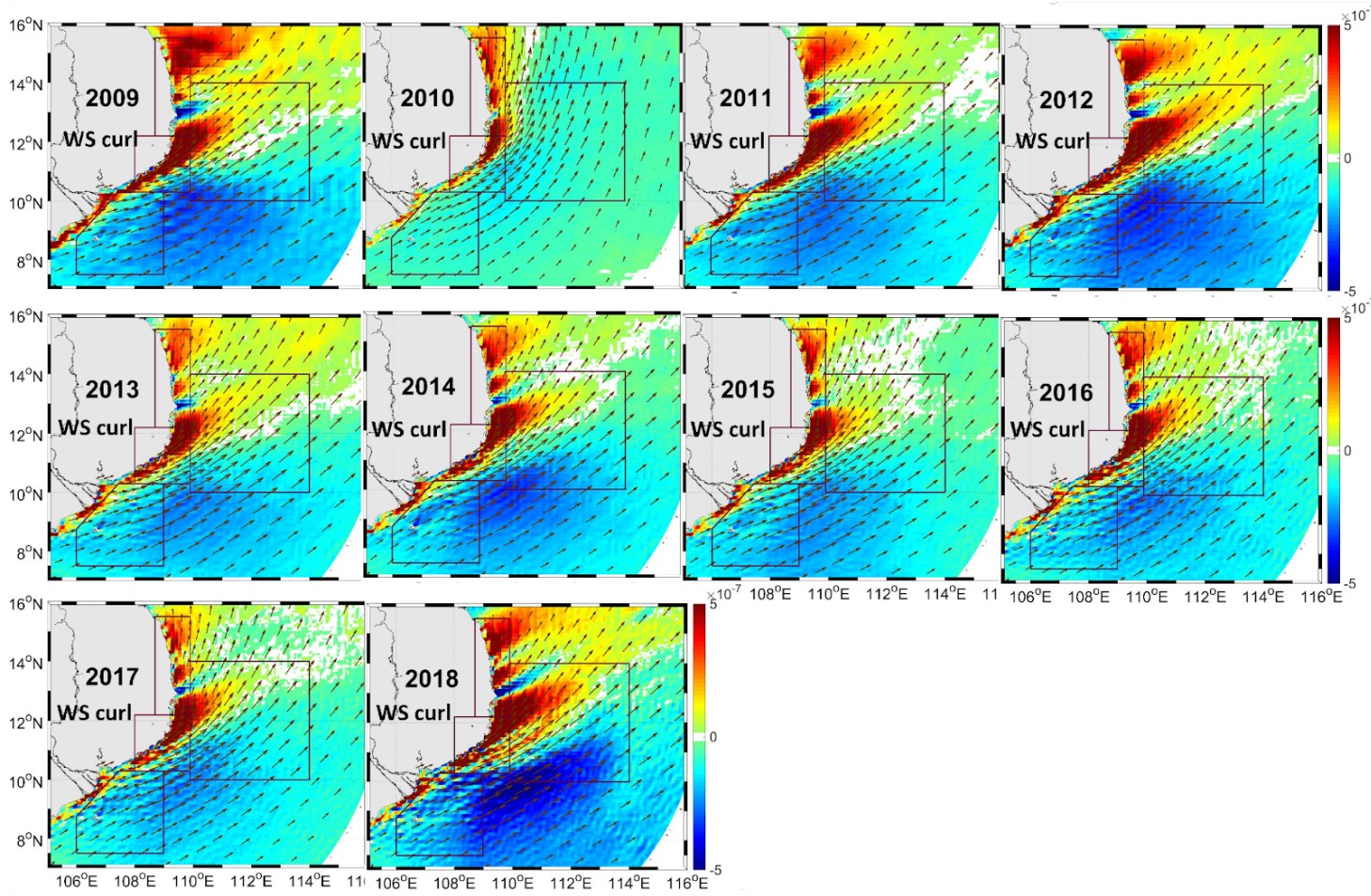

Figure 11: Maps of JJAS averaged wind stress (arrows, N.m$^{-2}$) and wind stress curl (colors, N.m$^{-3}$) for each year of the
simulation.

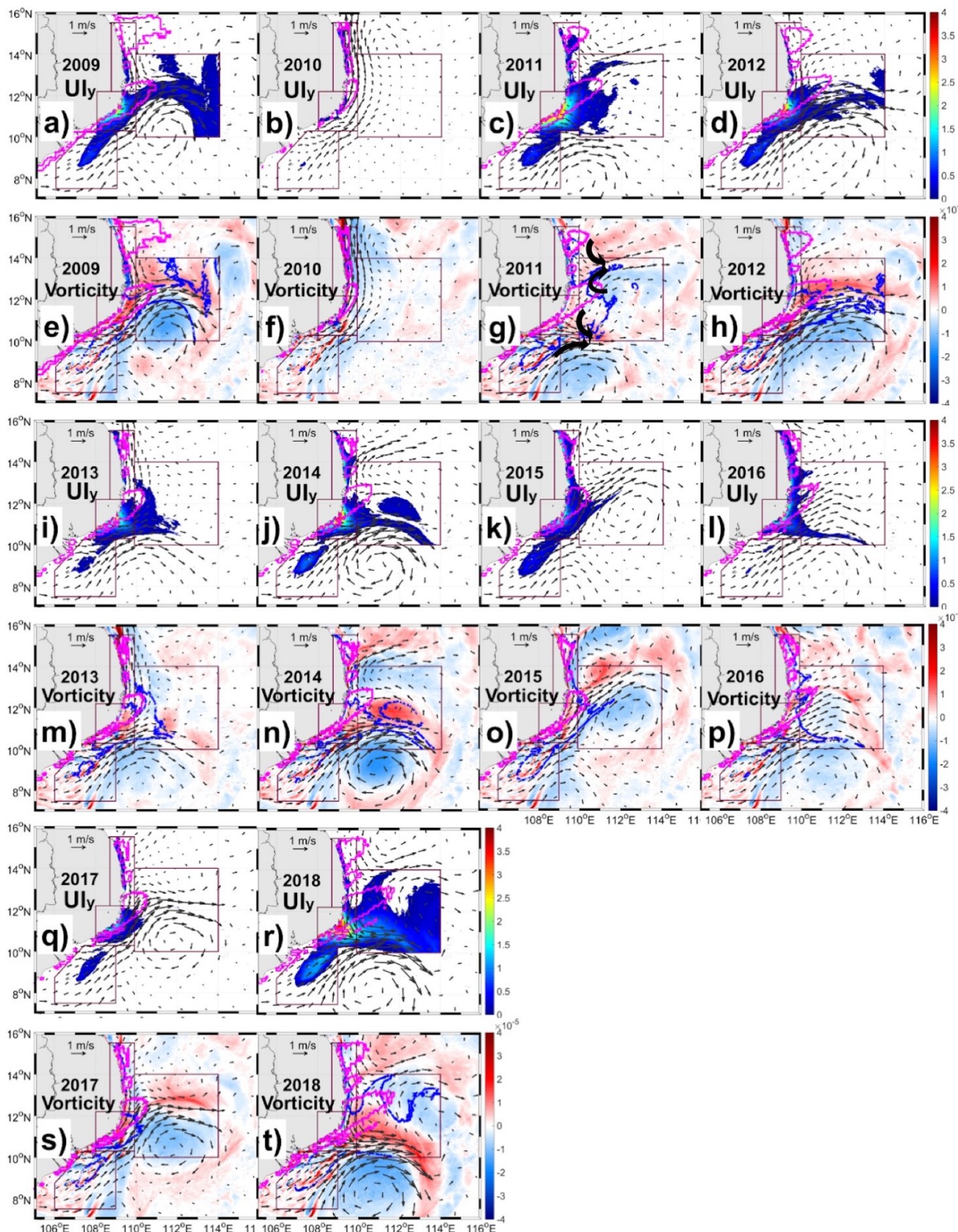

Figure 12: Maps of spatial yearly upwelling index $UI_y$ (°C, a,b,c,d,i,j,k,l,q,r) and of speed and direction (arrows, m.s$^{-1}$) and vorticity (colors, s$^{-1}$) of JJAS averaged surface current (e,f,g,h,m,n,o,p,s,t) for each year of the simulation. Pink contours show the $+3.10^{-7}$N.m$^{-3}$ isoline of JJAS average positive wind stress curl. Blue contours show the 0.1°C isoline of $UI_y$.

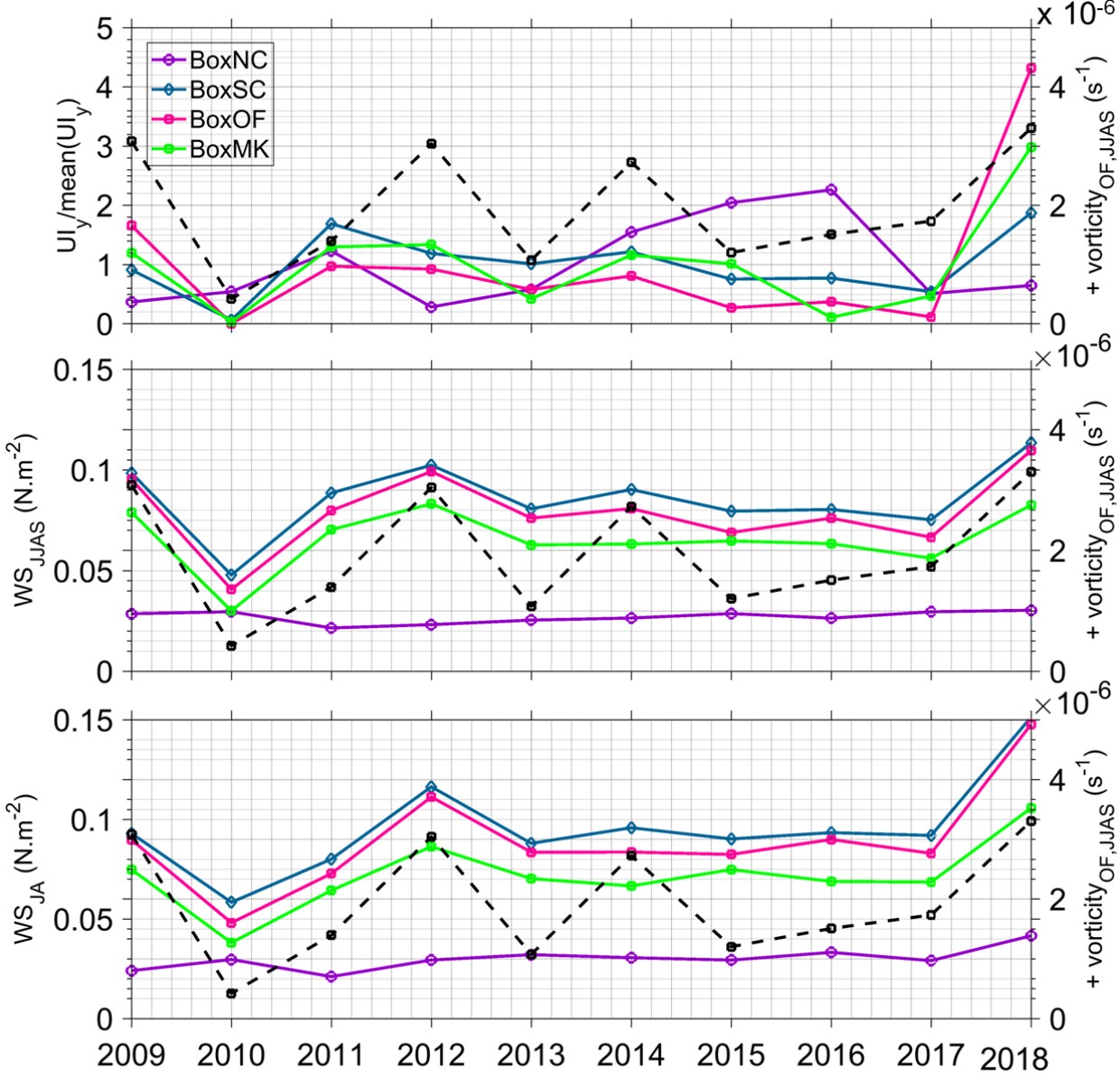

Figure 13: Interannual time series of normalized yearly upwelling index ($1^{st}$ row), average JJAS ($2^{nd}$ row) and JA ($3^{rd}$ row) wind stress ($N.m^{-2}$) over each upwelling box (BoxNC: purple ; BoxSC: blue; BoxOF: pink; BoxMK: green). The black dashed curve shows the integrated positive summer surface current vorticity over BoxOF ($s^{-1}$).

1240

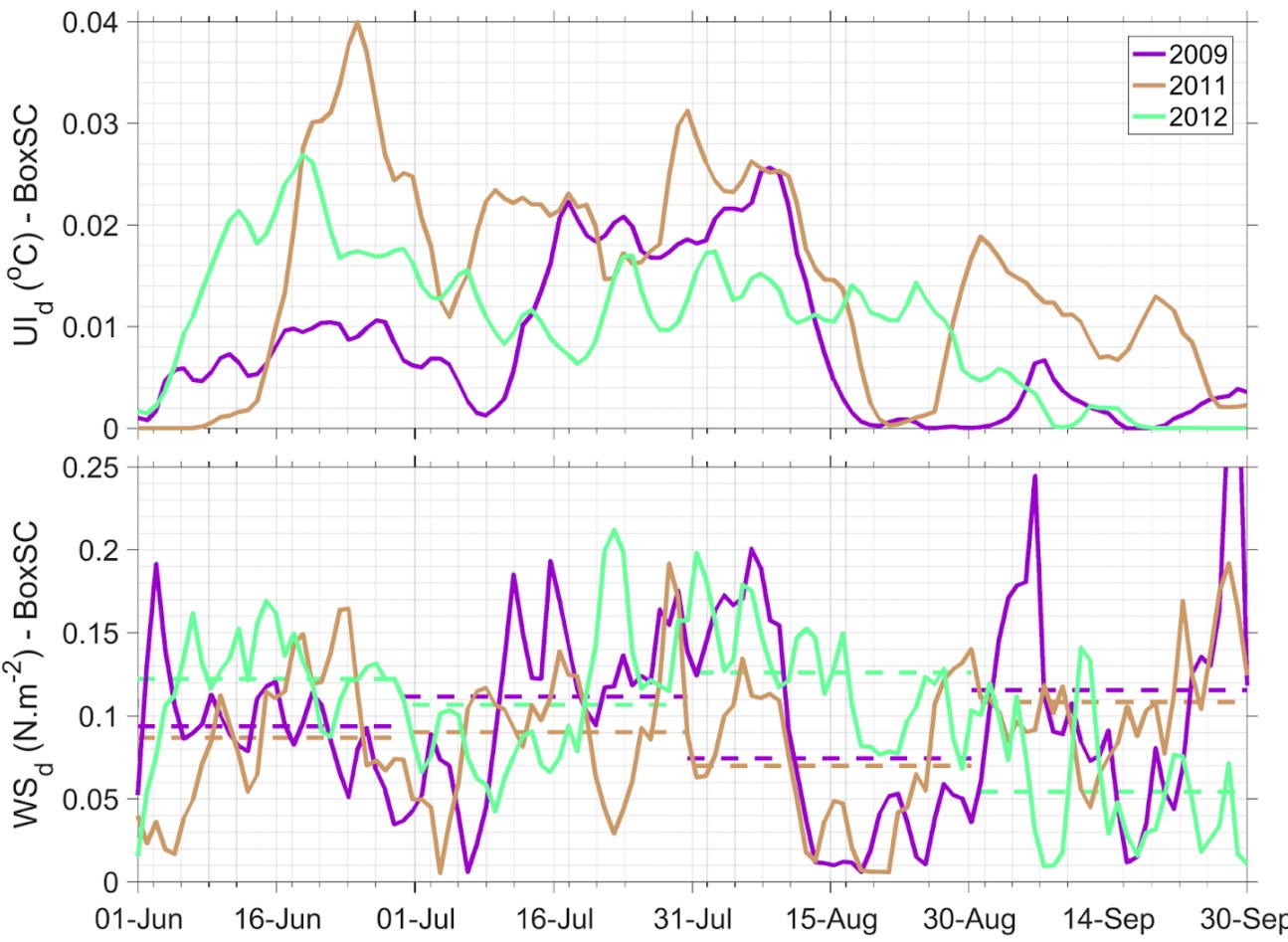

1245

Figure 14: Time series of daily upwelling index (top) and average daily (plain line) and monthly (dashed line) wind stress (bottom) over BoxSC between June and September for years 2009 (purple), 2011 (orange) and 2012 (green). For the sake of readability, the y-axis is cut off for the last extreme value of September 2009, which corresponds to the passage of storm KETSANA over the SVU region (see the NOAA storm track website on https:/coast.noaa.gov/hurricanes).

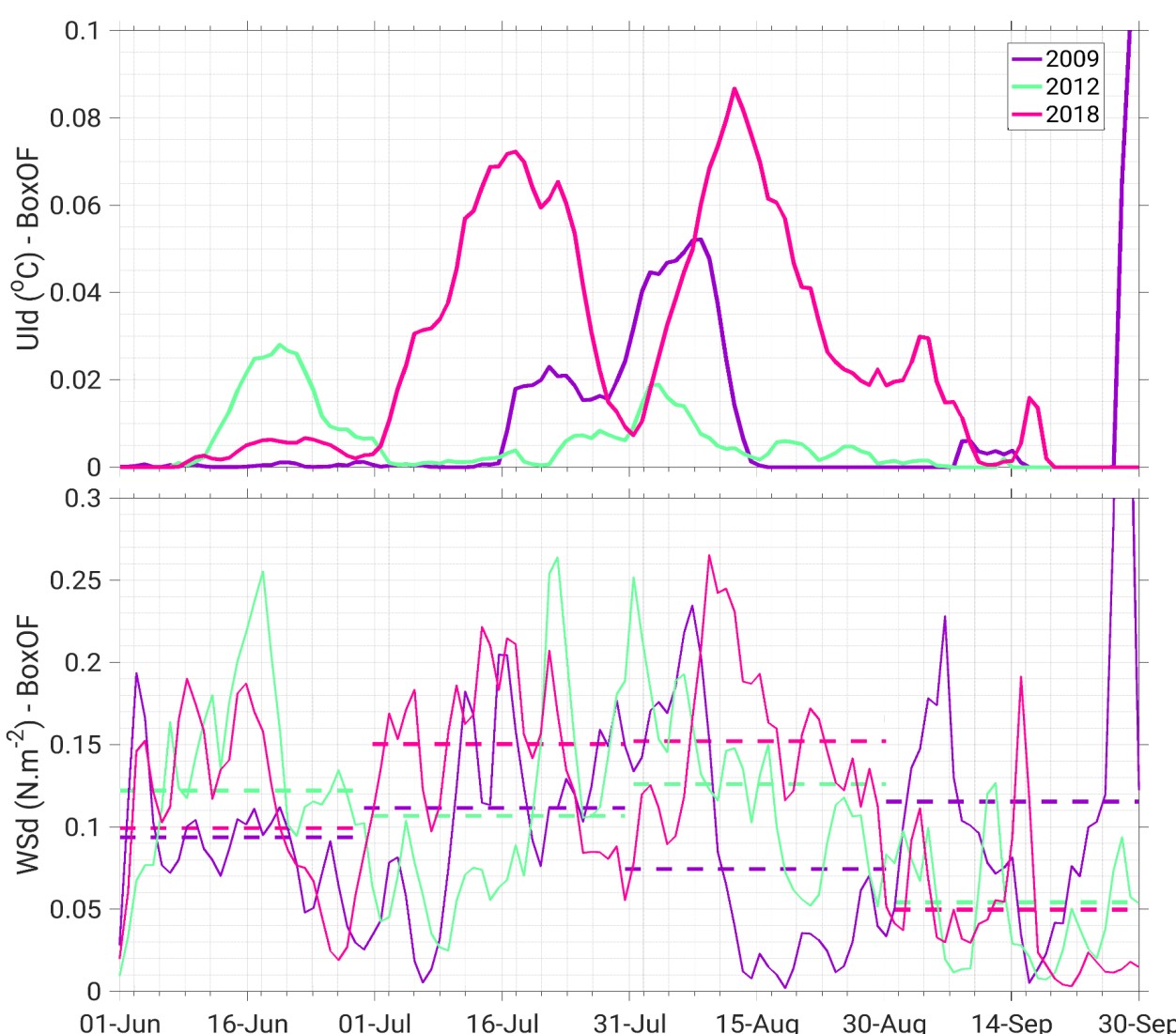

Figure 15: Same as Figure 9 for BoxOF and years 2009 (purple), 2012 (green) and 2018 (magenta). For the sake of
1260  readability, the y-axis is cut off for the last extreme value of September 2009, which corresponds to the passage of
storm KETSANA over the SVU region (see the NOAA storm track website on https:/coast.noaa.gov/hurricanes).

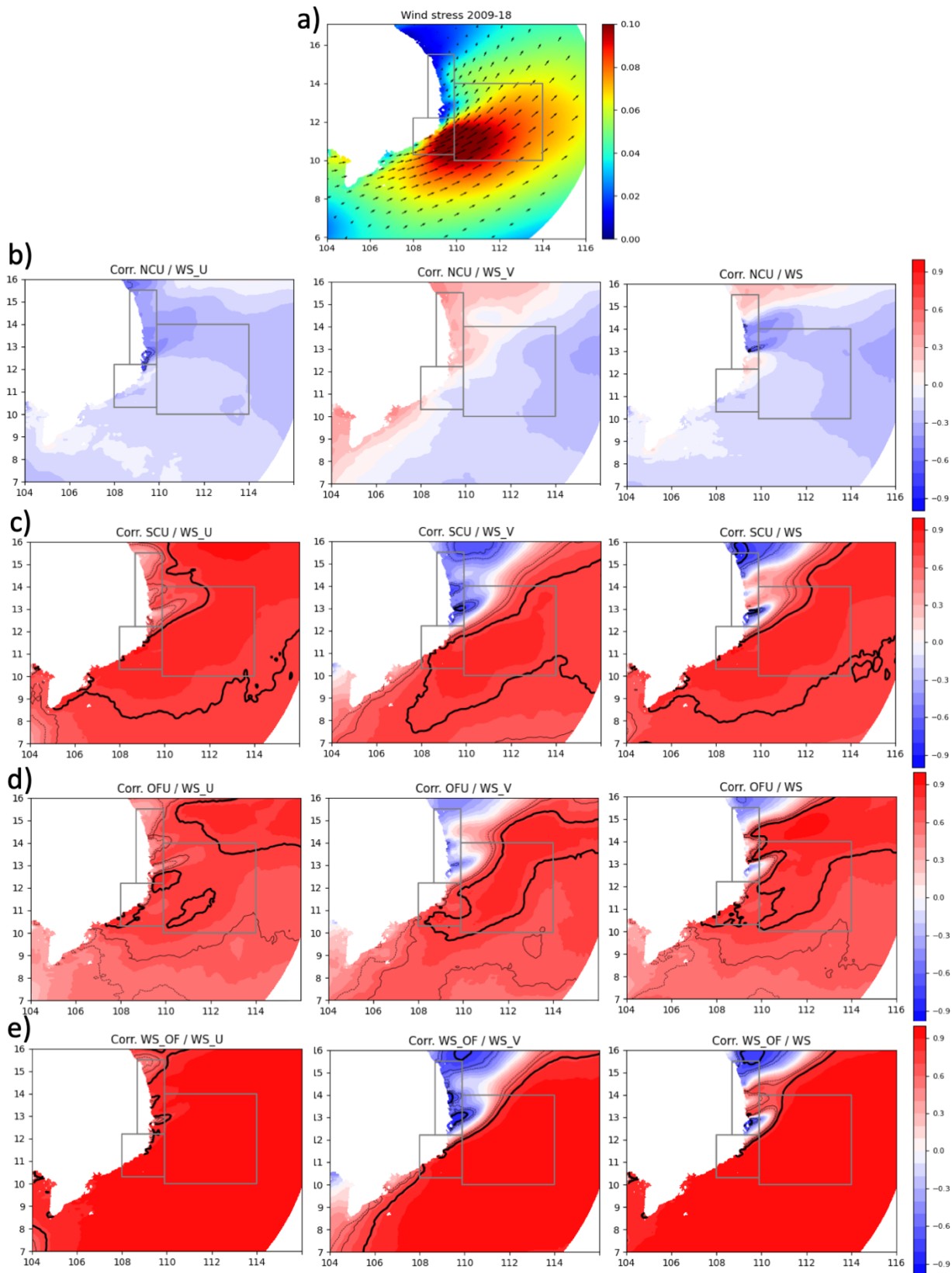

Figure 16 : (a) Direction (arrows) and intensity (N.m$^{-2}$) of average JJAS wind stress over the SVU area during 2009-2018 from ECMWF analysis. (b) to (e) Correlation between yearly time series of the zonal (left) and meridional (middle) components of wind stress and of its intensity (right) and (b) UI$_{y,NC}$ (c), UI$_{y,SC}$ (d), UI$_{y,OF}$ and (d) JJAS wind stress intensity over BoxOF. Thick, plain and dotted lines correspond respectively of the isolines p=0.01, p=0.05 and p=0.10.

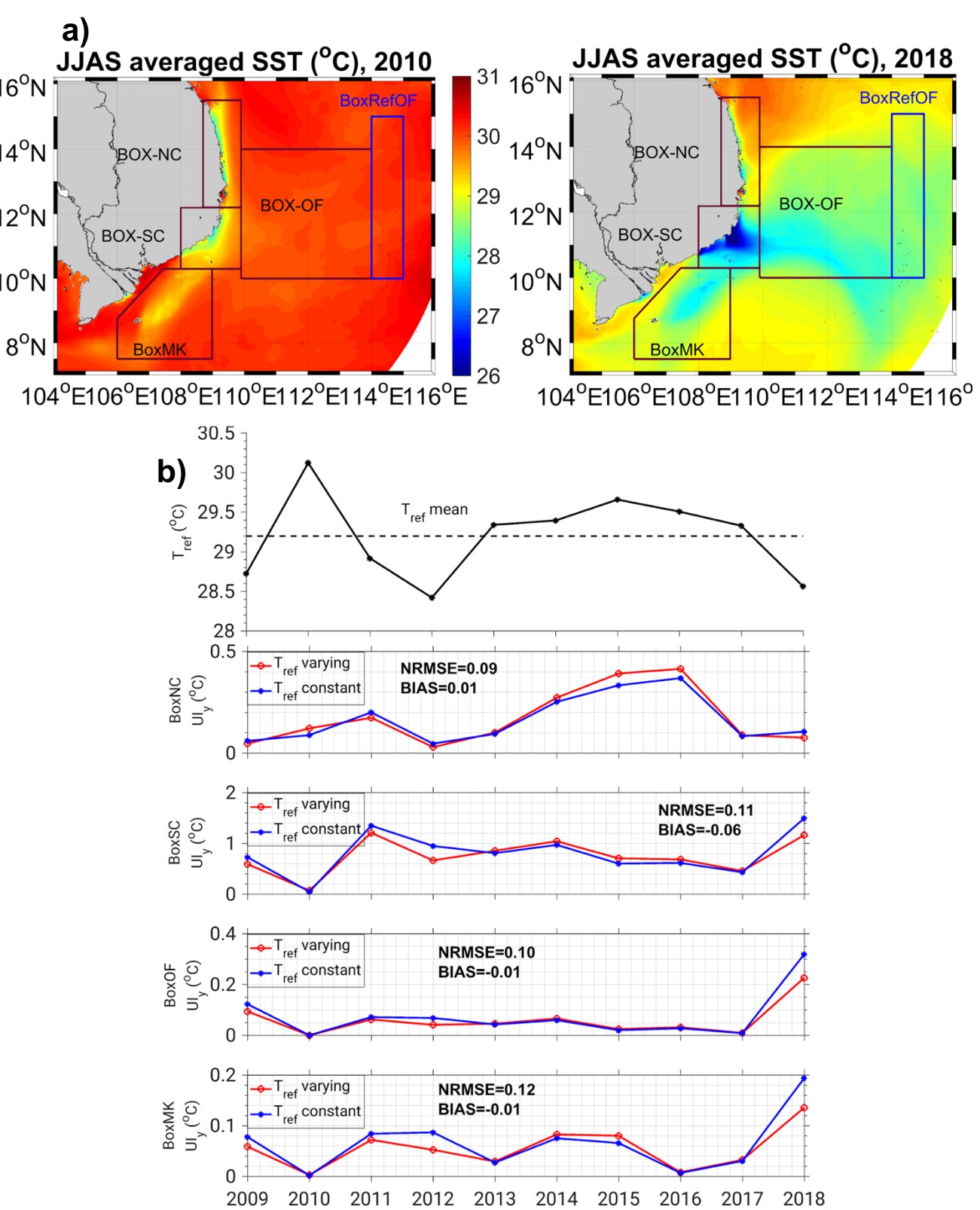

Figure 17: (a) Maps of JJAS averaged SST in 2010 and 2018 in the 2009-2018 simulation (°C). (b) (1st row) Yearly
time series of the JJAS averaged SST over the reference box for each year (i.e. varying Tref, full black line). The
dashed line shows the value of the climatological summer averaged SST over the reference box (i.e. constant Tref). (2nd
to 5th row) Yearly time series of upwelling indexes UIy computed using a varying (red) vs. constant (blue) Tref for
BoxNC, BoxSC, BoxOF and BoxMK.

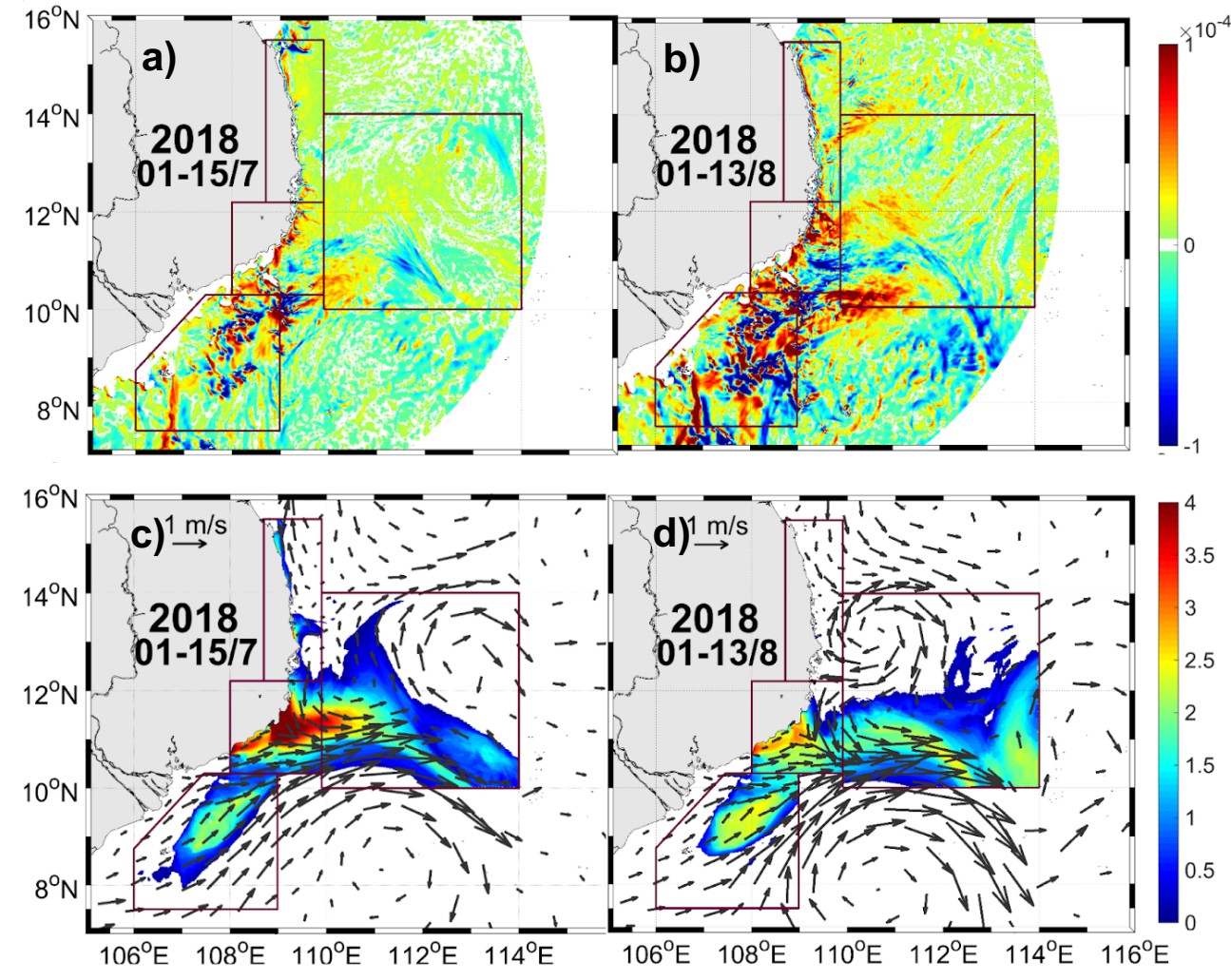

Figure 18 : Vertical velocity (m.s$^{-1}$, 1$^{st}$ row) and daily upwelling index (°C, 2$^{nd}$ row) and surface currents (m.s$^{-1}$) averaged over the July (a,c) and August (b,d) periods of upwelling development in summer 2018.