# Peer review of "Role of wind, mesoscale dynamics and coastal circulation in the interannual variability of South Vietnam Upwelling, South China Sea. Answers from a high-resolution ocean model."

_Ocean Science, 2021_

## Referee Comment (RC1)

**Comments on " Role of wind, mesoscale dynamics and coastal circulation in the interannual variability of South Vietnam Upwelling, South China Sea. Answers from a high resolution" by Tai To Duy et al. in Ocean Science**

In this study, the authors attempted to explore the dynamics governing the interannual variability of South Vietnam Upwelling based on model simulation. Befere discussing the modelling results, they first compared the general patterns of simulated SST, SSS, and SLA with the satellite-remosed data, and compared the simulated temperature and salinity profiles with the in-situ measturements of ARGO, Seaglider, and R/V cruise. After reviewing, I think the manuscript needs an subsustainsial revision before being acceptable based on the following comments.

**Major Comments:**

**1. Methodology:**
a) The authors explored the South Vietnam Upwelling in four areas: BoxOF, BoxNC, BoxSC, and BoxMK. For the first three areas, the cold temperature are evident in both the simulated SST and satellite SST. However, for the BoxMK, the cold temperature seems only appear in simulated SST, but not in satellite SST, although the authors have referred to the finding of two literatures. This argument does not make sense, because a reader cannot directly confirm the rationality of considering BoxMK a spereate area to be explored. If this is not a natural phenomenon, the discussion becomes meaningless.

b) The authors performed the simulation of the model SYMPHONIE from 2009-2018 by comparing output data (SST, SSS, SLA, and T-S profiles) with high-resolution satellite data and in-situ observations, showing that this model is an innovative tool that can reproduce oceanic dynamics properly not only at the surface but also at deeper sub-layers, and at wide-range time scales. To investigate the daily-to-interannual variability of the VNU, however, they employed only the surface data (SST and velocity) and the discussions are all statistically, which brings not many new results in the comparison with previous studies using satellite data. In other words, this study can be performed by the satellite data without SYMPHONIE. I think the authors should utilize the advantage of modelling to conduct numerical experiments to examine whether the proposed factors are really factors controlling the interannual variablility of South Vietnam Upwelling in each area.

c) In Section 2.3, the author introduced several SST-based upwelling indicators (daily, yearly and spatial upwelling index), which are applied for 4 upwelling areas. Each area uses different reference boxes, which is taken as the areas not impacted by surface cooling. However, the boxes (besides RefOF) they chose may be highly possible to be influenced by other upwelling areas. For example, RefNC could be impacted by the offshore upwelling if the offshore upwelling have more northern extension. In addition, the authors use the time-averaged Tref in each Reference box, but the temperature in the SCS suffers interannual variations, e.g., Figure 3b. This could make a great impact on the calculation of SST-UI, and result in a large dependency as discussed in section 4.5.

d) Another concern is that the spatial upwelling index could be not a continuous field as shown in Figure 8 because the authors use different Trefs.

e) Some calculations have been done but not defined (i.e. wind stress, vorticity, coefficient of variation, …). Specifically, the authors adopted wind stress for many places, but they did not define the wind stress: meridional wind stress, zonal wind stress, along-shore wind stress or cross-shore wind stress.

**2. Result and discussion:**

a) The authors wrote long paragraphs to describe known results and few lines for un-solid conclusions. For example:

Section 4.1: The impact of intra-seasonal and inter-annual variability of wind forcing on SCU has been revealed. I suggest the authors reconstruct this section by referencing known results in the introduction, using several sentences to describe the similarity with previous outcomes and highlighting new finding they have discovered. In the case of oceanic factors, word usage is not direct to point, for instant, "background coastal circulation" and "mesoscale structure". Quantitative assessment is missing for the oceanic factors.

b) Similar comments for Sections 4.2, 4.3, and 4.4.

Section 4.3: The authors proposed 4 situations that help/prevent NCU occurs.
      (1) Strong southward alongshore current prevents NCU.
      (2) Strong northward alongshore current weakens NCU.
      (3) Secondary dipole and the relating secondary offshore jet strengthens NCU.
      (4) Weaken dipole structure and offshore jet strengthen NCU.
These situations seem to conflict with each other and no further quantitative analyses are employed to prove their hypothesis.

c) Figure 4i-4l, The authors compared the basin-scale SCS circulation based on the sea level anomaly field, which only expresses the anomalous flow field. This is not proper for describing basin-wide circulation, because it should include both mean flow and anomalous flow. Besides, the authors claimed an eastward jet appears in the modelled and satellite-derived anomalous flow field (L239-243); however, I cannot see that!

3. **Conclusion:**
Factor like wind stress curl has not been carried out in the analyses but still appear in the conclusion.

**Minor Comments:**

L18 "mesoscale ocean dynamics" should be more concise or direct to the point.

L63 "influences"

L73 "varies"

L158 The mean bar notation should be put overline

L197 "The fourth area"

L226 For accuracy, comparison between the spatial-mean simulated and observed SST, SSS and SLA could be done over a smaller area such as the VNU rather than the whole VNC domain.

L254-257 "Though SYMPHONIE is overall ... Woo et al. (2020)". Quantitative assessment of the overestimating of the surface cooling in the southern Vietnam coasts is missing, which is important for evaluating SST in the upwelling region. The reader is left wondering, the SST overestimating is caused

by SYMPHONIE output or OSTIA? It further raises the question that if upwelling occurs in BoxMK in reality.

L233 L245, L261, L270, L275, L280 Inconsistencies in describing NRMSE, sometimes use "%", sometimes use decimal.

L235, globally?

L292 - L299 Long description

L307 Figure SM1 is not found in the manuscript.

L341 "... the lowest of the 4 boxes...". The lowest of what?

L391 Definition of "OIV" haven't been mentioned.

L357-L358 Values of UI need to be checked again.

L356-L365 The authors compare the differences between 2009 and 2012 of the daily/monthly wind stress and daily upwelling index and conclude that the daily to intra-seasonal variability of wind forcing modulate the SCU interannual variability. However, this analysis does not make sense to me because they are the different time scales. Similar comments for the OU and NCU.

L372, L373 "is (not) related to" should be " (does not) relate(s) to"

L393 Vorticity calculation has not been described. What kind of vorticity? How do the authors define the surface current? Which depth layer of velocity do they use for the calculating?

L503-509 "This current constitutes …the stable position of MKU". This inference needs more evidences.

**Figures:**

Figure1: VNC configuration should be described in detail, especially the coastal region, rather than locations of 4 upwelling areas, which are displayed again in Figure 2c.

Figures 3 and 4: There is an inconsistency that exists between the order of figures and the text's description, which makes the reader hard to follow.

Figure 5: "... ARGO (a, black dots and purple for mean), GLIDER (b, black dots and cyan for mean), IO-18 (c, black dots and green for mean) observations and from SYMPHONIE (yellow dots and red for mean) colocalized outputs..." should be "... ARGO (a, black dots and purple line), GLIDER (b, black dots and cyan line), IO-18 (c, black dots and green line) observations and from SYMPHONIE (yellow dots and red line) colocalized outputs...". However, the caption should be better clarified.

Figure 7: Legendaries of x and y axes are overlaid (year 2018)

Figure 8: Purple contours in a,b,c,d,i,j,k,l,q,r and blue contours in e,f,g,h,m,n,o,p,s,t have no explanation.

Figure 9 and Figure 10: The ranges of y-axis should be fixed with the presented data

---

## Referee Comment (RC2)

**Comments on the paper entitled "Role of wind, mesoscale dynamics and coastal circulation in the interannual variability of South Vietnam Upwelling, South China Sea. Answers from a high resolution ocean model" by To Duy, Herrmann, Estournel, Marsalex, Duhaut, submitted to Ocean Science (os-2021-121)**

This paper investigates the interannual variability of the South Vietnam upwelling by using a modeling approach. The high resolution coastal circulation model is extensively validated by comparison with the data from different sources which makes the results convincing. The modeling approach seems appropriately designed for studying the upwelling events and their variability in a wide range of time scales: from daily to interannual. By considering high-frequency variations of the wind stress at the regional scale the authors have clearly demonstrated that the magnitude of the wind variability at scales of days to weeks can partially explain large differences in upwelling intensity observed during years with rather similar mean wind forcing. This is the first valuable result of the study. The second issue addressed is how some specific features of the regional circulation can impact the interannual variations of upwelling by modifying the Ekman transport, precisely by adding a not wind-driven component to the total current velocity. It was shown that the surface currents act differently in four considered sub-regions of a vast upwelling system of the South Vietnam. The background current can weaken of reinforce the upwelling intensity thus affecting the interannual variability.

The authors furnished an effort in analyses of modeling results, the data form observations, and they overall made up nice figures. I believe that conclusions of their work are interesting and could contribute to the general knowledge on scales of variability of the upwelling circulation in this part of the ocean and in other ocean regions.

I am convinced that this paper is worth publication after some major revisions. I provide below a list of the most important comments.

**Major comments**

**Abstract**

The text after line 30 should be rewritten in order to demonstrate the forceful results. In the present version, I don't feel the major findings are presented in appropriate way. There a lot of generalities without precision and quantification. For example, it is difficult to understand what the authors mean by " … the impact of the … temporal organization of mesoscale ocean structures and atmospheric forcing". What is the message addressed in the last two lines: "… an interannual variability of upwelling (in Mekong box) is mostly determined by the summer wind and summer driven circulation in the region". I agree, but what novelty is behind this statement? I would suggest to avoid this kind of sentences and make the presentation of the results sharper, more incisive.

**Introduction**

The scope of the study needs a more clear definition (ln.94-100). This research didn't start from zero. The role of the background current variability in the interannual variability of upwelling was already highlighted by Da et al. (2019). What was discovered before and what is focused in particular in the present study should be better introduced. This concerns the "processes", which were not clearly defined, and "scales" which are targeted.

Ln 99: The text should be reworded with respect to my previous comment. "The objective is … scientific" should be removed.

Perhaps a clear definition of the numerical tool should be provided here. If it is different from the numerical model, it should be specified.

**Section 2**

The numerical model is briefly presented in this section. I think some terms require clarification. The first concerns "the biharmonic viscosity of momentum" and the second concerns "nudging". I assume the authors mean how the tidal motions were prescribed at the open boundaries. But the word tidal is missed in the text.

Ln 126: the authors use the term "zoom on the VN coast". I don't have impression that the technique of zoom was implemented in the model. This needs clarification.

Presentation of the data sources. Sometimes, the information provided is absolutely useless: for example the program code, the name of PI. On the contrary, some acronyms need clear definition such as IO.

The term "hydrological characteristics of water masses" is misleading. The temperature and salinity are used for water masse characterization. What role the hydrology plays (the freshwater input, as I understand the term) in modifying T,S characteristics is unclear.

My major concern is about the definition of the upwelling indicators and the choice of the reference box which area is much smaller than that of the corresponding upwelling box. The authors should justify their choice of the reference box size and the temperature values. When the size is small and the reference location is close to the upwelling region, the reference temperature is obviously dependent on the temperature observed during the upwelling event. To what degree this quantity is independent? This needs clarification as the results could be sensitive to the choice of the reference value. What could be the difference if an overall mean temperature (space and time mean) is used as the reference value?

**Presentation of the results**

**Section 3**

As I indicated above, the authors furnished an effort in analyses of the data from different sources and in validation of the modeling results. The results are convincing. However only spatial distribution of different quantities at the surface is used in comparison. But the model is three-dimensional and high resolution. A demonstration of the model capability in reconstructing the upwelling circulation (and related water properties) in the vertical plan can be an added value. This can support the choice of high-resolution in the horizontal and also in the vertical.

The authors often use the term "coastal scale". The word coastal is not appropriate for scale definition. The scale needs clarification.

**Section 4**

In this section the authors explore in detail the upwelling variability for each year. Nine years in total are considered with strong and weak upwelling events. The effect of the mean wind in interannual variability of upwelling was verified but this is not a novel result. Mechanisms which can explain large difference in upwelling intensity between years with a similar mean wind are identified.

I have two major points of criticism regarding this part of the study.

The first point concerns the interpretation of the variability of the wind stress at high time resolution which is presented as the "other factor modulating the wind induced interannual variability of SCU" (ln 354).

The idea behind seems clear, but what is less clear how to quantify and interpret the effect of high-frequency variability. A method or a metric should be used in this demonstration. A visual inspection of the wind stress curves given in Fig. 9,10 is not sufficient. The authors use CV, the coefficient of variation. But how it helps in quantification of the contribution of high frequency compared to low frequency variability? This needs clarification.

I see a small inconsistency in the interpretation of the "other factor modulating the wind induced variability". First, in this particular case, the high and low frequency variability of upwelling is wind-induced. The physical process involved in upwelling generation is the same. Second, I cannot imagine how the high frequency can modulate the low frequency signal. Different averaging techniques can provide different values of the mean quantity. But this is not sought as modulation. The authors should find a better formulation.

The second point of criticism concerns the role of the wind stress and surface currents in upwelling variability in different years. From my point of you, the surface current velocity and the current velocity curl are tightly related to the wind stress curl (example of anti-symmetric eddies and the eastern current). I have impression that only the wind stress magnitude is used in analysis. The added value of the study will increase if analysis of the wind curl and perhaps the wind stress vector field can be introduced and comparison with current velocities be made. This will help in interpretation of the surface current variability. What part of the current variability is wind-induced? and what part is remotely induced? The choice of the method of quantification is an important issue. And this is related to the statement (used for the second time) "another factor of non-wind origin" controlling the variability of upwelling. A part of the background current variability, independent of the wind, should be clearly identified and characterized. This requires a method of identification. I didn't see a clear description of such a method in the manuscript.

**Discussion and conclusions**

Section 4.5 and section 5 (Conclusion) should be reorganized. Section 4.5 contains the discussion of the modeling results and should be entitled "Discussion". A part of section 5 also contains the discussion and should be relegated to "Discussion".

Conclusion section should contain the major and novel results in a condense form. The comparison with previous studies is already done (in principal) in Discussion section. The form is important. Please avoid sentences of five lines difficult to follow. Highlight what knew knowledge the study brought and in what it is different compared to the results of previous studies.

**Technical corrections**

Ln 30: move "driving" to different location … our results confirm the role of the … in driving the interannual …

Ln 34: perhaps "structures of circulation" ?

I would recommend replacing the word ability, when you talk about the model skill, by capability, in the whole text.

Ln. 103-105: When describing the paper structure, it is better to use the word Section, not Part.

Ln 45-46: Please reword the text concerning the CSS contribution.

Ln 91 ageostrophic dynamics

Ln 151-152 … level 3 SSS derived from SMOS … (MIRAS) measurements at 0.25° resolution

Ln163-164: The modeled outputs were spatially and temporally co-localized with observations and used for comparison.

Ln 165: put a dot after "area". The data are available …

Ln 201: I think there is a conventional way how to refer to a Figure in another publication: (cf. Fig. 1, Da et al., 2019)

Ln 206-2020: The text should be edited to make it clear. Frequency should be removed. I suggest : … from the analysis of the occurrence of …

Please choose the right order in index definition: UI should match upwelling index

Perhaps is it more simple to use a number 122 (days in four months) instead of NDjjas ?

Ln 224: the title: Surface circulation, temperature and salinity in the SCS. Perhaps it is better than hydrological characteristics. The text in four lines following the title should be rewritten. I don't understand "interannual yearly averages". Do you mean" monthly mean and yearly mean values" ? Please check the sentence structure and articles.

Ln 221: Title : cycle ("c") an variability of what?

Ln 237: remove "the" before Introduction and put "the" before northern monsoon wind.

Ln 254: in very good overall agreement

Ln 285: Choose a better title.

p9 "coastal scale" is used in some places but the scale is not defined.

Ln 323-325: This text and the text in ln 326-330 is repetition. Lines can be removed.

Ln 328: 10-year long simulation.

Ln 329: prefer " in four regions"  to 4 regions

Ln 350: UI=1.49. I read 2.0 in Fig. 7.

Ln 374-375: the next needs rewording. If possible, provide the exact location of each of four eddies in Fig 8. Put a symbol for example, or provide coordinates in the text.

Ln 390: perhaps the word "chaotic" is not appropriate if the structures are visible in the mean field. Do you mean large scale turbulent structures? If not, clarify the meaning.

Ln 411-415. Perhaps reword the text. Too long and difficult to follow.

Ln 532: these differences

Ln 541-542: the SCS. The text should be rewritten.

571: zonal location is preferable to position

---

## Author Response (AR1)

To :
Katsuro Katsumata, Editor of Ocean Science

From :

Marine Herrmann, To Duy Thai, Claude Estournel, Patrick Marsaleix, Thomas Duhaut
LEGOS, Toulouse, France

Bui Hong Long
Institute of Oceanography, Nha Trang, Vietnam

Trinh Bich Ngoc
USTH, Hanoi, Vietnam

Toulouse, April 28th, 2022

Dear Editor,

Please find a revised version of our paper entitled "Role of wind, mesoscale dynamics and coastal circulation in the interannual variability of South Vietnam Upwelling, South China Sea. Answers from a high resolution ocean model".

We warmly thank the reviewers for their careful reading and evaluation of our manuscript and for their constructive comments. All those comments were carefully addressed in the revised version of the manuscript. In particular, we worked on:

- the presentation of results: previous studies were more precisely described in the Introduction, and we clearly stated in Section 4 and in the Conclusion which results where confirmation of previous results, or new results. The objectives our work were more precisely described in the Introduction.

- the organization of Discussion and Conclusion : a new Section 5 was devoted to discussions and developed, including things that were previously in the Conclusion, but also points raised by the reviewers (role of wind curl, role of chaotic variability, impact of the choice of reference temperature …)

- a better highlight of the added-value of our study, related mainly to the high-resolution of our model that allows to better simulate, capture and study the small spatiotemporal scales of temperature, currents and upwelling, and their impact on averaged seasonal intensity of upwelling. In particular, an analysis of in-situ and satellite data revealed that upwelling really develops offshore the Mekong Mouth (BoxMK). Limitations of gridded products do not allow them to capture it correctly and high-resolution model is a valuable tool to capture and study it.

Best regards

Marine Herrmann, coordinating author

**Comments on " Role of wind, mesoscale dynamics and coastal circulation in the interannual variability of South Vietnam Upwelling, South China Sea. Answers from a high resolution" by Tai To Duy et al. in Ocean Science**

In this study, the authors attempted to explore the dynamics governing the interannual variability of South Vietnam Upwelling based on model simulation. Befere discussing the modelling results, they first compared the general patterns of simulated SST, SSS, and SLA with the satellite-remosed data, and compared the simulated temperature and salinity profiles with the in-situ meastsurements of ARGO, Seaglider, and R/V cruise. After reviewing, I think the manuscript needs an subsustainsial revision before being acceptable based on the following comments.

We warmly thank the reviewer for this careful and constructive review of our paper. We addressed all the comments below in our revised version of the manuscript. In this document, the reviewer's comments appear in black, and our answers in green. Changes done in the manuscript following the comment of the reviewer are also highlighted in green in the revised version of the manuscript. Line numbers and pages in this document refer to the highlighted version.

Major Comments:

1. Methodology:

a) The authors explored the South Vietnam Upwelling in four areas: BoxOF, BoxNC, BoxSC, and BoxMK. For the first three areas, the cold temperature are evident in both the simulated SST and satellite SST. However, for the BoxMK, the cold temperature seems only appear in simulated SST, but not in satellite SST, although the authors have referred to the finding of two literatures. This argument does not make sense, because a reader cannot directly confirm the rationality of considering BoxMK a spereate area to be explored. If this is not a natural phenomenon, the discussion becomes meaningless.

Indeed, though upwelling in the southern (BoxSC) and northern (BoxNC) coastal areas and in the offshore area (BoxOF) was already mentioned and studied in previous studies (see for example *Da et al. 2019, Ngo and Hsin 2021*), no mention of an upwelling appearing offshore the Mekong Delta (BoxMK) was previously mentioned. A careful examination below of analysis products, satellite data and available in-situ data over the area shows that this can be explained by the scarce satellite and in-situ cover of upwelling that occurs over this small area and the smoothing associated with analysis construction.

OSTIA level 4 analysis SST data are indeed built combining complementary satellite and in situ observations within Optimal Interpolation systems on a global 0.054 degree grid (*UK Met Office. 2005*). This Optimal Interpolation induces spatial and temporal smoothing of extreme values. Similarly, the COPERNICUS PSY4QV3R1 analysis is produced using a 1/12° ocean model that assimilates available satellite and in-situ data, and does not include tide.

The satellite and in-situ coverage of the SVU area in summer is not very good, due to the high cloud cover. It is all the more the case of the small BoxMK area. We can examine this availability in JAXA satellite data : we use the daily SST (Level 3) dataset provided from Himawari Standard Data by JAXA over the period 2015 - present with a 2 km spatial resolution. For summer 2018, a summer of very strong upwelling, we show the percentage of days during which JAXA data are available in Figure A below. This percentage never exceeds ~85% over the SCS, it is about 75-80% over the offshore VNU region, and it is lower than 60% offshore the Mekong mouth in the area were MKU occurs (and over the Gulf of Tonkin and Gulf of Thailand). For those reasons, the surface cooling over BoxMK may not be correctly captured in analysis data, as we will see below.

[Figure]

**Figure A : percentage of days during which JAXA data are available with a quality level equal or higher than 4 during summer 2018.**

Examining directly in-situ and raw satellite data that cover the BoxMK region during the period of upwelling shows that surface cooling indeed occurs over BoxMK at the same time and places as simulated by our model:

→ **Summer 2014 :** RV Alis crossed BoxMK during 25-26 June 2014. Figure B(left) below shows a zoom of ALIS SST data offshore the Mekong mouth, with a very good agreement between simulated and observed data. In particular, a surface cooling over BoxMK was clearly observed by ALIS-TSG, and simulated by the model, with values of minimum SST ~28.2°C near Con Dao Island (~106.6°E-8.6°N) on 25/06/2014. In-situ data therefore confirm the reality of MKU simulated in SYMPHONIE. Figure B(right) shows the maps of SST from SYMPHONIE, OSTIA and COPERNICUS on 25/06/2014. OSTIA and COPERNICUS actually show a significant surface cooling in the same area as SYMPHONIE, with SST < Tref=29.2°C. However, as explained above, smoothing and data availability result in weaker surface cooling in those analysis products, with minimum SST not going below ~28°C, vs. ~27.5°C in SYMPHONIE.

Figure C,a below shows the daily time series of minimum SST over BoxMK for SYMPHONIE, OSTIA and COPERNICUS during summer 2014. The 3 times series show similar temporal variations with a minimum end of June – beginning of July, however, minimum SST values over BoxMK are ~1.5°C lower in SYMPHONIE than in OSTIA and COPERNICUS during this period.

→ **Summer 2018 :** in the 2009-2018 simulation, 2018 is the year with the strongest upwelling over BoxMK (Figure 13 of the revised manuscript). Figure C,b below shows the daily time series of minimum SST over BoxMK for SYMPHONIE, OSTIA, COPERNICUS and JAXA satellite data during summer 2018. Again, time series are following similar variations in OSTIA, COPERNICUS and SYMPHONIE, but with much lower values (by ~1.5°C) in SYMPHONIE. Moreover, values simulated in SYMPHONIE are very close to values observed from JAXA satellite data, with peaks at the same period: minimum values of SST are obtained first during mid-June (~26.6°C), mid-July (~25.6°C) and mid-August (~26.2°C).

Figure D shows the SST maps during those 3 upwelling peaks (June 20[th], July 16[th,] August 14[th]) for JAXA, OSTIA, COPERNICUS, and SYMPHONIE. A surface cooling over BoxMK is clearly visible in JAXA and SYMPHONIE during those MKU peak periods, with SST < 28°C. Again, as observed for 2014, a surface cooling is also produced by OSTIA and COPERNICUS analysis, but with warmer values. OSTIA also produces a strong surface cooling during mid-July with minimum SST reaching ~27.5°C.

Last, surface cooling is stronger in JAXA and SYMHONIE, but, both 2014 and 2018, the area of cooling over BoxMK is very similar in SYMPHONIE and in analysis data (OSTIA and COPERNICUS).

Those results show that upwelling really occurs over BoxMK in summer. It is captured by ALIS TSG in-situ data and JAXA satellite data, and simulated accordingly by SYMPHONIE. They moreover confirm that the corresponding surface cooling is also captured by OSTIA and COPERNICUS, but is strongly smoothed. These results therefore highlight the added-value of a high resolution (~1 km at the coast) resolution model to simulated and study the upwelling in the SVU area, able to simulate better the spatial and temporal fine scale structures of SST compared to coarser resolution gridded satellite data.

→ Following this comment, we added a whole section (**Section 3.1 Upwelling over BoxMK, page 9**) and **Figures 4, 5, 6** to discuss this point this in our revised version of the manuscript

[Figure]

**Figure B : Left : Simulated (SYMPHONIE) and observed (ALIS-TSG) SST (°C) during ALIS R/V trajectory offshore the Mekong mouth in June 2014. The arrow shows the location of minimum SST~28.2°C in both data and model, recorded near Con Dao Island (~8.6°E – 106.6°E) on 25/06/2014. Right : SST in June over BoxMK on 25/06/2015 (°C) from SYMPHONIE, OSTIA and COPERNICUS. The pink contour shows the isotherm Tref (=29.2°C, see answer to comment about Tref below).**

[Figure]

**Figure C : Daily time series of minimum SST (°C) over BoxMK in (a) SYMPHONIE, OSTIA, COPERNICUS during summer 2014, and in (b) SYMPHONIE, OSTIA, COPERNICUS and JAXA during summer 2018.**

[Figure]

**Figure D : Daily SST (°C) on 20/06, 16/07 and 14/08/2014 from JAXA, OSTIA, COPERNICUS and SYMPHONIE. The pink contour shows the isotherm Tref (=29.2°C).**

b) The authors performed the simulation of the model SYMPHONIE from 2009-2018 by comparing output data (SST, SSS, SLA, and T-S profiles) with high-resolution satellite data and in-situ observations, showing that this model is an innovative tool that can reproduce oceanic dynamics properly not only at the surface but also at deeper sub-layers, and at wide-range time scales. To investigate the daily-to-interannual variability of the VNU, however, they employed only the surface data (SST and velocity) and the discussions are all statistically, which brings not many new results in the comparison with previous studies using satellite data. In other words, this study can be performed by the satellite data without SYMPHONIE. I think the authors should utilize the advantage of modelling to conduct numerical experiments to examine whether the proposed factors are really factors controlling the interannual variablility of South Vietnam Upwelling in each area.

Indeed, the goal of our work is 1 ) to develop a high resolution model implemented over the SVU for the study of the upwelling dynamics and variability and to evaluate and show its quality by extensively comparing it with available in-situ and satellite observation ; 2) to investigate the interannual variability of the SVU, confirming results from previous studies made at lower resolution, and going further by investigating 4 different areas at very high resolution ; and 3) to perform sensitivity experiments to further explore the different physical processes and scales of variability involved in the SVU. For this 3rd step, we already conducted several numerical experiments, including ensemble simulations, to investigate the contribution of different factors to the daily to yearly variability of the upwelling: wind, tides, rivers, ocean intrinsic variability (*Da et al. 2019, Li et al. 2014* indeed suggested the importance of those factors). The results of those experiments were presented in the PhD manuscript of To Duy Thai (*To, 2022*). In the present paper, we presented steps 1) and 2). For the sake of conciseness and clarity, we did not present results from step 3) : for that, a second paper is in preparation and will be submitted soon to the same journal. The reviewer writes that "they employed only the surface data (SST and velocity) and the discussions are all statistically, which brings not many new results in the comparison with previous studies using satellite data. In other words, this study can be performed by the satellite data without SYMPHONIE". Our study however contains several elements which make it, in our opinion, a valuable contribution to the understanding of SVU interannual variability. To our knowledge, the most recent and complete studies up to date about the interannual variability of the SVU were done by *Da et al. 2019* and *Ngo and Hsin 2021*. Here are the new elements provided by our study:

- In terms of methodology: we use a very high-resolution numerical model (1 km at the coast to 4 km offshore, and including tides). *Da et al. 2019* used a 14-years 1/12°(~9 km) resolution simulation, not including the effect of tides. *Ngo and Hsin 2021* used ¼° (~28 km) resolution datasets of 38-years SST and reanalysis winds and 28-years satellite-altimeter derived sea surface current. We showed above that the resolution and smoothing associated with analysis data prevent them from fully capturing the signature of upwelling, especially this related to small scales. Our simulation allows to capture, and study, much more accurately the spatial and temporal variations of SST, currents and upwelling than those previous studies.

- In terms of knowledge of the area of development of the SVU over BoxMK (see answer to comment above): our model revealed that upwelling also occurs offshore the Mekong mouth (BoxMK). A careful examination of satellite data and analysis products confirmed that this upwelling over BoxMK is real, but that analysis data can not capture well this upwelling (see answer to the specific comment about MKU above). These results therefore highlight the added-value of a high-resolution model to represent and study the upwelling in the SVU area, able to simulate better the spatial and temporal fine scale structures of SST compared to coarser resolution gridded satellite data.

- In terms of knowledge of the functioning of upwelling over BoxNC : our model revealed that NCU is not primarily driven by the intensity of the summer wind over the SVU of BoxNC region, but rather by the submesoscale scale dynamics that develop over BoxNC (see answer to comment below about BoxNC). This conclusion is a new finding that was obtained thanks to the high-resolution coverage of this small coastal area by our model. Role of submesoscale dynamics in the development and daily to yearly variability of MKU will be examined in further details in the paper in preparation mentioned above.

→ following this comment, we highlighted in the text in more details the added-value of our study regarding those aspects : high-resolution methodology, better representation of fine scale structures of SST and currents over the coastal areas, in particular BoxMK and BoxNC, future studies in preparation to examine more precisely the contribution of different factors. In particular, the Introduction was rewritten in order to better present the existing knowledge about the SVU and its limitations (**lines 70-107**), and the goal of our work regarding the study of contribution of small spatial and temporal scales in the SVU interannual variability (**lines 109-129**). Those aspects were also underlined in the Discussion (**lines 694-715**) and in the Conclusion (**lines 759-756, 774-780, 784-788**).

We also agree that the 3D coverage model can be used to better understand the upwelling, and as explained above we will present in a coming paper results from sensitivity simulations. An important question is in particular

whether the surface cooling observed in BoxOF is really the result of an upwelling developing over BoxOF, or of the advection from BoxSC by the eastward jet of cold surface water upwelled at the coast. To answer this question,

Figure F shows the maps of simulated vertical velocity at 20 m, daily upwelling index and surface currents averaged over the two periods of upwelling development in summer 2018 (summer of strongest OFU, Figure 13 of the revised manuscript) : the first two weeks of July 2018 and the first week of August 2018 (Figure 15). Those figures qualitatively show that BoxOF surface cooling indeed partly results from the advection of cold water from BoxSC, in particular in July. However, strong upward (positive) vertical velocities are simulated, not only along the coast and over BoxMK, but also over BoxOF, in particular in August. This confirms that a significant part of surface cooling over BoxOF results from a local upwelling. Further dedicated studies, including box analysis following the method used for dense water formation by Herrmann et al. (2008), are now required : they will help to quantitatively assess the respective contribution of lateral advection, surface forcing, vertical advection and internal mixing to the formation of cold water over BoxOF, and their temporal and spatial variability, in the formation of cold surface water over BoxOF. This kind of analysis is out of the scope of the present paper, but will be developed at the daily scale and presented in a next paper.

[Figure]

**Figure E : Vertical velocity (m.s⁻¹, 1ˢᵗ row) and daily upwelling index (°C, 2ⁿᵈ row) and surface currents (m.s⁻¹) averaged over the July (a,c) and August (b,d) periods of upwelling development in summer 2018.**

→ following this comment, we added a new section in the discussion (**Section 5.5. Surface cooling over BoxOK: offshore upwelling vs. lateral advection of cold water**) and **Figure 17**.

c) In Section 2.3, the author introduced several SST-based upwelling indicators (daily, yearly and spatial upwelling index), which are applied for 4 upwelling areas. Each area uses different reference boxes, which is taken as the areas not impacted by surface cooling. However, the boxes (besides RefOF) they chose may be highly possible to be influenced by other upwelling areas. For example, RefNC could be impacted by the offshore upwelling if the offshore upwelling have more northern extension. In addition, the authors use the time-averaged Tref in each Reference box, but the temperature in the SCS suffers interannual variations, e.g., Figure 3b. This could make a great impact on the calculation of SST-UI, and result in a large dependency as discussed in section 4.5.

d) Another concern is that the spatial upwelling index could be not a continuous field as shown in Figure 8 because the authors use different Trefs.

We answer to comment c) and d) together.

First, we completely agree that the SST over the reference boxes chosen for BoxMK, BoxSC and BoxNC may be influenced by the propagation into the reference box of water upwelled in other areas. This is all the more the case when the reference box is small. Moreover, using different Tref makes the upwelling index field spatially discontinuous. We therefore recomputed everything with the same Tref for all the boxes, taking Tref for BoxOF, which can be considered as large and far enough from the upwelling areas. We obtain a value of Tref=29.20°C. Using different boxes, our former values were Tref$_{NC}$=29.66°C, Tref$_{SC}$ = Tref$_{MK}$= Tref$_{OFF}$ 29.20 °C. Except for BoxNC, using different references boxes actually was thus equivalent as using a unique reference box. Note also that even for BoxNC, the difference of upwelling index value induced by the new choice of Tref will be very small. Given the formulae used for UId and UId (Equations 1 and 2 of the revised manuscript, see below), the relative difference between UIy and UId computed using the new and the old Tref will be equal to (Tref,new - Tref,old)/Tref,old = (29.2-29.7)/29.7 = -0.017 ~ -2%).

$$UI_{d,boxN}(t) = \frac{\iint_{(x,y) \text{ in boxN so that } SST(x,y,t)<T_O}(T_{refN}-SST(x,y,t)).dx.dy}{A_{boxN}}(1) \quad \text{and} \quad UI_{y,boxN} = \frac{\int_{JJAS} UI_{d,boxN}(t)dt}{ND_{JJAS}}(2)$$

Table A below shows the updated values for table 2 of the manuscript. Logically (since Tref only changes for BoxNC, by less than 2%), UIy (mean, std, CV) values do not or only very slightly change. Moreover, most of the correlation values remained unchanged, and those which changed only slightly changed by less than 0.03 (see table A). Our conclusions concerning the relationships between the different factors were robust to this choice of reference temperature.

**Table A : Modified Table 2 with a constant and unique Tref=29.2° (values that were modified compared to the previous manuscript are highlighted in red and italics). From 1st to last line : temporal mean and standard deviation of UIy,boxN over 2009-2018 for each box and coefficient of variation CV (which is the ratio between STD and mean), correlations (correlation coefficient and associate p-values) between time series of significant factors : yearly upwelling index over each box vs. yearly upwelling index over other boxes, vs. average wind stress averaged over June-September (JJAS) and July-August (JA) over each box, vs. integrated positive vorticity over BoxOF. Correlations significant at more than 99% (p<0.01) are highlighted in bold.**

| | BoxNC | BoxSC | BoxOF | BoxMK |
|---|---|---|---|---|
| UI$_{y,boxN}$ mean (°C) | 0.163 *vs. 0.195 (0.14 in the previous manuscript was a typo)* | 0.798 *(0.42 in the previous manuscript was a typo)* | 0.074 *(0.09 in the previous manuscript was a typo)* | 0.065 |
| UI$_{y,boxN}$ STD (°C) | 0.118 *vs. 0.14* | 0.423 | 0.093 | 0.055 |
| CV (%) | 72 *vs.71* | 53 | 126 | 85 |
| Correlation between | UI$_{y,NC}$ | UI$_{y,SC}$ | UI$_{y,OF}$ | UI$_{y,MK}$ |
| UI$_{y,SC}$ (°C) | 0.00(0.99) *vs. +0.01(0.98)* | 1 | **+0.73(0.02)** | **+0.83(0.00)** |

| | | | | |
|---|---|---|---|---|
| $UI_{y,OF}$ (°C) | -0.26(0.47) | **+0.73(0.02)** | 1 | **+0.92(0.00)** |
| $UI_{y,MK}$ (°C) | -0.19(0.59) | **+0.83(0.00)** | **+0.92(0.00)** | 1 |
| $WS_{JJAS,NC}$ (N.m$^{-3}$) | -0.09(0.78) *vs. -0.10(0.77)* | -0.41(0.24) | +0.23(0.53) | +0.07(0.85) |
| $WS_{JJAS,SC}$ (N.m$^{-3}$) | -0.13(0.72) | +0.85(0.00) | +0.76(0.01) | +0.83(0.00) *vs. +0.84 (0.99)* |
| $WS_{JJAS,OF}$ (N.m$^{-3}$) | -0.19(0.60) | +0.81(0.00) | +0.77(0.01) | +0.80(0.01) |
| $WS_{JJAS,MK}$ (N.m$^{-3}$) | -0.08(0.81) | +0.78(0.01) | +0.63(0.05) | +0.72(0.02) |
| $WS_{JA,NC}$ (N.m$^{-3}$) | +0.04(0.90) *vs. +0.07(0.92)* | +0.18(0.62) *vs. +0.12(0.62)* | +0.54(0.11) | +0.38(0.28) |
| $WS_{JA,SC}$ (N.m$^{-3}$) | -0.15(0.69) | **+0.70(0.03)** | **+0.84(0.00)** | **+0.84(0.00)** |
| $W_{JA,OF}$ (N.m$^{-3}$) | -0.15(0.67) | **+0.69(0.03)** | **+0.84(0.00)** | **+0.82(0.00)** |
| $W_{JA,MK}$ (N.m$^{-3}$) | -0.11(0.77) | **+0.72(0.02)** | **+0.78(0.01)** | **+0.82(0.00)** |
| $\zeta_{+,OF}$ (s-1) | -0.28(0.43) | **+0.60(0.07)** | **+0.69(0.03)** | **+0.74(0.01)** |

Second, we chose a constant Tref, as done previously by *Da et al. (2019)*, whereas other studies, like *Ngo and Hsin (2021)*, used an interannually varying Tref, to take into account the fact that SST can also vary on an interannual basis. We completely agree that SST varies interannually. We justify our choice of a constant Tref by the fact that even if the reference box was chosen outside the upwelling area, the SST in this box can be influenced by the eastward advection of water upwelled in the other boxes. This can be seen on Figure F that shows the JJAS map of simulated SST for summers 2010 (year of weakest upwelling) and 2018 (year of strongest upwelling). In 2018, the SST in the reference area is cooler than in 2010 due to the eastward advection of upwelled water. The upwelling index computed from the difference $SST(x,y,t) - SSTref$, would therefore be smaller in 2018 and larger in 2010 using a varying Tref vs. a constant Tref. In other words, a varying Tref would increase the weak values and decrease the strong values, hence reducing the interannual variability. Note however that the impact on weak values is actually limited by the use of the threshold temperature $T_0$. We investigated the influence of this choice on our results. Figure G shows the time series of Tref computed annually instead of a constant Tref, and the resulting yearly time series of UIy for each box. Table B provides the modified values of Table A. The resulting UIy values slightly vary, in particular for stronger values (see year 2018), but that the change is not significant. The interannual variability remains nearly the same, though it slightly decreases, as expected. Correlation coefficients consequently also slightly decrease (by at most ~0.10). However, correlations that were statistically significant (or not), remain statistically significant (or not). Our conclusions are therefore still valid.

[Figure]

**Figure F : JJAS average SST in the 2009-2018 simulation in 2010 and 2018 (°C).**

**Table B : Modified Table A with a varying Tref (values that were modified compared to Table A are highlighted in red) :**

| | BoxNC | BoxSC | BoxOF | BoxMK |
|---|---|---|---|---|
| $UI_{y,boxN}$ mean (ºC) | 0.171 | 0.743 | 0.060 | 0.055 |
| $UI_{y,boxN}$ STD (ºC) | 0.140 | 0.344 | 0.065 | 0.040 |
| CV (%) | 82 | 46 | 108 | 73 |
| Mean of $\zeta_{+,OF}$ (s$^{-1}$) | 1.95x10$^{-6}$ | | | |
| Correlation between : | $UI_{y,NC}$ | $UI_{y,SC}$ | $UI_{y,OF}$ | $UI_{y,MK}$ |
| $UI_{y,SC}$ (ºC) | +0.11(0.78) | 1 | +0.63(0.05) | +0.73(0.02) |
| $UI_{y,OF}$ (ºC) | -0.28(0.42) | +0.63(0.05) | 1 | +0.81(0.00) |
| $UI_{y,MK}$ (ºC) | -0.09(0.80) | +0.73(0.02) | +0.81(0.00) | 1 |
| $WS_{JJAS,NC}$ (N.m$^{-3}$) | -0.01(0.97) | -0.43(0.21) | +0.21(0.56) | +0.09(0.80) |
| $WS_{JJAS,SC}$ (N.m$^{-3}$) | -0.25(0.48) | +0.72(0.02) | +0.77(0.01) | +0.76(0.01) |
| $WS_{JJAS,OF}$ (N.m$^{-3}$) | -0.30(0.39) | +0.66(0.04) | **+0.77(0.01)** | +0.69(0.03) |
| $WS_{JJAS,MK}$ (N.m$^{-3}$) | -0.20(0.57) | +0.66(0.04) | **+0.63(0.05)** | +0.65(0.04) |
| $WS_{JA,NC}$ (N.m$^{-3}$) | +0.05(0.89) | +0.14(0.70) | +0.53(0.11) | +0.30(0.39) |
| $WS_{JA,SC}$ (N.m$^{-3}$) | -0.23(0.52) | +0.54(0.11) | +0.82(0.00) | +0.73(0.02) |
| $W_{JA,OF}$ (N.m$^{-3}$) | -0.23(0.52) | +0.53(0.11) | +0.82(0.00) | +0.70(0.03) |
| $W_{JA,MK}$ (N.m$^{-3}$) | -0.20(0.59) | +0.59(0.07) | +0.77(0.01) | +0.74(0.02) |
| $\zeta_{+,OF}$ (s$^{-1}$) | -0.37(0.30) | +0.43(0.22) | +0.68(0.03) | +0.64(0.05) |

[Figure]

**Figure G : (1st row )** yearly time series of an interannually varying summer averaged SST over the reference box (=varying Tref, full black line) and value of the climatological summer averaged SST over the reference box (=constant Tref, dashed line). Yearly time series of upwelling indexes UIy computing using a varying (red) vs. constant (blue) Tref, for BoxNC (2nd row), BoxSC (3rd row), BoxOF (4th row) and BoxMK (5th row).

→ Following this comment, **Figures 12, 13, 14, 15, Table 2 of the revised manuscript** were modified after using the same and constant Tref for all boxes. The definition of Tref was modified (**Lines 225-229 in Section 2.3)**. A new section (**Section 5.1, p19**) and a new figure (**Figure 16**) were moreover added to discuss the sensitivity of our results to a varying vs. constant Tref.

e) Some calculations have been done but not defined (i.e. wind stress, vorticity, coefficient of variation, …). Specifically, the authors adopted wind stress for many places, but they did not define the wind stress: meridional wind stress, zonal wind stress, along-shore wind stress or cross-shore wind stress.

Wind stress (taux,tauy) (zonal and meridional components) is computed from ECMWF wind velocity based on the bulk formula of Large and Yeager (2004)  : (taux,tau$_y$) = rhoa sqrt(Cd) (u10,v10)

where *(u10,v10)* is the wind velocity at 10 m height , *rhoa* is the air density computed from sea level pressure *SLP* and air temperature at 2m *T2m* : *rho = SLP / (287,058 T2m),* and *Cd* is the nonlinear drag coefficient computed from Large and Yeager (2004).

The horizontal wind stress curl *WS* is computed as the vertical component of wind stress rotational: $WS = dtau_y / dx - dtau_x / dy$.

Similarly surface current vorticity $\zeta$ is computed as the vertical component of surface current rotational: $\zeta = dv_{surf} / dx - du_{surf} / dy$.

→ Following this comment we added those definitions in the revised paper (**lines 157-163 in Section 2.1 and 413-416 in Section 4**)

CV is defined as the ratio between STD (row 2 of Table 2) and mean.

→This was explicitly written in the text (**line 418**), and added in **caption of Table 2 (2ⁿᵈ line)**

2. Result and discussion:

a) The authors wrote long paragraphs to describe known results and few lines for un-solid conclusions. For example:

Section 4.1: The impact of intra-seasonal and inter-annual variability of wind forcing on SCU has been revealed. I suggest the authors r**econstruct this section by referencing known results in the introduction, using several sentences to describe the similarity with previous outcomes and highlighting new finding they have discovered**. In the case of oceanic factors, word usage is not direct to point, for instant, "background coastal circulation" and "mesoscale structure". Qu**antitative assessment is missing for the oceanic factors**.

b) Similar comments for Sections 4.2, 4.3, and 4.4.

→ Following this comment, and the comment of the other reviewer, we developed in the Introduction the part about the existing knowledge about SVU (areas of development, role of wind, eddies and intrinsic ocean variability, **lines 70 to 107**). We also detailed the scope of the present study (**lines 114 to 129**) which fundamental objective is to better monitor, represent and understand the behavior of upwelling at smaller scales (meso to submesoscales), and over detailed areas (coastal to offshore) and the role of high frequencies (daily to intraseasonal). In **Sections 4.2 to 4.5**, we highlighted:

- what was a confirmation of previous results (those parts were strongly reduced, and previous results were described in the Introduction): **lines 452-461 for SCU, 498-509 for OFU,**
- what was a new result: **lines 463-491 for SCU, 511-535 for OFU, whole section 4.4 and lines 694-715 in section 5.4 for NCU, whole section 4.5 for MKU**.

Last, the **Conclusion** was almost completely rewritten and shorten to really highlight the new results of this study. We also avoided using terms as "background circulation" and "mesoscale structures".

Section 4.3: The authors proposed 4 situations that help/prevent NCU occurs.

(1) Strong southward alongshore current prevents NCU.

(2) Strong northward alongshore current weakens NCU.

(3) Secondary dipole and the relating secondary offshore jet strengthens NCU.

(4) Weaken dipole structure and offshore jet strengthen NCU.

These situations seem to conflict with each other and no further quantitative analyses are employed to prove their hypothesis.

This comment of the reviewer shows that the previous version of manuscript regarding NCU was not clear enough. We tried to explain this better here:

1) The NCU is inhibited when alongshore currents, either southward or northward, prevail over BoxMK. Southward alongshore currents prevail during summers of strong wind over the SVU region, when the dipole and

eastward jet, hence positive vorticity $\zeta_{+,OF}$, are highly marked, inducing strong OFU and SCU (see summers 2009, 2012 and 2018, Figures 11,12,13 of the revised document). These southward currents are associated with the western part of the northern cyclonic gyre and a divergent circulation, hence with a coastward component and a coastal downwelling which inhibits the NCU. Northward alongshore currents prevail during years of weak or average wind over the region (summers 2010, 2013, 2017, Figures 11,12,13). During those years, offshore circulation (AC/C dipole and eastward jet) is average (2017), weak (2013) or even absent (2010), resulting in weak average Ekman transport and pumping, hence in weak SCU and OFU. The weakness of the offshore circulation allows the development of an alongshore northward current all along the Vietnamese coast (Figure 12), which also inhibits the NCU. Southward or northward longshore currents over BoxNC therefore result from two opposite situations in terms of wind and offshore circulation, but induces both NCU inhibition.

2) The NCU is enhanced when offshore oriented circulation prevails over BoxMK. Offshore oriented circulation can result first from the development of a secondary dipole north of the usual dipole structure (see for summers 2011, 2014 and 2015 the alternation of negative and positive vorticity between 12°N and 16°N, Figure 12). This secondary dipole is associated with a second coastal area of convergence over BoxNC, hence a secondary eastward jet that induces the strong NCU. This situation is not related to the intensity of wind intensity or summer offshore circulation: it occurs both for summers of slightly stronger (2011 and 2014, Figure 13) or weaker than average (2015) wind and strong (2014, Figure 12) or weak (2011, 2015) offshore circulation. Offshore oriented circulation can also develop when a weaker but wider than average eastward jet prevails over a large part of the coastal region, including BoxNC (see summer 2016, Figure 12). This results in the offshore advection of cold water all along the coast hence in the development of a stronger than average NCU. NCU is therefore favored by the development of offshore oriented currents along the coast that result from a favorable spatial organization of submesoscale to mesoscale dynamics.

We therefore mainly show that the development of NCU is inhibited or favored depending on the circulation that prevails over BoxNC, independently of the large scale forcing wind and offshore circulation.

→ Following this comment, we completely rewrote Section 4.4 in the revised version of the manuscript **(lines 545-569).** We moreover commented more into details the differences between conclusions of Ngo and Hsin (2021) and our conclusions regarding NCU : those differences could be related partly to the lower resolution of satellite dataset used by *Ngo and Hsin (2021)* but also to the strong impact of OIV over BoxNC, related to the strong influence of submesoscale to mesoscale dynamics in the functioning of NCU **(Sections 5.3 and 5.4, lines 666-675 and 693-714).**

c) Figure 4i-4l, The authors compared the basin-scale SCS circulation based on the sea level anomaly field, which only expresses the anomalous flow field. This is not proper for describing basin-wide circulation, because it should include both mean flow and anomalous flow. Besides, the authors claimed an eastward jet appears in the modelled and satellite-derived anomalous flow field (L239-243); however, I cannot see that!

We computed and plotted the field of absolute sea surface height (SSH) and associated total geostrophic current for the model and for data (AVISO) in see figure G (1st row) below. The 2nd row (former Figure 4) shows their anomaly. The patterns of currents and of their anomalies are actually quite similar. On the total geostrophic current, the AC/C dipole and eastward jet is actually better visible, and highlighted by grey arrows.

→ **Following this comment, we showed SSH and associated total geostrophic, rather than their anomaly, in Figure 2k-l of the revised manuscript.**

[Figure]

**Figure G: Spatial distribution of simulated and observed winter (DJF) and summer (JJA) climatological averages of SSH (i,j,k,l, m) and total surface geostrophic current (m.s$^{-1}$) (1$^{st}$ line) and of their anomaly (2$^{nd}$ line), and spatial correlation coefficient R (here the p-value is always smaller than 0.01). Grey arrows on panels k,l highlight the summer AC/C dipole and eastward jet.**

3. Conclusion:

Factor like wind stress curl has not been carried out in the analyses but still appear in the conclusion.

→ Following the comment of the other reviewer, the link between wind stress curl and wind stress, and circulation, is not discussed in the new **Section 4.1 Interannual variability of wind and offshore summer circulation (p14)**

Minor Comments:

L18 "mesoscale ocean dynamics" should be more concise or direct to the point.

→ This was replaced by "mesoscale to regional ocean circulation." **(line 19)**

L63 "influences"

→ This was corrected, **line 64**

L73 "varies"

→ This was corrected, **line 83**

L158 The mean bar notation should be put overline

→ This was corrected, **line 282**

L197 "The fourth area"

→ This was corrected, **line 220**

L226 For accuracy, comparison between the spatial-mean simulated and observed SST, SSS and SLA could be done over a smaller area such as the VNU rather than the whole VNC domain.

→ Indeed, our study focuses on the SVU. Following this comment, we added **in Figure 7** of the revised manuscript the times series of the same variables but averaged over the SVU domain (104-116°E; 7-16°N). **Figure 7 of the revised manuscript** shows that (except for SLA for which seasonal variations over the smaller SVU domain are weak) both VNC and SVU domains show very similar behaviors in terms of seasonal cycles and interannual variations of SST, SSS and SLA, and performances are very slightly better on the SVU region. We commented the updated figure in parts 3.2.1 and 3.2.2 of the revised manuscript (see changes highlighted in green at the beginning of each paragraph, **lines 288-291, 310-313321-323, 336, 343-344, 349-350**).

L254-257 "Though SYMPHONIE is overall ... Woo et al. (2020)". Quantitative assessment of the overestimating of the surface cooling in the southern Vietnam coasts is missing, which is important for evaluating SST in the upwelling region. The reader is left wondering, the SST overestimating is caused by SYMPHONIE output or OSTIA? It further raises the question that if upwelling occurs in BoxMK in reality.

This question is related to the question of spatial observation of surface cooling that occurs in small coastal areas (see our answer to the question of upwelling over BoxMK above) : the scarcity of observations in summer over those coastal areas, due in particular to a high cloud cover, and the smoothing done to produce analysis data like OSTIA, explains a large part of the difference between SYMPHONIE and OSTIA. Note however that other factors could be involved that would explain an overestimation of surface cooling in the model : biases in atmospheric fluxes, overestimation of vertical mixing in SYMPHONIE due to the numerical design (schemes of advection and diffusion, vertical coordinates …) (though the comparison with in-situ temperature and salinity profiles shows the good performance of the model in the representation of water masses characteristics, section 3.2). Those questions are actually important topics of research in our group.

→ **A paragraph about this was added in the revised manuscript (lines 301-309)**

L233 L245, L261, L270, L275, L280 Inconsistencies in describing NRMSE, sometimes use "%", sometimes use decimal.

→ We provided all NRMSE values in decimal everywhere in the paper

L235, globally?

→ we removed this word

L292 - L299 Long description

→ We shorten this (**lines 359-3656**) removing in particular the long water masses names whose definition is provided in the caption of **Figure 8** of the revised manuscript.

L307 Figure SM1 is not found in the manuscript.

→ We provided this figure in the document containing all the figures, after Figure 17

L341 "... the lowest of the 4 boxes...". The lowest of what?

We meant the weakest upwelling interannual variability of the 4 boxes (lowest value of CV)

→ Our sentence was indeed not clear, we replaced it by "The inter-annual variability of SVU is significant, although it is the weakest of the 4 boxes" (**line 452**)

L391 Definition of "OIV" haven't been mentioned.

OIV (ocean intrinsic variability) was actually defined in the introduction when citing the work of Da et al. 2019 (**line 97**), but it was quite far from this occurrence of this acronym.

→ We recalled the definition **on line 664**

L357-L358 Values of UI need to be checked again.

→ We carefully checked our values of UIy for boxSC using a unique and constant Tref fpr all boxes (figure 16b): it is ~0.75°C in 2009, ~1.35°C in 2011, ~0.95°C in 2012, ~1.50°C in 2018, i.e. the same as obtained before.

L356-L365 The authors compare the differences between 2009 and 2012 of the daily/monthly wind stress and daily upwelling index and conclude that the daily to intra-seasonal variability of wind forcing modulate the SCU interannual variability. However, this analysis does not make sense to me because they are the different time scales. Similar comments for the OU and NCU.

Indeed, interannual variability of upwelling intensity daily variability to intraseasonal variability) of wind stress and upwelling are two different time scales. Our goal here was to show that the second (intraseasonal variability variability) influences the first (interannual variability), and how: depending on the daily chronology of wind forcing, the summer average of upwelling intensity, and thus its interannual variability, varies. For SCU, regular wind peaks all along the JJAS period result in a stronger summer average of upwelling intensity than intermittent peaks. For OFU, a stronger wind stress during the July-August period results in a stronger upwelling intensity. To better quantify the role of intraseasonal variability of wind stress in the summer average of upwelling and its interannual variability, we performed an additional simulation from June to September 2018 (the summer that shows the strongest wind stress and upwelling over BoxSC, BoxOF and BoxMK ). We prescribed during the whole summer period a temporally constant (but spatially varying) wind stress to the model: for each point of the model. This constant was computed as the JJAS average of the 2018 daily wind stress at this point. Initial conditions for this simulation were taken as the conditions of June 1st, 2018 of the 2009-2018 simulation. The average summer wind in this simulation and the average 2018 summer wind in the 2009-2018 simulation are therefore equal by construction, but wind in the sensitivity simulation does not show any daily to intraseasonal variability. In this sensitivity simulation, the surface cooling is much weaker than in the 2009-2018 simulation, and no upwelling develops on any of the boxes during the whole summer. This result quantitatively highlights the fundamental role of wind intraseasonal variability in the development of upwelling and in its summer average intensity.

→ Following this comment, we more clearly wrote the sentences about the link between intraseasonal variability of wind and summer average of upwelling intensity and its interannual variability in the revised version of the manuscript (**lines 466, 473-475, 520, 526**). We also added a section in the discussion in the revised manuscript where we discuss this point, to present the sensitivity simulation and to call for sensitivity simulations (**section 5.2 Role of intraseasonal variability of atmospheric forcing, p19** and **lines 785-787** in the conclusion).

L372, L373 "is (not) related to" should be " (does not) relate(s) to"

→ We replaced "is (not) related to" by "results from " or "does not results from"

L393 Vorticity calculation has not been described. What kind of vorticity? How do the authors define the surface current? Which depth layer of velocity do they use for the calculating?

Vorticity $\zeta$ is computed as the vertical component of surface current rotational: $\zeta = dv_{surf} / dx - du_{surf}/ dy$. The surface current is taken as the current of the first layer of the model, whose depth varies from ~1.00m over most of the domain to ~0.7 m in shallow areas very close to the coast (less than ~20 km).

→ This was detailed in the revised manuscript (**lines 413-416**)

L503-509 "This current constitutes …the stable position of MKU". This inference needs more evidences.

→ This part was revised to be less affirmative (**lines 589-595 and 603-605**)

Figures:

Figure1: VNC configuration should be described in detail, especially the coastal region, rather than locations of 4 upwelling areas, which are displayed again in Figure 2c.

→ we added a zoom in **Figure 1c** to show the details of the grid and bathymetry over the SVU region

Figures 3 and 4: There is an inconsistency that exists between the order of figures and the text's description, which makes the reader hard to follow.

Indeed, in the figure, the order is SST, SSS, SLA, and in the previous version, in the text, we commented successively SLA, SST and SSS, which made it difficult to follow.

→ In the revised manuscript, in **sections 3.2.1 and 3.2.2**, we now comment successively SST, SST and SLA, i.e. using the same order as for **Figures 2 and 7**.

Figure 5: "... ARGO (a, black dots and purple for mean), GLIDER (b, black dots and cyan for mean), IO-18 (c, black dots and green for mean) observations and from SYMPHONIE (yellow dots and red for mean) colocalized outputs..." should be "... ARGO (a, black dots and purple line), GLIDER (b, black dots and cyan line), IO-18 (c, black dots and green line) observations and from SYMPHONIE (yellow dots and red line) colocalized outputs...". However, the caption should be better clarified.

→ We have reworded the legend to make it clearer (**Caption of Figure 8**)

Figure 7: Legendaries of x and y axes are overlaid (year 2018)

→ This was corrected (**now Figure 13**)

Figure 8: Purple contours in a,b,c,d,i,j,k,l,q,r and blue contours in e,f,g,h,m,n,o,p,s,t have no explanation.

The pink contours show the area of positive wind stress curl. The blue contours corresponds to the contours of UIy=0.01°C

→ This was added in the caption of **Figure 12,** and wind stress and wind stress curl is shown **in Figure 11.**

Figure 9 and Figure 10: The ranges of y-axis should be fixed with the presented data

→ A hurricane hit the SVU region at the end of 2009 (see figure  H below extracted from the NOAA website). Wind and wind stress during this event were therefore exceptionally strong. Adapting the y-range would make the rest of the wind stress time series difficult to read, and the storm occurred at the very end of the simulation (1 value of the time series), it therefore does not have a strong impact on the upwelling on the summer average. We therefore deliberately keep the y-axis unchanged, but mentioned this in the **caption of Figures 14 and 15** of the revised mauscript. Note however that for BoxOF this storm, presumably due to the very strong positive wind stress curl, induced a strong upwelling during the last day of JJAS. This will be interesting to be investigated in more detailed studies.

[Figure]

**Figure H : Copy of NOAA website for the KETSANA hurricane that hit Vietnam end of September 2009**

References :

UK Met Office. 2005. OSTIA L4 SST Analysis. Ver. 1.0. PO.DAAC, CA, USA. Dataset accessed [YYYY-MM-DD] at https://doi.org/10.5067/GHOST-4FK01

Da, N. D., Herrmann, M., Morrow, R., Niño, F., Huan, N. M., and Trinh, N. Q. (2019). Contributions of wind, ocean intrinsic variability, and ENSO to the interannual variability of the south vietnam upwelling: A modeling study. Journal of Geophysical Research: Oceans, 124(9), 6545–6574. https://doi.org/10.1029/2018jc014647

M. Herrmann, S. Somot, F. Sevault, C. Estournel and M. Déqué (2008). Modeling deep convection in the Northwestern Mediterranean Sea using an eddy-permitting and an eddy-resolving model: case study of winter 1986-87. J. Geophys. Res. 113, C04011, http://dx.doi.org/10.1029/2006JC003991

Li, Y., Han, W., Wilkin, J. L., Zhang, W. G., Arango, H., Zavala-Garay, J., Levin, J., and Castruccio, F. S. (2014). Interannual variability of the surface summertime eastward jet in the South China Sea. Journal of Geophysical Research: Oceans, 119(10), 7205–7228. https://doi.org/10.1002/2014jc010206

Ngo, M., and Hsin, Y. (2021). Impacts of wind and current on the interannual variation of the summertime upwelling off southern Vietnam in the South China Sea. Journal of Geophysical Research: Oceans, 126(6). https://doi.org/10.1029/2020jc016892

To Duy Thai, 2022, Interannual to intraseasonal variability of the South Vietnam Upwelling. Role of multi-scale wind and ocean dynamics. A. high-resolution modeling study. PhD Thesis. Université de Toulouse 3, Toulouse, France.

**Comments on the paper entitled "Role of wind, mesoscale dynamics and coastal circulation in the interannual variability of South Vietnam Upwelling, South China Sea. Answers from a high resolution ocean model" by To Duy, Herrmann, Estournel, Marsalex, Duhaut, submitted to Ocean Science (os-2021-121)**

This paper investigates the interannual variability of the South Vietnam upwelling by using a modeling approach. The high resolution coastal circulation model is extensively validated by comparison with the data from different sources which makes the results convincing. The modeling approach seems appropriately designed for studying the upwelling events and their variability in a wide range of time scales: from daily to interannual. By considering high-frequency variations of the wind stress at the regional scale the authors have clearly demonstrated that **the magnitude of the wind variability at scales of days to weeks can partially explain large differences in upwelling intensity observed during years with rather similar mean wind forcing. This is the first valuable result of the study.** The second issue addressed is **how some specific features of the regional circulation can impact the interannual variations of upwelling by modifying the Ekman transport, precisely by adding a not wind-driven component to the total current velocity**. It was shown that the surface currents act differently in four considered sub-regions of a vast upwelling system of the South Vietnam. The background current can weaken of reinforce the upwelling intensity thus affecting the interannual variability.

The authors furnished an effort in analyses of modeling results, the data form observations, and they overall made up nice figures. I believe that conclusions of their work are interesting and could contribute to the general knowledge on scales of variability of the upwelling circulation in this part of the ocean and in other ocean regions.

I am convinced that this paper is worth publication after some major revisions. I provide below a list of the most important comments.

*We warmly thank the reviewer for this careful and constructive review of our paper. We addressed all the comments below in our revised version of the manuscript. In this document, the reviewer's comments appear in black, and our answers in blue. Changes done in the manuscript following the comments of the reviewer are also highlighted in blue in the revised version of the manuscript. Line numbers and pages in this document refer to the highlighted version.*

**Major comments**

- Abstract

The text after line 30 should be rewritten in order to demonstrate the forceful results. In the present version, I don't feel the major findings are presented in appropriate way. There a lot of generalities without precision and quantification. For example, it is difficult to understand what the authors mean by " … the impact of the … temporal organization of mesoscale ocean structures and atmospheric forcing". What is the message addressed in the last two lines: "… an interannual variability of upwelling (in Mekong box) is mostly determined by the summer wind and summer driven circulation in the region". I agree, but what novelty is behind this statement? I would suggest to avoid this kind of sentences and make the presentation of the results sharper, more incisive.

*→ The abstract was completely rewritten to take into account this comment, writing things in a clearer and more precise way (**p 2**).*

- Introduction

The scope of the study needs a more clear definition (ln.94-100). This research didn't start from zero. The role of the background current variability in the interannual variability of upwelling was already highlighted by Da et al. (2019). What was discovered before and what is focused in particular in the present study should be better introduced. This concerns the "processes", which were not clearly defined, and "scales" which are targeted.

Ln 99: The text should be reworded with respect to my previous comment. "The objective is … scientific" should be removed.

→ Following this comment, and the comment of the other reviewer, we developed in the Introduction the part about the existing knowledge about SVU (areas of development, role of wind, eddies and intrinsic ocean variability, **lines 70 to 107**). We also detailed the scope of the present study (**lines 114 to 129)** which fundamental objective is to better monitor, represent and understand the behavior of upwelling at smaller scales (meso to submesoscales), and over detailed areas (coastal to offshore) and the role of high frequencies (daily to intraseasonal).

Perhaps a clear definition of the numerical tool should be provided here. If it is different from the numerical model, it should be specified.

→ The numerical tool is actually the numerical model implemented over the area of study. This was not very clear so we removed this expression "numerical tool" and used "model" everywhere

- Section 2

The numerical model is briefly presented in this section. I think some terms require clarification. The first concerns "the biharmonic viscosity of momentum" and the second concerns "nudging". I assume the authors mean how the tidal motions were prescribed at the open boundaries. But the word tidal is missed in the text.

Following this comment we corrected the sentences to be more precise and rigorous :

→ "the biharmonic viscosity of momentum" was changed to "the viscosity of momentum associated with this biharmonic scheme" (the viscosity *per se* is not biharmonic ) **(lines 144-145)**

→ Open boundary conditions are prescribed for the temperature, salinity, and total (i.e. including effect of tide but not only) currents and sea surface height : "The lateral open boundary conditions, based on radiation conditions combined with nudging conditions, are described in Marsaleix et al. (2006) and Toublanc et al. (2018)." was changed to "The lateral open boundary conditions for temperature, salinity, current and sea surface height, based on radiation conditions combined with nudging conditions, are described in Marsaleix et al. (2006) and Toublanc et al. (2018)." **(lines 146-148)**

Ln 126: the authors use the term "zoom on the VN coast". I don't have impression that the technique of zoom was implemented in the model. This needs clarification.

Indeed technically it is not a classical zoom with 2 configurations at different resolutions.

→ We replaced "zoomed" by "with a refined resolution" **(lines 155)**

Presentation of the data sources. Sometimes, the information provided is absolutely useless: for example the program code, the name of PI. On the contrary, some acronyms need clear definition such as IO.

→ Text was modified to take into account this comment (remove useless information and add definitions such as IO) in **Section 2.2.2**, **p7)**

The term "hydrological characteristics of water masses" is misleading. The temperature and salinity are used for water mass characterization. What role the hydrology plays (the freshwater input, as I understand the term) in modifying T,S characteristics is unclear.

Indeed, some authors use the term "hydrological characteristics" for temperature and salinity characteristics (eg. *Criado-Aldeanueva et al. 2006*), but we agree that it is quite uncommon and may be confusing.

→ We consequently changed the occurrences of "hydrological characteristics" in the manuscript to "Temperature and salinity (TS) characteristics" (**lines 193, 353, 359, 384**)

My major concern is about the definition of the upwelling indicators and the choice of the reference box which area is much smaller than that of the corresponding upwelling box. The authors should justify their choice of the reference box size and the temperature values. When the size is small and the reference location is close to the upwelling region, the reference temperature is obviously dependent on the temperature observed during the upwelling event. To what degree this quantity is independent? This needs clarification as the results could be sensitive to the choice of the reference value. What could be the difference if an overall mean temperature (space and time mean) is used as the reference value?

Following this comment and a comment from the other reviewer, we slightly modified (and justified) our computation of Tref.

First, we completely agree that the SST over the reference boxes chosen for BoxMK, BoxSC and BoxNC may be influenced by the propagation into the reference box of water upwelled in other areas. This is all the more the case when the reference box is small. Moreover, using different Tref makes the upwelling index field spatially discontinuous. We therefore recomputed everything with the same Tref for all the boxes, taking Tref for BoxOF, which can be considered as large and far enough from the upwelling areas. We obtain a value of Tref=29.20°C. Using different boxes, our former values were $Tref_{NC}$=29.66°C, $Tref_{SC}$ = $Tref_{MK}$= $Tref_{OFF}$ 29.20 °C. Except for BoxNC, using different references boxes actually was thus equivalent as using a unique reference box. Note also that even for BoxNC, the difference of upwelling index value induced by the new choice of Tref will be very small. Given the formulae used for UId and UId (Equations 1 and 2 of the revised manuscript, see below), the relative difference between UIy and UId computed using the new and the old Tref will be equal to (Tref,new - Tref,old)/Tref,old = (29.2-29.7)/29.7 = -0.017 ~ -2%).

$$UI_{d,boxN}(t) = \frac{\iint_{(x,y)in\ boxN\ so\ that\ SST(x,y,t)<To}(T_{refN}-SST(x,y,t)).dx.dy}{A_{boxN}}(1) \quad and \quad UI_{y,boxN} = \frac{\int_{JJAS}UI_{d,boxN}(t)dt}{ND_{JJAS}}(2)$$

Table A below shows the updated values for Table 2 of the manuscript. Logically (since Tref only changes for BoxNC, by less than 2%), UIy (mean, std, CV) values do not or only very slightly change. Moreover, most of the correlation values remained unchanged. Those which changed changed by less than 0.03 (see table below). Our conclusions concerning the relationships between the different factors were therefore robust to this choice of reference temperature.

**Table A : Modified Table 2 with a constant and unique Tref=29.2° (values that were modified compared to the previous manuscript are highlighted in red and italics). From 1st to last line : temporal mean and standard deviation of UIy,boxN over 2009-2018 for each box and coefficient of variation CV (which is the ratio between STD and mean), correlations (correlation coefficient and associate p-values) between time series of significant factors : yearly upwelling index over each box vs. yearly upwelling index over other boxes, vs. average wind stress averaged over June-September (JJAS) and July-August (JA) over each box, vs. integrated positive vorticity over BoxOF. Correlations significant at more than 99% (p<0.01) are highlighted in bold.**

|  | BoxNC | BoxSC | BoxOF | BoxMK |
|---|---|---|---|---|
| $UI_{y,boxN}$ mean (°C) | 0.163 *vs.* *0.195* (0.14 in the previous manuscript was a typo) | 0.798 (0.42 in the previous manuscript was a typo) | 0.074 (0.09 in the previous manuscript was a typo) | 0.065 |
| $UI_{y,boxN}$ STD (°C) | 0.118 *vs. 0.14* | 0.423 | 0.093 | 0.055 |
| CV (%) | 72 *vs. 71* | 53 | 126 | 85 |

| Correlation between : | $UI_{y,NC}$ | $UI_{y,SC}$ | $UI_{y,OF}$ | $UI_{y,MK}$ |
|---|---|---|---|---|
| $UI_{y,SC}$ (ºC) | 0.00(0.99) *vs.* *+0.01(0.98)* | **1** | **+0.73(0.02)** | **+0.83(0.00)** |
| $UI_{y,OF}$ (ºC) | -0.26(0.47) | **+0.73(0.02)** | **1** | **+0.92(0.00)** |
| $UI_{y,MK}$ (ºC) | -0.19(0.59) | **+0.83(0.00)** | **+0.92(0.00)** | **1** |
| $WS_{JJAS,NC}$ (N.m$^{-3}$) | -0.09(0.78) *vs.* *-0.10(0.77)* | -0.41(0.24) | +0.23(0.53) | +0.07(0.85) |
| $WS_{JJAS,SC}$ (N.m$^{-3}$) | -0.13(0.72) | +0.85(0.00) | +0.76(0.01) | +0.83(0.00) *vs.* +0.84 (0.99) |
| $WS_{JJAS,OF}$ (N.m$^{-3}$) | -0.19(0.60) | +0.81(0.00) | +0.77(0.01) | +0.80(0.01) |
| $WS_{JJAS,MK}$ (N.m$^{-3}$) | -0.08(0.81) | +0.78(0.01) | +0.63(0.05) | +0.72(0.02) |
| $WS_{JA,NC}$ (N.m$^{-3}$) | +0.04(0.90) *vs.* *+0.07(0.92)* | +0.18(0.62) *vs.* *+0.12(0.62)* | +0.54(0.11) | +0.38(0.28) |
| $WS_{JA,SC}$ (N.m$^{-3}$) | -0.15(0.69) | **+0.70(0.03)** | **+0.84(0.00)** | **+0.84(0.00)** |
| $W_{JA,OF}$ (N.m$^{-3}$) | -0.15(0.67) | **+0.69(0.03)** | **+0.84(0.00)** | **+0.82(0.00)** |
| $W_{JA,MK}$ (N.m$^{-3}$) | -0.11(0.77) | **+0.72(0.02)** | **+0.78(0.01)** | **+0.82(0.00)** |
| $\zeta_{+,OF}$ (s-1) | -0.28(0.43) | **+0.60(0.07)** | **+0.69(0.03)** | **+0.74(0.01)** |

**Table B : Modified Table A with a varying Tref (values that were modified compared to Table A are highlighted in red) :**

|  | BoxNC | BoxSC | BoxOF | BoxMK |
|---|---|---|---|---|
| $UI_{y,boxN}$ mean (ºC) | 0.171 | 0.743 | 0.060 | 0.055 |
| $UI_{y,boxN}$ STD (ºC) | 0.140 | 0.344 | 0.065 | 0.040 |
| CV (%) | 82 | 46 | 108 | 73 |
| Correlation between : | $UI_{y,NC}$ | $UI_{y,SC}$ | $UI_{y,OF}$ | $UI_{y,MK}$ |
| $UI_{y,SC}$ (ºC) | +0.11(0.78) | 1 | +0.63(0.05) | +0.73(0.02) |
| $UI_{y,OF}$ (ºC) | -0.28(0.42) | +0.63(0.05) | 1 | +0.81(0.00) |
| $UI_{y,MK}$ (ºC) | -0.09(0.80) | +0.73(0.02) | +0.81(0.00) | 1 |
| $WS_{JJAS,NC}$ (N.m$^{-3}$) | -0.01(0.97) | -0.43(0.21) | +0.21(0.56) | +0.09(0.80) |
| $WS_{JJAS,SC}$ (N.m$^{-3}$) | -0.25(0.48) | +0.72(0.02) | +0.77(0.01) | +0.76(0.01) |
| $WS_{JJAS,OF}$ (N.m$^{-3}$) | -0.30(0.39) | +0.66(0.04) | **+0.77(0.01)** | +0.69(0.03) |
| $WS_{JJAS,MK}$ (N.m$^{-3}$) | -0.20(0.57) | +0.66(0.04) | **+0.63(0.05)** | +0.65(0.04) |
| $WS_{JA,NC}$ (N.m$^{-3}$) | +0.05(0.89) | +0.14(0.70) | +0.53(0.11) | +0.30(0.39) |
| $WS_{JA,SC}$ (N.m$^{-3}$) | -0.23(0.52) | +0.54(0.11) | +0.82(0.00) | +0.73(0.02) |
| $W_{JA,OF}$ (N.m$^{-3}$) | -0.23(0.52) | +0.53(0.11) | +0.82(0.00) | +0.70(0.03) |
| $W_{JA,MK}$ (N.m$^{-3}$) | -0.20(0.59) | +0.59(0.07) | +0.77(0.01) | +0.74(0.02) |
| $\zeta_{+,OF}$ (s$^{-1}$) | -0.37(0.30) | +0.43(0.22) | +0.68(0.03) | +0.64(0.05) |

Second, we chose a constant Tref, as done previously by *Da et al. (2019)*, whereas other studies, like *Ngo and Hsin (2021)*, used an interannually varying Tref, to take into account the fact that SST can also vary on an interannual basis. We justify our choice of a constant Tref by the fact that even if the reference box was chosen

outside the upwelling area, the SST in this box can be influenced by the eastward advection of water upwelled in the other boxes. This can be seen on Figure A that shows the JJAS map of simulated SST for summers 2010 (year of weakest upwelling) and 2018 (year of strongest upwelling). In 2018, the SST in the reference area is cooler than in 2010 due to the eastward advection of upwelled water. The upwelling index computed from the difference $SST(x,y,t) - SSTref$, would therefore be smaller in 2018 and larger in 2010 using a varying Tref vs. a constant Tref. In other words, a varying Tref would increase the weak values and decrease the strong values, hence reducing the interannual variability. Note however that the impact on weak values is actually limited by the use of the threshold temperature $T_0$. We investigated the influence of this choice on our results. Figure B shows the time series of Tref computed annually instead of a constant Tref, and the resulting yearly time series of UIy for each box. Table B provides the modified values of Table A. The resulting UIy values slightly vary, in particular for stronger values (see year 2018), but that the change is not significant. The interannual variability remains nearly the same, though it slightly decreases, as expected. Correlation coefficients consequently also slightly decrease (by at most ~0.10). However correlations that were statistically significant (or not), remain statistically significant (or not). Our conclusions are therefore still valid.

[Figure]

**Figure A : JJAS average simulated SST in 2010 and 2018 (°C).**

→ Following this comment**, Figures 12, 13, 14, 15, Table 2 of the revised manuscript** were modified after using the same and constant Tref for all boxes. The definition of Tref was modified (**Lines 225-229 in Section 2.3**). A new section **(Section 5.1, p19)** and a new figure (**Figure 16**) were moreover added to discuss the sensitivity of our results to a varying vs. constant Tref.

[Figure]

**Figure B : (1st row ) Yearly time series of interannually varying summer averaged SST over the reference box (i.e. =varying Tref, full black line) and value of the climatological summer averaged SST over the reference box (i.e. =constant Tref, dashed line). Yearly time series of UIy computed using a varying (red) vs. constant (blue) Tref, for BoxNC (2nd row), BoxSC (3rd row), BoxOF (4th row) and BoxMK (5th row).**

Presentation of the results
- Section 3

As I indicated above, the authors furnished an effort in analyses of the data from different sources and in validation of the modeling results. The results are convincing. However only spatial distribution of different quantities at the surface is used in comparison. But the model is three-dimensional and high resolution. A demonstration of the **model capability in reconstructing the upwelling circulation (and related water properties) in the vertical plan** can be an added value. This can support the choice of high-resolution in the horizontal and also in the vertical.

The reviewer underlines the need to demonstrate the model capability in reconstructing the upwelling circulation (and related water properties) in the vertical plan. This was indeed our objective when comparing observed and simulated TS diagrams over the region (ARGOS, Figure 8a of the revised manuscript, and GLIDER, Figure 8b) but also in the upwelling area (IO-18 data, Figure 8c). Those data are not restricted to the surface, and go until ~200m to ~1000m, i.e. covering vertically the extent of the upwelling.

There are only few data that allow to observe the vertical dimension of the SVU during its period and over its are of development. IO-18 CTD profiles collected during September 2018 allow to explore this dimension. Figure C below shows the T and S profiles observed by IO-18 and simulated by SYMHONIE at each station of the IO-18 cruise. IO-18 TS diagram and profiles reveal two types of profiles in the deeper regions (i.e. reaching 150 m depth), corresponding to the nearly vertical part of the TS diagram. The first type of profiles was sampled along the section at ~12.7°N (points 2.1 to 2.5, located in BoxNC) and in the coastal part of the section along 10.5°N (points 3.2,

3.4, 3.5, located in BoxSC). It shows high salinities (> 34.5) and low temperatures (< 25°C) below a pycnocline shallower than ~30 m. It corresponds to location where upwelling still occurs at this period. The second type of profiles is sampled in the offshore part of a section at 10.5°N (points 3.6 to 3.8, in BoxOF). It has deeper haloclines and thermoclines, reaching 90 m, with SSS between 32.8 and 33.6 and warmer SST around 28°C : the upwelling already ceased in this region at this period. Both the TS diagram (**Figure 8c** of the revised document) and the TS profiles show that the model is able to reproduce this diversity of TS profiles in the coastal and offshore upwelling regions in very good agreement with IO-18 data, without any significant bias, even after 9 years of simulation.

→To take into account this comment and better highlight this vertical aspect, we therefore also show the vertical TS profiles of simulated and observed during IO-18 cruise (**Figure 10 of the revised manuscript**) and commented those figures in the revised paper of the manuscript (**end of section 3.3, lines 386-401**).

Last, collecting current data as well as high resolution observations in the SVU region and period, for example from glider campaigns, would allow to evaluate more precisely the ability of the model to reproduce the dynamics and water masses over the vertical dimension.

→We added a sentence to underline this need for future observations in the Discussion (**lines 731-735 in Section 5.4).**

[Figure]

**Figure C : Temperature (°C) and salinity profiles sampled during IO-18 campaign (blue) and simulated by SYMPHONIE (red) between September 12 and 25 in the SVU region.**

The authors often use the term "coastal scale". The word coastal is not appropriate for scale definition. The scale needs clarification.

→ Indeed, we sometimes actually more precisely meant "coastal area", or "coastal circulation". We modified the text accordingly everywhere where "coastal scale" was used before.

- Section 4

In this section the authors explore in detail the upwelling variability for each year. Nine years in total are considered with strong and weak upwelling events. The effect of the mean wind in interannual variability of upwelling was verified but this is not a novel result. Mechanisms which can explain large difference in upwelling intensity between years with a similar mean wind are identified.

I have two major points of criticism regarding this part of the study.

The first point concerns the interpretation of the variability of the wind stress at high time resolution which is presented as the "other factor modulating the wind induced interannual variability of SCU" (ln 354).

The idea behind seems clear, but what is less clear how to quantify and interpret the effect of high-frequency variability. A method or a metric should be used in this demonstration. A visual inspection of the wind stress curves given in Fig. 9,10 is not sufficient. The authors use CV, the coefficient of variation. But how it helps in quantification of the contribution of high frequency compared to low frequency variability? This needs clarification.

The conclusion that the intraseasonal chronology of the wind forcing influences the seasonal average of the upwelling over Box SC and BoxOF was indeed obtained from the study of four summers (2009, 2011, 2012 and 2018). To quantitatively confirm this conclusion, we performed an additional simulation from June to September 2018 (the summer that shows the strongest wind stress and upwelling over BoxSC, BoxOF and BoxMK, Figure 13 of the revised manuscript). We prescribed during the whole summer 2018 a temporally constant (but spatially varying) wind stress to the model: for each point of the model, this constant was computed as the JJAS average of the 2018 daily wind stress at this point. Initial conditions for this simulation were taken as the conditions of June 1st, 2018 of the 2009-2018 simulation described in section 2. The summer average of summer wind in this simulation is therefore equal by construction to the average of summer 2018 wind in the 2009-2018 simulation, but it does not show any daily to intraseasonal variability. In this sensitivity simulation, the surface cooling is much weaker than in the 2009-2018 simulation, and no upwelling develops on any of the boxes during the whole summer. This quantitatively highlights the fundamental role of wind intraseasonal variability in the development of upwelling and in its summer average intensity.

Additional simulations would now be required to more quantitatively assess the role of intraseasonal variability vs. seasonal average of wind stress on the yearly upwelling intensity. In particular, ensembles of simulations with wind of same seasonal average but different daily chronology, and conversely, would help to estimate the variability of summer averaged upwelling induced by the variability of the intraseasonal chronology vs. by the interannual variability of summer average wind.

→ Following this comment, we added a section in the discussion (**Section 5.2 Role of intraseasonal variability of atmospheric forcing, p19**) in the revised manuscript where we discuss this point, present the sensitivity simulation and call for sensitivity simulations.

I see a small inconsistency in the interpretation of the "other factor modulating the wind induced variability". First, in this particular case, the high and low frequency variability of upwelling is wind-induced. The physical process involved in upwelling generation is the same. Second, I cannot imagine how the high frequency can modulate the low frequency signal. Different averaging techniques can provide different values of the mean quantity. But this is not sought as modulation. The authors should find a better formulation.

Indeed, the term "modulation" is a specific technical term in the domain of frequency signals, and using it here was a misuse of language.

→ We removed this term, and to replace it by more appropriate expressions : e.g. "contribute to", "influence" everywhere where it was used in the previous manuscript.

The second point of criticism concerns the role of the wind stress and surface currents in upwelling variability in different years. From my point of you, the **surface current velocity and the current velocity curl are tightly related to the wind stress curl** (example of anti-symmetric eddies and the eastern current). I have impression that only the wind stress magnitude is used in analysis. The added value of the study will increase if **analysis of the wind curl and perhaps the wind stress vector field can be introduced and comparison with current velocities be made**. This will help in interpretation of the surface current variability. What part of the current variability is wind-induced? and what part is remotely induced? The choice of the method of quantification is an important issue. And this is related to the statement (used for the second time) "another factor of non-wind origin" controlling the variability of upwelling. A part of the background current variability, independent of the wind, should be clearly identified and characterized. This requires a method of identification. I didn't see a clear description of such a method in the manuscript.

[Figure]

**Figure D: Maps of JJAS averaged wind stress (arrows, N.m⁻²) and wind stress curl (colors, N.m⁻³) for each year of the simulation.**

Following this comment, we developed the description and discussion of link between wind stress, wind stress curl, vorticity and upwelling, over the different areas of development :

1) Figure D above shows the maps of JJAS averaged wind stress and wind stress curl for each summer of the simulation, computed from ECMWF atmospheric forcing. The intensity of wind stress and wind stress curl shows a strong interannual variability, but their spatial patterns are quite similar from one year to another and related to the summer monsoon wind : a wind stress curl dipole develops offshore Vietnam, with an area of strong positive curl along and off the Vietnam coast in the north (covering BoxSC and a part of BoxOF), and an area of strong negative curl in the south (covering BoxMK and a part of BoxOF). Time series of the summer wind stress

over BoxSC and BokMK are almost equal to the summer wind stress over BoxOF (Figure 13) and completely correlated with it (>0.97, p<0.01, Table 2). The intensity and interannual variability of summer wind over BoxSC and BoxMK is thus completely driven by the large-scale wind over the region.

Correlations between JJAS wind stress (WS), wind stress curl (WSC) over each box and ocean vorticity over boxOF ($\zeta_{+,OF}$) where also computed (see Table C below). There is a highly significant correlation (> 0.95) between wind stress over BoxOF, BoxSC and BoxMK, and wind stress curl over both BoxSC (area of maximum positive wind stress curl of the wind dipole) and BoxMK (area of maximum negative wind stress curl of the wind dipole). The intensity of wind stress over the regions of upwelling is therefore strongly related to the intensity of the wind curl dipole located along the Vietnamese coast. We already showed that the interannual variability of JJAS regional wind stress partly drives the regional scale circulation (current jet + AC/C dipole, correlation of 0.89 between $\zeta_{+,OF}$ and $WS_{JJAS,OF}$, Table 2 of the manuscript). This circulation is also related to the wind stress curl (correlation of 0.82 and -0.81 between $\zeta_{+,OF}$ and $WSC_{JJAS,SC}$ and $WSC_{JJAS,MK}$, respectively).

**Table C : correlation (and p-value) between JJAS average of wind stress curl (WSC) over each box, and JJAS average of wind stress (WS) and of BoxOF positive vorticity**

| Correlation between : | $WSC_{JJAS,NC}$ | $WSC_{JJAS,SC}$ | $WSC_{JJAS,OF}$ | $WSC_{JJAS,MK}$ |
|---|---|---|---|---|
| $WS_{JJAS,SC}$ | 0.40(0.26) | **0.95(0.00)** | **0.69(0.03)** | **-0.96(0.00)** |
| $WS_{JJAS,OF}$ | 0.42(0.23) | **0.96(0.00)** | **0.71(0.02)** | **-0.92(0.00)** |
| $WS_{JJAS,MK}$ | 0.24(0.51) | **0.96(0.00)** | **0.83(0.00)** | **-0.91(0.00)** |
| $\zeta_{+,OF}$ | 0.59(0.07) | **0.82(0.00)** | 0.46(0.18) | **-0.80(0.01)** |

→ **A section was added in the revised manuscript (Section 4.1, Interannual variability of wind and large scale circulation), where this question regarding the link between wind stress, wind stress curl and large scale circulation and vorticity is addressed. Correlations between wind stress, wind stress curl and ocean vorticity where added in Table 2. Maps of wind stress vector field and curl was added in Figure 11 of the revised manuscript.**

2) $\zeta_{+,OF}$ quantifies the intensity of cyclonic circulation over the offshore are, including both the intensity of the eastward jet, and of the cyclonic eddy circulation that develops northern of the jet. This eddy circulation is partly induced by wind, as shown by the highly significant correlations between $\zeta_{+,OF}$ and $WSC_{JJAS,SC}$, $WSC_{JJAS,MK}$ and $WS_{JJAS,OF}$. However it has by nature a strong chaotic part (see previous studies of *Waldman et al. 2018, Sérazin et al. 2016, Da et al. 2019*, cited in the Introduction). For example, wind is similar for 2014 and 2016 ($WS_{JJAS,OF} \simeq 0.08$ N.m$^{-2}$, Figure 13 of the revised manuscript), with similar patterns of wind stress curl (Figure D above), however $\zeta_{+,OF}$ is twice larger (~$2.6 \times 10^{-6}$ s$^{-1}$, Figure 13 of the revised manuscript) than in 2016 (~$1.5 \times 10^{-6}$ s$^{-1}$) and a very different circulation develops for both years (Figure 12 of the revised manuscript). This confirms that the variability of surface circulation and associated vorticity over the SVU region has a forced component driven by summer averaged regional wind, but also a chaotic component related to OIV. Ensemblist approaches that allow to distinguish and quantify the effect of the chaotic vs. forced component of ocean dynamics at different scales, are now required to understand which part of variability of the current and its vorticity, and of the upwelling, is wind-induced and which part is related to the formation and propagation of eddies of strongly chaotic nature

→ A section was added in the revised manuscript (**Section 5.3 Role of forced vs. chaotic variability, p20**) where this question about the forced vs. chaotic part of surface circulation and upwelling is discussed.

- Discussion and conclusions

Section 4.5 and section 5 (Conclusion) should be reorganized. Section 4.5 contains the discussion of the modeling results and should be entitled "Discussion". A part of section 5 also contains the discussion and should be relegated to "Discussion".

Conclusion section should contain the major and novel results in a condense form. The comparison with previous studies is already done (in principal) in Discussion section. The form is important. Please avoid sentences of five lines difficult to follow. Highlight what knew knowledge the study brought and in what it is different compared to the results of previous studies.

→ Following this comment, we deeply reorganized and rewrote the discussion (now the whole **Section 5, pages 19 to 22**) and conclusion (**Section 6, page 23-34**). In particular, we added in the new Section 5 a discussion about the sensitivity of our results to the choice of the reference temperature (**Section 5.1**), the role of intraseasonal variability of atmospheric forcing (**Section 5.2**), the role of forced vs. chaotic variability (**Section 5.3**), the role of local upwelling vs. lateral advection of cold water in OFU (**Section 5.5**). The conclusion was shortened and sharpened in order to highlight the new results of this study (it is now a little bit longer than one page).

Technical corrections

Ln 30: move "driving" to different location … our results confirm the role of the … in driving the interannual …

→ This was corrected, **lines 31-32**

Ln 34: perhaps "structures of circulation" ?

→ This was indeed not clear, and disappeared after the revision of the abstract

I would recommend replacing the word ability, when you talk about the model skill, by capability, in the whole text.

→ This was corrected (9 occurences in the text)

Ln. 103-105: When describing the paper structure, it is better to use the word Section, not Part.

→ This was corrected everywhere in the document

Ln 45-46: Please reword the text concerning the CSS contribution.

→ "contributes" to was replaced by "influences", **line 48**

Ln 91 ageostrophic dynamics

→ This was corrected, **line 112**

Ln 151-152 … level 3 SSS derived from SMOS … (MIRAS) measurements at 0.25° resolution

→ This was corrected, **line 184**

Ln163-164: The modeled outputs were spatially and temporally co-localized with observations and used for comparison.

→ This was corrected, **line 195**

Ln 165: put a dot after "area". The data are available …

→ This was corrected, **line 197**

Ln 201: I think there is a conventional way how to refer to a Figure in another publication: (cf. Fig. 1, Da et al., 2019)

→ This was corrected, **lines 243-244** (we also guess that this will be formatted in the journal format at the final stage of the publication)

Ln 206-2020: The text should be edited to make it clear. Frequency should be removed. I suggest : … from the analysis of the occurrence of …

Please choose the right order in index definition: UI should match upwelling index

Perhaps is it more simple to use a number 122 (days in four months) instead of NDjjas ?

→ The text was carefully edited to make it clearer, **lines 229-239**

Ln 224: the title: Surface circulation, temperature and salinity in the SCS. Perhaps it is better than hydrological characteristics.

→ This was corrected**, line 276**

The text in four lines following the title should be rewritten. I don't understand "interannual yearly averages". Do you mean" monthly mean and yearly mean values" ? Please check the sentence structure and articles.

→ Indeed it was confusing : we replace this by "Figure 3 shows the time series of climatological monthly averages and of yearly averages of simulated and observed SST" **, lines 278-279**

Ln 221: Title : cycle ("c") an variability of what?

→ We detailed the title : *Annual cycle and seasonal variability of surface circulation, temperature and salinity* , **line 287**

Ln 237: remove "the" before Introduction and put "the" before northern monsoon wind.

→ This was corrected, **lines 327-328**

 Ln 254: in very good overall agreement

→ This was corrected, **line 299**

Ln 285: Choose a better title.

→ We detailed the title : Interannual variability of surface circulation, temperature and salinity », **line 333**

p9 "coastal scale" is used in some places but the scale is not defined.

→ Following the comment above the expression"coastal scale" is not used anymore

Ln 323-325: This text and the text in ln 326-330 is repetition. Lines can be removed.

The text at the end of section 3.2 focuses on the observation and simulation of vertical profiles during the period and in the area of upwelling in 2018, whereas the text at the beginning of section 4 is a synthesis of the conclusion of section 3.

→ The text at the end of 3.2 was modified following the comment above about the model capability in reconstructing the upwelling circulation (and related water properties) in the vertical plan, **lines 386-401**

Ln 328: 10-year long simulation.

→ This was corrected, **line 406**

Ln 329: prefer " in four regions" to 4 regions

→ This was corrected, **line 407**

Ln 350: UI=1.49. I read 2.0 in Fig. 7.

On figure 7 (now Figure 13), we plot the relative values $UI_{y,boxN}/mean(UI_{y,boxN})$ to make the values readable. Indeed, ranges of absolute values of $UI_{y,boxN}$ vary between ~0-0.2°C for boxMK to ~0-1.5°C for BoxOF. We provide the values of $(meanUI_{y,boxN})$ in Table 2.

→We carefully checked the values of $UI_{y,boxN}$ that we provided in the text, especially after changing to a common Tref for all boxes.

Ln 374-375: the next needs rewording. If possible, provide the exact location of each of four eddies in Fig 8. Put a symbol for example, or provide coordinates in the text.

→ We added black arrows on Figure 12 to show the position of cyclonic and anticyclonic structures and the resulting jets, and rewritten this with shorter and clearer sentences, **lines 486-488**

Ln 390: perhaps the word "chaotic" is not appropriate if the structures are visible in the mean field. Do you mean large scale turbulent structures? If not, clarify the meaning.

Indeed those structures themselves are not chaotic, but their development and propagation has a strongly stochastic part (this is related to the comment about the forced vs. chaotic part of the vorticity above)

→ This was rewritten (see **Section 5.3, and lines 669, 707, 711, 781).**

Ln 411-415. Perhaps reword the text. Too long and difficult to follow.

→ This was rewritten, with shorter and clearer sentences, **lines 504-507**

Ln 532: these differences

→ This was corrected, **line 691**

Ln 541-542: the SCS. The text should be rewritten.

→ This was rewritten, **lines 738-745**

571: zonal location is preferable to position

→ This was corrected, **line 765**

References

Criado-Aldeanueva, F., García-Lafuente, J., Vargas, J. M., Del Río, J., Vázquez, A., Reul, A., & Sánchez, A. (2006). Distribution and circulation of water masses in the Gulf of Cadiz from in situ observations. Deep Sea Research Part II: Topical Studies in Oceanography, 53(11-13), 1144-1160., doi:10.1016/j.dsr2.2006.04.012

Da, N. D., Herrmann, M., Morrow, R., Niño, F., Huan, N. M., and Trinh, N. Q. (2019). Contributions of wind, ocean intrinsic variability, and ENSO to the interannual variability of the south vietnam upwelling: A modeling study. Journal of Geophysical Research: Oceans, 124(9), 6545–6574. https://doi.org/10.1029/2018jc014647

Sérazin, G., Meyssignac, B., Penduff, T., Terray, L., Barnier, B., & Molines, J. M. (2016). Quantifying uncertainties on regional sea level change induced by multidecadal intrinsic oceanic variability. Geophysical Research Letters, 43, 8151–8159. https://doi.org/10.1002/2016GL069273

Waldman, R., Somot, S., Herrmann, M., Sevault, F., & Isachsen, P. E. (2018). On the chaotic variability of deep convection in the Mediterranean Sea. Geophysical Research Letters, 45, 2433–2443. https://doi.org/10.1002/2017GL076319

Ngo, M., and Hsin, Y. (2021). Impacts of wind and current on the interannual variation of the summertime upwelling off southern Vietnam in the South China Sea. Journal of Geophysical Research: Oceans, 126(6). https://doi.org/10.1029/2020jc016892

---

## Author Response (AR2)

**Answer to Comments on "Role of wind, mesoscale dynamics and coastal circulation in the interannual variability of South Vietnam Upwelling, South China Sea. Answers from a high-resolution ocean model" by Tai To Duy et al. in Ocean Science**

In this document, the reviewer's comments appear in black, and our answers in blue.

This is the second round review of the manuscript. The authors have tried to addressed my concerns in the last review. Some major comments are still needed to be addressed before being acceptable.

We warmly thank the reviewer for this careful and positive review of our paper. We addressed all the comments below in our revised version of the manuscript. Changes done in the manuscript following the comment of the reviewer are highlighted in blue in the revised version of the manuscript. Line numbers correspond to the highlighted version.

Major Comments:

1. Section 2.3:
Regarding the definition of the upwelling index, I have three points of criticism.

a) In lines 229-232, "The threshold temperature below which upwelling happens is defined from the analysis of the occurrence of cold surface water: it is defined as the temperature that allows to cover the largest number of upwelling occurrences but avoids to include cold water advected between areas. We obtain To = 27.6°C". The authors obtained T0 based on the two conditions: (1) "the largest number of upwelling occurrences" and (2) "excluding advected cold water from other boxes"; however, no further explanation about how they dealt with two two conditions was given. Did the authors conduct any sensitive tests for the value of T0? Da et. al (2019) set up the threshold of T0=27.5 by applying sensitive tests of the frequency of occurrence but did not mention advected water. Did they use the same method or have any improvements?

We used exactly the same methodology as Da et al. (2019). SST-based upwelling index is a widely used method. Moreover, during his PhD, Da (2018) studied carefully the choice of upwelling criteria (Da et al. 2019, and see pages 85-86 of the PhD thesis of Da (2018) available on http://lotus.usth.edu.vn/uploads/PhD_NguyenDacDa_2018.pdf). We therefore did not redo this part of the study, and directly used the methodology of Da et al. (2019). Following this methodology, To is defined from the frequency of occurrence of cold surface water, as shown on Figure A below. For that, we vary To from 26.0°C to 28.0°C every 0.1°C to select the threshold temperature. The goal of those tests is to select the temperature that allows to cover the largest number of upwelling occurrences, but avoids to include cold water advected between areas. From our tests, 27.6°C is the optimal choice (Figure A).

→ Following this comment we modified the text, referred to Da (2020) and Da et al. (2019), added the part about the tests between 26°C and 28°C, and rewrote and rearranged the text to be more clear (Section 2.3, lines 223-242).

[Figure]

*Figure A : Upwelling index frequency (% of June-September (JJAS) period) defined as the frequency of events for which SST < To for different choices of To (color contours).*

b) In the revised manuscript, the authors decided to use one Tref for the four boxes of BoxSC, BoxNC, BoxOF and BoxMK to help UIy change continuously in space. And, the authors stated that the difference between the upwelling indices using old and new Tref is small (2%) based on a simple calculation (Tref,new - Tref,old)/Tref,old = (29.2-29.7)/29.7 = -0.017 ~ -2%). I think this argument is not correct. In fact, from Table A of Response, I find that difference between new and old UImean in BoxNC is 19.6% ((0.163 – 0.195)/0.163) and the difference between the standard deviation of new and old UI in BoxNC is 18.6% ((0.118-0.14)/0.118). In sum, the difference between the upwelling indices using old and new Tref is around 20%, not as small as they thought (just 2%).

Indeed, computing carefully the relative difference between upwelling index computed using the "old" and "new" Tref gives :

$$\text{Delta(UI)} / \text{UI} = (\iiint (\text{Tref,new} - T)\, dxdydt - \iiint (\text{Tref,old} - T)\, dxdydt )/ \iiint ( \text{Tref,old} - T)\, dxdydt$$

$$= \iiint (\text{Tref,new} - \text{Tref,old})\, dxdydt / \iiint (\text{Tref,old} - T)\, dxdydt$$

$$= ( \text{Tref,new} - \text{Tref,old} ) \iiint dxdydt / \iiint (\text{Tref,old} - T)\, dxdydt$$

Since since T < To, then $|\text{Tref,old} – T| > | \text{Tref,old} – To|$, and

$$|\text{Delta(UI)} / \text{UI}| < | \text{Tref,new} – \text{Tref,old} | \iiint dxdydt /\ | \text{Tref,old} – To |\iiint dxdydt$$

$$= \lvert \text{Tref,new} - \text{Tref,old} \rvert \,/\, \lvert \text{Tref,old} - \text{To} \rvert = (29.7\text{-}29.2)/(29.7\text{-}27.6) = 0.23$$

So finally $\lvert \text{Delta(UI) / UI} \rvert < 23\%$ (and not 2% !)

We thank the reviewer for pointing out this a mistake in the previous answer. Our corrected calculation shows that the relative difference between the upwelling index computed with a different Tref or a unique Tref is smaller than 23%, which is consistent with the 20% difference underlined by the reviewer. This mistake done in the previous answer to the reviewer's comment actually did not have any consequence on our computations of upwelling indexes and related conclusions in the paper. In the previous version of the paper, we indeed did not mention this computation. Moreover we showed in the previous answer that using a common Tref instead of a specific Tref for each box did not induce any significant change in the computed correlations.

→ Following this comment, we did not need to correct the manuscript itself, but mentioned the impact of this choice of common vs. specific Tref for each box in the text (Section 5.1, lines 650-655).

c) In Response and Section 5.1 of revised manuscript, the authors have addressed reasons why they chose the constant Tref but not the interannually varying Tref: (1) "Tref can be influenced by the advection water from the other box, not only the interannual variation of SST", (2) "a varying Tref would increase the weak values and decrease the strong values, hence reducing the interannual variability", and (3) "differences between UIs using constant Tref and interannually varying Tref are small". I think the main reason of the authors should be the third one and is reasonable; however, I still have two comments on the the choice of Tref.

- First. Tref in SST-based upwelling index represents the reference SST that the upwelling effect on this value is small. That is, the authors should re-consider the location of the reference box if "Tref is still strongly influenced by the advected upwelling water".

The reference box was chosen as far as possible from the upwelling zone and over the same latitude region. However, this choice was also constrained by the computational domain, i.e. the location of the eastern border. Figure 3 of the paper shows that on average, BoxTref is outside of the upwelling area. However we indeed showed that BoxTref could be influenced by the eastern advection of cold upwelled water during years of strong upwelling (Figure 17a of the paper and Figure B below). This would affect the computation of Tref, hence of the upwelling index. Taking a climatological average (i.e. a constant Tref) instead of an interannually varying Tref contributes to reduce this effect.

→ Following this comment, we added a paragraph in the text (Section 5.1, lines 626-629)

- Second. I don't agree with the second reason proposed by the authors that "using the interannual Tref will reduce the interannual variability of the upwelling". The interannual variability of the SCS SST is closely related to the ENSO events. In the cases of summers 2010 and 2018 that the authors had mentioned, summer 2010 was on the decay phase of the El Nino event 2009/2010, while summer 2018 followed the La Niña event 2017/2018 [Liu and Chu, 2019]. This fact explained why SST and Tref in 2018 were cooler than those in 2010, and ENSO is considered the main factor. If the same Tref is used for these years, upwelling is underestimated in 2010 but overestimated in 2018. Therefore, subtracting SST by time-varying Tref can help to eliminate climate variability coexisting in SST and Tref. In addition, due to the fact that the input SST for UI is daily data, using daily Tref consistently could make UI more accurate.

Indeed, the effect of ENSO on SST is clear and has been showed before (Qu et al. 2004, Wang et al. 2006) (note that despite our efforts we were unfortunately not able to find the paper of Liu and Chu 2019), with warmer surface following El Niño periods. However, Figure 17a of the paper also shows that the advection of upwelled water influences SST around the upwelling area. Moreover, previous studies also showed that ENSO influences the SVU, with weaker SVU following El Niño periods due to weaker monsoon winds (Da et al. 2019), implying weaker advection of cold water. Using a yearly-varying Tref partly eliminates the effect of climate variability, but not the effect of advection of upwelled water. Conversely, using a constant Tref reduces the effect of upwelled water advection, but does not take into

account the influence of large scale factors like ENSO. Choosing a constant or a yearly-varying Tref therefore involves making tradeoffs between those influence factors.

The choice of a seasonally averaged vs. daily varying Tref involves similar tradeoffs (lateral advection of upwelled water vs. seasonal variability). Using a daily Tref would indeed help to take into account the background variability of SST induced by large scale forcings, but it would also increase the effect of lateral advection. Figure B below indeed shows the daily SST for days 48 (peak of OFU, mid-July), 61 (end of July, period of OFU weakening) and 75 (2nd peak of OFU, mid-August) of the JJAS 2018 period : the SST in BoxRef is clearly influenced by the advection of water upwelled over BoxOF for in mid-August, and to a lesser extent, mid-July, and much less during day end of July. Using a daily varying Tref would therefore result in an overestimation of upwelling, over all boxes, during the weak upwelling period vs. an underestimation during upwelling peaks.

→ Following this comment of the reviewer, we acknowledged and discussed the fact that those choices of Tref involves tradeoffs in the revised version of our paper (Section 5.1, lines 632-641).

[Figure]

*Figure B : Daily SST simulated during day 48 (mid-July), 61 (end of July) and 75 (mid-August) of the 2018 June-September period (°C).*

2. Section 3
a) The authors used several sources of data to prove the occurrence of upwelling in BoxMK, and they convinced me. However, there is an issue that the percentage of covering days of JAXA over BoxMK in summer 2018 is low (below 60%), and on some days such as July 16th, the number of valid pixels in the satellite image is inadequate for a significant statistics, making values of minimum SST meaningless in Figure 5b. For this reason, I suggest the authors to use a scatterplot for JAXA as they did in Response, and mark values derived from the data with a low percentage of coverage by another color.

There are more than 15 000 JAXA pixels in BoxMK. We show in Figure C the daily time series of the percentage of pixels of quality level ≥ 4 over BoxMK. We also show the scatterplot of minimum SST, with days with a coverage lower than 50%, higher than 50% and higher than 60% highlighted. There are more than 60% of pixels of quality level ≥ 4 during 68 over the 122 days of summer 2018, i.e. 56 % of the time. Figure C shows in particular that at least 60% of BoxMK is covered with data of quality level ≥ 4 during the periods of low SST (mid-June, Mid to end July, Mid to end August). Those results show that at the first order, it is statistically meaningfull to use JAXA observations to assess the evolution of minimum SST during the summer period over BoxMK.

→ Following this comment, we included the figure with scatterplot in our revised version of the paper, and added a few lines of comments in the text (Section 3.1, lines 266-269 and Figure 5).

[Figure]

[Figure]

*Figure C : Daily time series of JAXA coverage of BoxMK (top, %) and of minimum SST over BoxMK in SYMPHONIE, OSTIA, COPERNICUS and JAXA (bottom, °C). For JAXA, we only consider data of quality level ≥ 4. Dots, crosses and stars correspond respectively to a coverage smaller than 50%, between 50% and 60% and above 60%*

b) Section 3.2: The length of this Section can be optimized by reducing the repetition of information that has been mentioned in the Introduction.

➔ Following this comment, and the detailed comments below, we reduced as much as possible the repetition of information already provided in the introduction (Section 3.2.1, lines 295-298 and 327-330).

c) The authors presented TS characteristics simulated by SYMPHONIE in the whole Section 3.3 (Line 353-401), with the aim of highlighting the capability of SYMPHONIE in simulating water masses in the coastal areas and open-sea as well. The description did not make any further contribution to upwelling analysis afterwards, which turns this Section into a disconnection message from the general structure of the paper.

This section was written in order to show that the model is able to realistically simulate water masses in the coastal and open-sea areas of our domain. It was even developed during the first revision, following the request of the 2nd reviewer. The 2nd reviewer indeed insisted on the fact that even if we use surface variables to quantify the strength of the SVU and investigate the factors involved in its interannual

variability, it is relevant to show that the model reproduces correctly the tridimensional water masses in the computational and study areas : this is a guarantee that the model reproduces realistically the SCS dynamics, including the surface dynamics that are influenced by vertical stratification and vertical dynamics. Note that including this evaluation in the paper will also be useful if the model is used, by us or by other authors willing to use our simulations, to investigate other processes than the SVU.

→ Following this comment, we decided that the comment of the 2nd reviewer was also reasonable and decided to keep section 3.3. We however added some text in the revised paper to justify this and take into account this comment of the reviewer here (section 3.3, lines 353-357)

3. Section 4
a) The authors used the spatial integral of the positive relative vorticity over BoxOF in JJAS ($\zeta+$) as an indicator of the intensity of the summer circulation (AC/C dipole + eastward jet) in the offshore region. I don't think this is a good indicator. The negative and positive vortices of a dipole do not always develop in conjunction, so the intensity of a dipole cannot be evaluated just by one side. For example, in the years 2011, 2013, and 2015, values of $\zeta+$ were similar but the dipole evolved in three different patterns. As a result, the direction and intensity of the jet in these years were various.

Circulation in the SVU region is characterized by the dipole structure and eastward jet. To represent this circulation, one can indeed use several indicators, and even combine them : positive relative vorticity over BoxOF but also negative vorticity, speed of the eastward jet, meridional position of the eastward jet, e.g. Ngo and Hsin 2021.... We chose the $\zeta_{+,OF}$ indicator because 1) it is an indicator of the intensity of the cyclonic activity that develops northern of the jet over BoxOF ; 2) it is an indicator of the strength of the jet (a stronger jet is characterized by a higher positive vorticity on its northern flank) ; 3) the cyclonic activity (both related to eddies and eastward jet) has been shown to influence OFU and SCU (Da et al. 2019).

We completely agree with the reviewer that this choice, as every choice of an integrated indicator, is associated with limitations. In particular, as underlined by the reviewer, it does not account for the spatial characteristics of the positive vorticity and does not allow to fully represent the full range of situations. This can be seen by examining the cases of years 2011, 2013 and 2015 in Figure 12 of the paper. All years show similar $\zeta_{+,OF}$ and similar JJAS wind stress intensity (Figure 13). However, the spatial distribution of this positive vorticity and the values of UIy bof BoxSC and BoxOF differ. 2011 shows a double dipole structure favoring the upwelling. 2013 shows an eastward jet located in the south and weak activity northern of the jet, not particularly favorable to upwelling. 2015 shows an eastward jet located in the north with a strong anticyclone located over BoxOF, preventing the upwelling.

To go beyond the limitations of this indicator, it is therefore necessary to examine into details the impact of spatial patterns of the mesoscale circulation on the upwelling. We began to do this in the present paper, and will go into much more details using ensemble simulations to examine the impact of OIV in a coming paper.

→ Following this comment, we added a paragraph to discuss the impact and limitations of the choice of this integrated indicator (Section 5.3 lines 705-712 and 721-722 and Section 4.1.2. line 443)

b) One of the important conclusions (or new findings) is that NCU is not primarily driven by the intensity of the summer wind over the SVU of BoxNC region, which is drawn due to the low correlation coefficient between UI of BoxNC and WS averaged over the same area (I "assume" the area is around [108.5-110E, 12-15N] because no detailed information was provided). Besides some issues in the definition of UI as mentioned above, I can try to figure out certain differences between their results and Ngo and Hsin (2021), which is the study they compared with.

Ngo and Hsin (2021) used the wind-based upwelling index to find that wind stress-induced upwelling-favorable condition changes correspondingly with the change of orientation of the coastline along the southern Vietnamese coast, and the roles of two components of wind stress (WSU and WSV) also change in driving Ekman transport along the coast. In the NCU, the favorable condition of wind stress for the upwelling is mainly contributed by the WSV because the WSV induces Ekman transport pushing water

offshore. Whereas, the WSU plays a role in restricting the development of the coastal upwelling (i.e. pulling water onshore). Thus, it appears clearly that the role of wind stress on the coastal upwelling cannot be illuminated if the total wind stress is just considered.

In addition, Ngo and Hsin (2021) analyzed the spatial distributions of correlation coefficients between SSTUI in their NCU with both WSU and WSV over the whole SCS, instead of only presenting the averaged WS over a chosen region, whose size or location is sensitive to the result of correlation analysis. Their results showed that the the SSTUI in the NCU is positively correlated with the local WSV (R=0.4~0.67) and has a negative remote connection with WSV in the southern SCS (R=-0.4~-0.7). Based on the above two points, I think that the role of wind on NCU has been investigated carefully by Ngo and Hsin (2021) and are more reliable than the the authors' argument unless the detailed evidence can be provided. Besides, the dissimilarity in the number of upwelling events in NCU (in other upwelling regions as well) also contributes to the variance in results between the two studies.

We warmly thank the reviewer for this comment : it encouraged us to examine into more details the relationship between wind stress and NCU. Following this comment and the analysis of Ngo and Hsin (2021), Figure D shows the map of average JJAS wind stress over the 2008-19 period, and the maps of correlations between yearly time series of UIy (for SCU, NCU and OFU) and JJAS wind stress intensity over BoxOF on one side, and the (spatially dependent) times series of JJAS wind stress components and intensity on the other side.

First, the intensity of JJAS wind stress over boxOF is highly significantly (p<0.01) positively correlated with the zonal and meridional components and the intensity of wind stress over most of the domain (Figure D-e). But it is negatively correlated with the zonal component and intensity of wind stress in the northeastern part of the domain (including BoxNC). The intensification of the summer monsoon wind over the region is thus associated with an intensification of the wind over most of the area, but a weakening in the northwestern part of its intensity and meridional component, i.e. a less northward and more eastward direction; and vice versa.

Second, correlations maps for UIy for OFU and SCU are very similar to the correlations maps for JJAS wind stress intensity over boxOF, with larger areas of highly significant correlations for SCU than for OFU (Figure D-c,d). This confirms the link, stronger for SCU than for OFU, between the summer intensity of upwelling over those regions and of the intensity of summer monsoon wind.

Third, the correlations maps for NCU are of opposite signs to the other maps. When the summer monsoon wind intensifies (favorable to SCU and OFU), the wind stress and its meridional component weaken in the northeastern region, where it becomes more eastward. The wind is therefore less parallel to the coast over BoxNC and more offshore oriented, i.e. not favorable to NCU. It is the opposite when the summer monsoon wind weakens. Wind conditions favoring OFU and SCU (and MKU) therefore prevent NCU, and vice versa.

Last, correlations are much more statistically significant (p<0.01) for SCU and, to a lesser extent, OFU, than for NCU (p<0.05 or 0.01 only in very small coastal areas over BoxNC). Those results therefore suggest that wind also participates to the interannual variability of NCU, with favorable conditions opposite of those favorable to SCU, OFU and MKU, as already showed by Ngo and Hsin (2021). However, they further reveal that the influence of wind is not as important as for the other areas, and that other factors may be playing roles of equal or stronger importance, in particular OIV and mesoscale structures.

→ Following this comment, we added a figure (Figure 16) in the document and paragraphs about this in the paper (Section 4.4, lines 540-564 and Section 5.4, lines 745-759 and 766-767), and modified the abstract (lines 37-38) and conclusion accordingly (lines 823-829).

[Figure]

*Figure D : (a) direction (arrows) and intensity (N.m-2) of average JJAS wind stress over the SVU area during 2009-2018 from ECMWF analysis. (b) to (e) Correlation between yearly time series of the zonal (left) and meridional (middle) components of wind stress and of its intensity (right) and (b) UIy,NC (c), UIy,SC (d), UIy,OF and (d) JJAS wind stress intensity over BoxOF. Thick, plain and dotted lines correspond respectively of the isolines p=0.01, p=0.05 and p=0.10.*

c) Influences of the alongshore and offshore currents on the NCU have a better presence. The finding that the enhancements of SCU and NCU could relate to the appearances of the secondary dipole east of BoxNC and the associated jet in-between is novel and interesting. However, the condition for the appearance of the secondary dipole and the associated jet is still unclear and needs more quantitative analysis to get a solid conclusion.

Indeed, in the previous version of the paper we did not investigate into details the conditions that favor/prevent the development of a secondary dipole and of the NCU. However, the answer to comment 3b above were we investigated into more details the link between NCU and the wind comments contributed to better understand this : wind in the NCU area with a strong eastward component contribute to prevent the development of NCU. Moreover, we performed sensitivity simulations on the 2018 case study that revealed that the ocean intrinsic variability also strongly triggers the development of those structures : with the same external conditions (atmospheric, oceanic and continental), simulations with perturbed initial conditions can result or not in secondary dipoles, due to the chaotic behavior of the submesoscale to mesoscale dynamics. For the sake of conciseness and clarity, those results were not presented here but a second paper is being written to present them.

→ Following this comment, we added a sentence in the text to mention the need to perform further studies to better understand factors triggering the development of the circulation, and secondary dipole, along the northern coast (Conclusion, lines 828-829).

Besides, the authors used the current in the upper 1m depth for their analysis, which is primarily driven by wind forcing. Can the current at the sea surface demonstrate properly (sub)mesoscale circulation, OIV, or ocean dynamics in the SCS?

To answer this question, we explored the vertical extension of surface mesoscale structures. Figure D below shows the example of August 1st, 2018. At this date, a large anticyclonic eddy develops near ~11°E / 8°N, and a smaller one near ~13°N, with a strong eastward current in between. We plot the horizontal velocity normal to sections A and B that cross those eddies : those eddy structures extend vertically, with high velocities (between 0.5 and 1 m.s$^{-1}$) until 50 to 100 m depth, and significant velocities (up to 0.2 m.s$^{-1}$) down to 500 m depth. Exploring other days of the summer period shows similar results. The surface mesoscale currents simulated by our model is therefore representative of the circulation over the surface and subsurface layer, and can therefore be used to examine (sub)mesoscale dynamics in the SCS.

→ Following this comment, we added a paragraph in the revised version of our paper (Section 5.3, lines 723-727), and added this figure in the supplementary materials (Figure SM2).

[Figure]

*Figure 17 : Simulated daily sea surface height (m, top) and velocity (m.s⁻¹) normal to sections A (bottom, left) and B (bottom, right) on August 1st, 2018. Positive (negative) values correspond to a normal velocity oriented to the north (south).*

d) In the MKU, the authors showed a strong relationship between UI and summer wind and summer offshore circulation based on high correlation coefficients; however, physical mechanisms for this upwelling region is lack.

Indeed, in this study we focused on the interannual scale, and wanted to cover the four boxes. For the sake of conciseness, we therefore chose not to explore here the functioning of the MKU, that our study revealed for the first time. However, it is indeed fundamental to perform detailed studies dedicated to the understanding of the physical mechanisms of this upwelling. This is an undergoing work in our group, where we are investigating the effects of different factors (wind, currents, topography, rivers, tides…). Our results will be presented in a coming paper that we are presently preparing.

→ Following this comment, we acknowledged the need to understand into details the physical mechanisms involved in the functioning and variability of the MKU (Conclusion, lines 841-843)

e) The authors revealed that "the daily to intra-seasonal chronology of wind stress contributes to the summer average of SCU and OFU intensity, and their interannual variability". I totally agree that but this is not a new finding. The summer SCU or OFU is calculated by averaging the daily SCU or OFU which has a strong correlation with the daily wind stress. Thus, summer SCU and OFU obviously are influenced by daily or intra-seasonal wind stress.
Indeed, previous authors already showed that upwelling develops under the influence of wind peaks. Liu et al. (2012) for example showed that on the intraseasonal time scale, SST cooling develops with the

evolution of the southwesterly wind anomalies with nearlyone week delay, and revealed the influence of MJO but also typhoons. We further show in our study that the chronology of those peaks matter : with the same summer average (see 2018 vs. 2009 and 2012), and even with the same July-August average (see 2012 vs. 2009 and comment below), involving strong wind peaks for the 3 cases, the upwelling is stronger if wind peaks occur regularly during the July-August period, with peaks spaced by a sufficient period to let the upwelling develop and maintain it.

→ Following this comment, we acknowledge the fact that previous authors already highlighted the role of intraseasonal to daily chronology, and better explain that we further highlighted here the role of the timing of the wind peaks itself (Introduction, lines 96-98, Section 5.2, lines 661-669 and Conclusion, lines 816-818 and 821-822).

f) By examing three summers of 2009, 2012, and 2018, the authors concluded that the summer OFU is stronger when wind peaks occur during the core of the summer season (July-August) than at the beginning (June) or end (September). My question is whether it true for every year when the summer monsoon matures in July-August?

This comment is partly related to the comment above. Indeed, at the second order, other factors are certainly involved, as can be deduced by considering again the cases of 2009 vs. 2012 (Figure 15 of the paper). The wind stress over Box OF is stronger in June and August in 2012 than in 2009, and similar in July, and the JJAS vorticity $\zeta_{+,OF}$ is similar (Figure 15 of the paper). However, OFU is 50% weaker in 2012 than in 2009 on the yearly average (Figures 13,15 of the paper). A first factor explaining this difference could be the daily chronology of wind inside the July-August period : 2009 shows 2 distinct wind maximum periods in mid-July and mid-August, whereas 2012 shows 2 close peaks during the $2^{nd}$ half of July. Those results suggest that the wind chronology not only at the intraseasonal scale but even at the daily scale inside the July-August period affects the development of the upwelling. The fact that the summer monsoon matures in July-August is therefore important, but is not the only factor. This result is supported by previous studies (*Xie et al. 2007, Liu et al. 2012),* who highlighted the effect of intraseasonal wind variability including the effect of Madden Julian Oscillation and typhoons. The impact of OIV (ocean intrinsic variability) could also contribute to the difference, as shown by Da et al. (2019). Studies including sensitivity ensemble simulations testing in particular the role of those factors will allow to better understand the intraseasonal variability of the upwelling. This will be the scope of coming studies based in particular on the study of the strong SVU 2018 case study.

→ Following this comment of the reviewer, we developed the discussion concerning the impact of intraseasonal to daily variability of wind, underlining the need to explore those factors in dedicated sensitivity simulations (Section 5.2, lines 661-669 and Conclusion, lines 821-822). The need to develop studies about OIV was already mentioned in the previous version of the paper.

4. Section 5.2
The authors performed a simulation using summer wind stress in 2018 to exam role of intraseasonal variability of atmospheric forcing on the SVU. I think this part should be presented in another paper with more details on input wind data and the modelling results.
This simulation was performed following the recommendation of the other reviewer, who requested us to quantitatively highlight the importance of intraseasonal variability of wind. Following this comment, we therefore did not removed completely this part, but reduced it and explained that it will be examined into more details in further studies (Section 5.2, lines 671-681).

Minor Comments:

Line 70: "Ngo et al. 2021" should be "Ngo and Hsin 2021"
→ this was corrected

Line 220-221: "both in observed and simulated …" should be "both in observed and in simulated …"
→ this was corrected

Line 255: "vs." is not suitable here.

➔ this was replaced by "whereas it reaches"

Line 281: "the SSH temporal average over 2009-2018" should be "the averaged SSH over 2009-2018"
➔ this was corrected

Line 413-414: should be " … horizontal surface current rotation"
➔ this was corrected

Line 288: remove "very"
➔ this was removed

Line 291-299: "Under …2000" and "In winter … datasets". These sentences prolong the paragraph unnecessarily.
➔ These sentences were written to comment the main patterns of SST. Following this comment they were strongly shortened in the revised version of the paper

Line 327-331: "As explained …Figure 2k-l". I think this part should be shortened.
➔ those sentences were shortened

Line 424: "….and related to the summer monsoon wind". This is an obvious argument.
➔ indeed, but we kept it for the sake of readers who are not specialist of this area

Line 469: "all" should be replaced by "both"
➔ this was corrected

Line 485: "double" should be removed.
➔ this was removed

Line 485: "classically". This word is not appropriate when describing the dipole and the eastward jet.
➔ this was removed

**References**

Da, N. D., Herrmann, M., Morrow, R., Niño, F., Huan, N. M., and Trinh, N. Q. (2019). Contributions of wind, ocean intrinsic variability, and ENSO to the interannual variability of the south vietnam upwelling: A modeling study. Journal of Geophysical Research: Oceans, 124(9), 6545–6574. https://doi.org/10.1029/2018jc014647

Da (2018). The interannual variability of the South Vietnam Upwelling : contributions of atmospheric, oceanic, hydrologic forcing and the ocean intrinsic variability. PhD thesis, University of Toulouse, Toulouse.

Liu, X., J. Wang, X. Cheng, and Y. Du (2012), Abnormal upwelling and chlorophyll-a concentration off South Vietnam in summer 2007, J. Geophys. Res., 117, C07021, doi:10.1029/2012JC008052.

Ngo, M., and Hsin, Y. (2021). Impacts of wind and current on the interannual variation of the summertime upwelling off southern Vietnam in the South China Sea. Journal of Geophysical Research: Oceans, 126(6). https://doi.org/10.1029/2020jc016892

Qu, T., Y. Y. Kim, M. Yaremchuk, T. Tuzuka, A. Ishida, and T. Yamagata (2004), Can Luzon Strait transport play a role in conveying the impact of ENSO to the South China Sea?, J.Clim., 17(18), 3644–3657, doi:10.1175/1520-0442(2004)017<3644:CLSTPA>2.0.CO;2.

Xie, S.-P., Chang, C.-H., Xie, Q., and Wang, D. (2007). Intraseasonal variability in the summer South China Sea: Wind jet, cold filament, and recirculations. Journal of Geophysical Research, 112(C10). https://doi.org/10.1029/2007jc004238